# PROVABLE LEARNING FOR DEC-POMDPS: FACTORED MODELS AND MEMORYLESS AGENTS

## ABSTRACT

This paper studies cooperative Multi-Agent Reinforcement Learning (MARL) under the mathematical model of Decentralized Partially Observable Markov Decision Process (DEC-POMDP). Despite the empirical success of cooperative MARL, its theoretical foundation, particularly in the realm of provable learning of DEC-POMDPs, remains limited. In this paper, we first present a hardness result in theory demonstrating that, without additional structural assumptions, learning DEC-POMDPs requires several samples that grows exponentially with the number of agents in the worst case, which is also known as the curse of multiagency. This motivates us to explore important subclasses of DEC-POMDPs for which efficient solutions can be found. Specifically, we propose new algorithms and establish sample-efficiency guarantees that break the curse of multiagency, for finding both local and global optima in two important scenarios: (1) when agents employ memoryless policies, selecting actions based solely on their current observations; and (2) when a factored structure is present, which enables key properties similar to value decomposition in VDN or Qmix.

## 1 INTRODUCTION

Cooperative multi-agent reinforcement learning (MARL) has gained significant attention due to its wide range of applications in real-world problems, such as autonomous vehicles and robotic swarms. However, one of the primary challenges in cooperative MARL is the exponential growth of the action space as the number of agents increases. Consequently, the number of samples required to learn an optimal policy grows exponentially with the number of agents. To address this issue, Oliehoek et al. (2008a); Kraemer and Banerjee (2016) proposed the method of centralized training with decentralized execution (CTDE). In CTDE, agents' policies are trained using global information, but the resulting policies are decentralized, allowing each agent to execute them using only local information during execution. Rashid et al. (2018) introduced the decentralized partially observable Markov decision process (DEC-POMDP) as a model for fully cooperative multi-agent reinforcement learning. The DEC-POMDP model is similar to a traditional partially observable Markov game, with the key distinction that in a DEC-POMDP, the objective is to maximize the cumulative rewards across all agents. Although cooperative MARL has achieved empirical success across a variety of real-world problems, and numerous significant algorithms have been developed—such as VDN (Sunehag et al., 2017), Q-MIX (Rashid et al., 2018), and Q-PLEX (Wang et al., 2021)—the theoretical foundations of MARL remain underdeveloped. Therefore, the objective of this paper is to develop a sample-efficient algorithm for DEC-POMDPs, with an emphasis on theoretical rigor and comprehensive guarantees.

This paper considers two types of optimality: (1) **global optimality**, where agents seek policies that maximize the total cumulative reward across all agents; and (2) **local optimality**, where each agent aims to find a policy that maximizes total cumulative reward, assuming the policies of all other agents are fixed. The latter concept is commonly referred to as the Nash equilibrium (Kreps, 1989) in game theory.

The challenge of learning in DEC-POMDPs arises from two primary factors. First, agents have access only to their individual trajectory histories, making it difficult to identify either global or local optima, both of which require cooperation among agents. Second, the joint action and observation space grows exponentially with the number of agents, creating significant obstacles in

designing algorithms with polynomial sample complexity, particularly as the number of agents increases. We illustrate these challenges by first proving statistical **hardness results** for learning DEC-POMDPs without additional structural assumptions (Theorem 5.2). This motivates us to focus on important subclasses of DEC-POMDPs that are rich enough to encompass practical applications but constrained enough to admit sample-efficient algorithmic solutions. Specifically, we consider the following two subclasses:

**Factored Structure Model**: Motivated by the value decomposition approach, which has been widely employed in practical algorithms such as VDN and Qmix, we consider DEC-POMDPs with a factored structure that enables key properties similar to value decomposition. We demonstrate that a factored value decomposition property holds for our model and develop algorithms with provable efficiency to achieve both local and global optimality. Specifically, the value decomposition-like conditions in our model allow the total action-value function to be decomposed into distinct components, each dependent only on the trajectory of one or a small subset of agents. As a result, each agent only needs to focus on a limited number of components when making decisions, which mitigates the exponential scaling of complexity as the number of agents increases.

**DEC-POMDP with Memoryless Policy**: We first investigate a model frequently studied in partially observable settings (Kara and Yuksel, 2022; Kara and Yüksel, 2023), where agents determine their actions based solely on their current observations. We propose a sample-efficient algorithm tailored to achieve local optimality in this context.

**Our Contribution**: Our contributions are centered around the development of a sample-efficient algorithm under reasonable conditions. We summarize our key contributions and results as follows:

1. We demonstrate that, without further assumptions, designing a sample-efficient algorithm to achieve global optimality in DEC-POMDPs is infeasible, regardless of whether agents employ general policies or are restricted to memoryless policies.
2. In the setting where agents use memoryless policies, we propose a sample-efficient algorithm that achieves local optimality.
3. For the factored structure model, we prove that the value function can be decomposed into components that depend only on the trajectories of a few agents rather than all agents. We also introduce a sample-efficient algorithm that can achieve both local and global optimality.
4. Our analysis of the factored structure model provides a partial theoretical explanation for empirical algorithms such as VDN, as we establish a sufficient condition for value decomposition and offer a sample-efficient guarantee under this condition.

## 2 RELATED WORK

Due to space limits, we briefly present a few previous works closely related to this paper and leave the comprehensive discussion on additional related work in the appendix.

**Learning POMDPs** planning in POMDPs is known to be PSPACE-complete (Papadimitriou and Tsitsiklis, 1987; Littman, 1994; Burago et al., 1996; Lusena et al., 2001). Uehara et al. (2022) impose a deterministic latent transition assumption on the model and design computationally efficient algorithms. Jin et al. (2020) design the observable operator model with the upper confidence bound algorithm for weakly revealing POMDPs, while Liu et al. (2022a) propose the optimistic maximum likelihood estimation (OMLE) algorithm for learning weakly revealing POMDPs. Chen et al. (2022) derive a unified analysis for OMLE with a sharper sample complexity. Furthermore, Liu et al. (2023) provides a generic framework for applying OMLE to a wide range of partially observable problems, including low-rank sequential decision-making problems and general sequential decision-making problems under the SAIL condition. Since OMLE learns the near-optimal policies of an enormously rich class of sequential decision-making problems in a polynomial number of samples, we also build our work upon the generic framework of OMLE.

**Learning DEC-POMDPs** Rashid et al. (2018) introduced the decentralized-partially observable Markov decision process (DEC-POMDP) as a fully cooperative multiagent reinforcement learning task. Empirical algorithms for solving DEC-POMDP include VDN (Sunehag et al., 2017), Q-MIX (Rashid et al., 2018), and Q-PLEX (Wang et al., 2021). Recent works on DEC-POMDP, such as those by Hu and Foerster (2019), Foerster et al. (2019), and Lerer et al. (2020), adopt ideas similar to the common information approach, leading to breakthroughs in challenging DEC-POMDP problems

like Hanabi. The common information approach of Dec-POMDP has been studied theoretically in (Zhang et al., 2019; Mao et al., 2023; Liu and Zhang, 2023). In particular, (Liu and Zhang, 2023) establish a sample quasi-efficient algorithm with quasi-polynomial sample complexity. In comparison, by imposing additional structural assumptions, our algorithms have polynomial sample complexity. Moreover, our lower bound shows that polynomial sample complexity is impossible without additional assumptions.

## 3 PROBLEM FORMULATION

### 3.1 PRELIMINARY

**Partially Observable Model:** We study the decentralized partially observable Markov decision process (DEC-POMDP)(Rashid et al., 2018) with $n$ agents, which is denoted by a tuple $(\mathcal{S}, \mathcal{O}, \mathcal{A}; H, \mu_1, \mathbb{T}, \mathbb{O}; r)$. Here $\mathcal{S}$ is the set of all possible states, where the states are not observable by the agents. For each agent $m \in [n]$, we let $\mathcal{A}_m$ and $\mathcal{O}_m$ denote the action and observation space of agent $m$ respectively. We define the joint action space and the joint observation space by $\mathcal{A} := \mathcal{A}_1 \times \cdots \times \mathcal{A}_n$ and $\mathcal{O} := \mathcal{O}_1 \times \cdots \times \mathcal{O}_n$, respectively. Besides, $H$ is the episode length, $\mu_1$ is the distribution of the initial state $s_1$, $\mathbb{T} = \{\mathbb{T}_{h,\mathbf{a}}\}_{(h,\mathbf{a}) \in [H-1] \times \mathcal{A}}$ is the Markov state transition kernel, $\mathbb{O} = \{\mathbb{O}_{h,m}\}_{h \in [H], m \in [n]}$ is the observation emission kernel, and $r = \{r_{h,m}\}_{h \in [H], m \in [n]}$ denotes the reward function. In particular, starting from $s_1 \sim \mu_1$, at each step $h \in [H]$, each agent $m$ observes an observation $o_{h,m} \in \mathcal{O}_m$ according to distribution $\mathbb{O}_{h,m}(\cdot \mid s_h)$, takes an action $a_{h,m}$, and receives a reward $r_{h,m}(o_{h,m}) \in [0,1]$, which is is a function of $o_{h,m}$. We consider this type of reward function to prevent information about latent states from leaking through rewards beyond what is provided by the observations. Let $\mathbf{a}_h = (a_{h,1}, a_{h,2}, \ldots, a_{h,n})$ denote the joint action at step $h$. The next state $s_{h+1}$ is sampled from distribution $\mathbb{T}_{h,\mathbf{a}_h}(\cdot \mid s_h)$. Such a process terminates when $s_{H+1}$ is reached. For each agent $m$, it collects data $\{o_{h,m}, a_{h,m}, r_{h,m}\}_{h \in [H]}$. Notably, each agent has access only to their own observation and reward, and therefore does not know the total payoff. The goal of the agents is to maximize the social welfare, i.e. the summation of cumulative rewards obtained by all the $n$ agents. Moreover, to simplify the notation, we let $S$, $A$, and $O$ denote $|\mathcal{S}|$, $\max_{m \in [n]} |\mathcal{A}_m|$, and $\max_{m \in [n]} |\mathcal{O}_m|$, respectively.

**DEC-POMDP:** In this work, we study the case where agents adopt decentralized policies. Namely, each agent only selects actions based on the history of her own trajectories. The joint policy $\boldsymbol{\pi}_h$ represents the decentralized policy product of the $n$ agents. We formally denote the policy class as $\boldsymbol{\pi} = \{\{\otimes_{m=1}^n \pi_{h,m}\}_{h \in [H]} \mid \pi_{h,m} : (\mathcal{O}_m \times \mathcal{A}_m)^{h-1} \times \mathcal{O}_m \to \Delta_m\}$. When considering a product policy $\boldsymbol{\pi}$, we denote its value function as $V^{\boldsymbol{\pi}}$, defined as the expected total reward received by all agents under policy $\boldsymbol{\pi}$:

$$V^{\boldsymbol{\pi}} = \mathbb{E}_{\boldsymbol{\pi}}\left[ \sum_{h=1}^H \sum_{m=1}^n r_{h,m}(o_{h,m}) \right].$$

$\forall \boldsymbol{\tau}_h = (\mathbf{o}_1, \mathbf{a}_1, \ldots, \mathbf{a}_h)$, we define the $Q$-function as $Q^{\boldsymbol{\pi}}(\boldsymbol{\tau}_h) = \mathbb{E}_{\boldsymbol{\pi}}[\sum_{j=h}^H \sum_{m=1}^n r_{j,m}(o_{j,m}) \mid \boldsymbol{\tau}_h]$. For each agent $m \in [n]$ and step $h \in [H]$, we define the trajectory notation of the $m^{\text{th}}$ agent as $\tau_{h,m} = (o_{1,m}, a_{1,m}, \ldots, o_{h,m}, a_{h,m})$.

**Learning Target:** We define two types of optimality as learning objectives for DEC-POMDPs: global optimality and local optimality. Their formal definitions are provided as follows:

**Definition 3.1** (Local Optimality). *A policy* $\boldsymbol{\pi} = \pi_1 \times \pi_2 \times \cdots \times \pi_n$ *is considered a local optimal policy for agents* $1, 2, \ldots, n$ *if it satisfies* $V^{\boldsymbol{\pi}} = \max_{i \in [n]}[\max_{\pi_i'} V^{\pi_i', \pi_{-i}}]$.

Our aim is to minimize the number of samples required to obtain an $\epsilon$-approximate local optimal policy. We define an $\epsilon$-approximate local optimal policy as a policy $\boldsymbol{\pi} = \pi_1 \times \pi_2 \times \cdots \times \pi_n$ that satisfies

$$V^{\boldsymbol{\pi}} \geq \max_{i \in [n]} \left[ \max_{\pi_i'} V^{\pi_i', \pi_{-i}} \right] - \epsilon.$$

**Definition 3.2** (Global Optimality). *A policy* $\boldsymbol{\pi} = \pi_1 \times \pi_2 \times \cdots \times \pi_n$ *is deemed a global optimal policy for agents* $1, 2, \ldots, n$ *if it satisfies* $V^{\boldsymbol{\pi}} = \max_{\pi_1', \pi_2', \ldots, \pi_n'} V^{\pi_1' \times \pi_2' \times \cdots \times \pi_n'}$.

Our goal is to minimize the regret, which is defined as

$$\text{Regret}(T) = \sum_{k=1}^T (V^{\boldsymbol{\pi}^*} - V^{\boldsymbol{\pi}^k}),$$

where $\boldsymbol{\pi}^* = \arg\max_{\boldsymbol{\pi}} V^{\boldsymbol{\pi}}$. We assume that agents interact with DEC-POMDPs for $T$ episodes, and in the $k$-th iteration for any $k \in [T]$, they follow the policy $\boldsymbol{\pi}^k = \pi_1^k \times \pi_2^k \times \cdots \times \pi_n^k$. Similarly, we can define an $\epsilon$-approximate global optimal policy in the local context, where a uniform mixture of $\boldsymbol{\pi}^1, \ldots, \boldsymbol{\pi}^T$ satisfies the definition when sub-linear regret is achieved.

**Weakly Revealing Condition:** Liu et al. (2022a) demonstrated that without any assumption on the model, there exist hard instances such that the number of samples required to learn an $\epsilon$-approximate optimal policy in single-agent POMDPs is exponential in the horizon length $H$. Given the difficulty of learning POMDPs without assumptions on the model, even in single-agent settings, we consider the weakly revealing condition. This assumption is commonly adopted in previous works on partially observable contextual settings (Jin et al., 2020; Liu et al., 2022a; Chen et al., 2022).

**Assumption 3.1.** *We define $\mathbb{O}_h^i$ as $\mathbb{O}_h^i(o_{h,i} \mid s_h) = \sum_{\{o_{h,j}\}_{j \in [n]/\{i\}}} \mathbb{O}_h(\mathbf{o}_h \mid s_h)$. There exist $\alpha > 0$ such that $\min_{h,i} \sigma_S(\mathbb{O}_h^i) \geq \alpha$, where for matrix $\mathbf{A}$ we use $\sigma_S(\mathbf{A})$ to denote the $S^{th}$ singular value of emission matrix $\mathbf{A}$.*

This condition guarantees that, with enough samples, the observations provide adequate information to differentiate between any two combinations of states.

**Additional Notations:** Throughout this paper, we adopt the following notation for sets of elements with subscripts: Let $\mathcal{R} = \{x_i\}_{i \in \mathcal{S}}$, where $\mathcal{S}$ denotes the set of subscripts of the elements in $\mathcal{R}$. For simplicity, we represent $\mathcal{R}$ as $\mathcal{R} = x_{\mathcal{S}}$.

### 3.2 Hardness Result for General DEC-POMDP

The following theorem demonstrates that, in the absence of specific assumptions on the model, achieving global optimality in DEC-POMDPs is not possible with a sample complexity that is not exponential in the number of agents.

**Theorem 3.1.** *For any randomized or deterministic algorithms, there exists an instance of DEC-MDP wherein the regret scales at least as $\Omega(\sqrt{A^n T})$.*

This result highlights the limitations of achieving sample efficiency in algorithms for DEC-POMDPs without making assumptions about the transition model. Consequently, in the following sections, we aim to develop a sample-efficient algorithm for DEC-POMDPs under reasonable assumptions about the model.

## 4 Learning DEC-POMDP with Factored Structure Model

### 4.1 Factored Structure Model

In this section, we consider a factored structure model, where the state space is decomposed as the Cartesian product of $n$ individual spaces, $\mathcal{S} = \mathcal{S}_1 \times \mathcal{S}_2 \times \cdots \times \mathcal{S}_n$. For all $\mathbf{s}' = (s_1', \ldots, s_n') \in \mathcal{S}$, $\mathbf{s} = (s_1, \ldots, s_n) \in \mathcal{S}$, $\mathbf{a} \in \mathcal{A}$, $\mathbf{o} \in \mathcal{O}$, and $h \in [H]$, the observation distribution and transition probability are factorized as:

$$\mathbb{O}_h(\mathbf{o} \mid \mathbf{s}) = \prod_{m=1}^n \mathbb{O}_{h,m}(o_m \mid s_m), \quad \mathbb{T}_h(\mathbf{s}' \mid \mathbf{s}, \mathbf{a}) = \prod_{m=1}^n \mathbb{T}_{h,m}(s_m' \mid s_m, a_m, a_{\mathsf{pa}(m)}),$$

where $\mathsf{pa}(m) \subset [n]$ represents the set of agents whose actions influence the transition of agent $m$. We further define $\overline{\mathsf{pa}}(m) = \mathsf{pa}(m) \cup \{m\}$. We assume that the local state transition of each individual agent depends solely on the actions of other agents, with no dependency between the states of different agents.

To represent the correlation between different agents, we introduce the following influence graph:

**Definition 4.1.** *We define a directed graph $G = (\mathcal{V}, \mathcal{E})$, where $\mathcal{V} = \{1, 2, \ldots, n\}$, and there is a directed edge from vertex $i$ to vertex $j$ for distinct vertices $i, j \in \mathcal{V}$ if and only if $i \in \mathsf{pa}(j)$. Additionally, we assume that the maximum indegree of the graph $G$ is $d$.*

Additionally, for clarity in presentation, we introduce several notations from graph theory:

**Definition 4.2.** *For each agent $m \in [n]$, we define two sets: the children set $\mathsf{ch}(m)$ and the ancestor set $\mathsf{an}(m)$. Specifically, for a vertex $i \in [n]$, if there exists a directed path $i = j_1, j_2, \ldots, j_l = m$*

*in the influence graph $G$, where there is a directed edge from $j_r$ to $j_{r+1}$ for all $r \in [l-1]$, then $i \in \mathsf{an}(m)$ and $m \in \mathsf{ch}(i)$. Moreover, for all $m \in [n]$, we define $\overline{\mathsf{ch}}(m) = \{m\} \cup \mathsf{ch}(m)$ and $\overline{\mathsf{pa}}(m) = \{m\} \cup \mathsf{pa}(m)$. We also define the complement set of $\overline{\mathsf{ch}}(m)$ as $\mathsf{nch}(m) = [n] \setminus \overline{\mathsf{ch}}(m)$.*

The factored structure of the model leads to the following property of value decomposition.

**Proposition 4.1.** *(**Value Decomposition**) For all $h \in [H]$ and trajectory $\boldsymbol{\tau}_h = (\mathbf{o}_1, \mathbf{a}_1, \ldots, \mathbf{o}_h, \mathbf{a}_h)$, the Q-function can be decomposed as follows:*

$$Q^{\boldsymbol{\pi}}(\boldsymbol{\tau}_h) = \sum_{m=1}^n Q_m(\tau_{h,\overline{\mathsf{an}}(m)}),$$

*where we define $Q_m(\tau_{h,\overline{\mathsf{an}}(m)}) = \mathbb{E}_{\pi_{\overline{\mathsf{an}}(m)}}[\sum_{j=h}^H r_{j,m}(o_{j,m}) \mid \tau_{h,\overline{\mathsf{an}}(m)}]$. In other words, the value function can be expressed as the sum of $n$ terms, where the $m$-th component depends only on the trajectory of the agents in $\overline{\mathsf{an}}(m)$.*

For each $i \in [n]$, we denote $\theta_i = (\mathbb{T}_i, \mathbb{O}_i, \mu_i)$ as the collection of parameters representing the transition and observation models of the $i$-th agent. We further use $\Theta_i$ to denote the set of all possible model parameters $\theta_i$. According to the factored structure condition, the joint trajectory probability can be rewritten as the product of individual trajectory operators, where the individual operator $\mathbb{P}_{\theta_i}^{\pi_i}(\tau_{H,i} \mid \tau_{H,\mathsf{pa}(i)})$ is defined as:

$$\mathbb{P}_{\theta_i}^{\pi_i}(\tau_{H,i} \mid \tau_{H,\mathsf{pa}(i)}) = \sum_{s_{[H],i}} \mu(s_{1,i}) \mathbb{O}_{1,i}(o_{1,i} \mid s_{1,i}) \pi_{1,i}(a_{1,i} \mid o_{1,i})$$

$$\cdot \left[ \prod_{h=1}^{H-1} \mathbb{T}_{h,i}(s_{h+1,i} \mid s_{h,i}, a_{h,\overline{\mathsf{pa}}(i)}) \mathbb{O}_{h+1,i}(o_{h+1,i} \mid s_{h+1,i}) \pi_{h+1,i}(a_{h+1,i} \mid \tau_{h-1,i}, o_{h,i}) \right]. \tag{1}$$

We further use $\{\theta_i^*\}_{i \in [n]}$ to denote the model parameters of the true transition model.

### 4.2 Achieving Global Optimality with Factored Structure

In this section, we introduce a sample-efficient algorithm (outlined in Algorithm 4.2) to achieve global optimality under the factored structure model.

**Algorithm Description:** Algorithm 4.2 consists of three main steps in each episode $k$:

- *Update policy and parameters (Line 3)*: We construct $n$ distinct confidence intervals, where the $i$-th confidence interval contains only the parameters of the $i$-th agent's transition and observation model. The total value function is considered as a function of the joint product policy of all agents and the model parameters. We select model parameters $\theta_i^k$ from the $i$-th confidence interval, along with a joint product policy $\boldsymbol{\pi}^k$, such that the value function is maximized. After selecting the policy, we iteratively execute the following two steps for each agent $i \in [n]$ to collect samples and update the confidence intervals.

- *Sample trajectories for Agent $i$ (Lines 6-8)*: We sample trajectories for different agents according to two distinct distributions. At step $h$, all agents initially select an action according to their policies. For agent $m \in [i]$, an observation sample is directly collected from the true model. For the remaining agents $m \in [n] \setminus [i]$, we denote $\mathbb{T}_{h,m}^k$ and $\mathbb{O}_{h,m}^k$ as the transition and observation models corresponding to the parameter $\theta_k$. Given that model $\mathbb{T}_{h,m}^k$ and $\mathbb{O}_{h,m}^k$ are known, agent $m$ samples $s_{h+1,m} \sim \mathbb{T}_{h,m}^k(\cdot \mid s_{h,m})$. Subsequently, agent $m$ collects an observation $o_{h+1,m} \sim \mathbb{O}_{h+1,m}^k(\cdot \mid s_{h+1,m})$ and stores a dummy state $s_{h+1,m}$ for exploration in the next episode.

- *Update confidence interval for agent $i$ (Line 10)*: After collecting the trajectories $\tau_m^k = (o_{1,m}^k, \ldots, o_{H,m}^k, a_{H,m}^k)$ for all $m \in [n]$, we add the tuple $(\pi_i^k, \tau_{\overline{\mathsf{pa}}(i)}^k)$ to the sample set $\mathcal{D}_i$. The confidence set is then updated according to:

$$\mathcal{B}_i^{k+1} = \Big\{ \hat{\theta}_i \in \mathcal{B}_i^1 : \sum_{(\pi_i, \tau_{\overline{\mathsf{pa}}(i)}) \in \mathcal{D}_i} \log \mathbb{P}_{\hat{\theta}_i}^{\pi_i}(\tau_i \mid \tau_{\mathsf{pa}(i)}) \geq \max_{\theta_i' \in \Theta_i} \sum_{(\pi_i, \tau_{\overline{\mathsf{pa}}(i)}) \in \mathcal{D}_i} \log \mathbb{P}_{\theta_i'}^{\pi_i}(\tau_i \mid \tau_{\mathsf{pa}(i)}) - \beta_i \Big\}. \tag{2}$$

That is, we include those model parameters $\theta_m$ for which the total log-likelihood assigned to the data is close to the maximum possible total log-likelihood.

**Technical Challenge and Insights:** The dimensionality of the model grows as $\Omega(A^n O^n)$, since the joint action and observation spaces expand exponentially with the number of agents. Consequently, the sample complexity for estimating the model parameters is susceptible to $\Omega(A^n O^n)$. To

address this challenge, we construct separate confidence intervals for estimating the different model parameters. This approach mitigates the exponential sample complexity in $n$, as the dimension of each parameter $\theta_m$ does not increase exponentially with $n$. Additionally, we employ a carefully designed sampling procedure (outlined in lines 6–8) instead of directly sampling from the true transition model. This enables us to precisely control the statistical error in estimating the joint trajectory probability by separately managing the statistical error of each individual trajectory operator.

---

**Algorithm 1** OMLE for Achieving Global Optimal in Factored Structure Model

---

1: **Initialize**: $\mathcal{B}_m^1 = \{\hat{\theta}_m \in \Theta_m : \min_h \sigma_S(\mathbb{O}_m(\hat{\theta}_m)) \geq \alpha)\}$, $\mathcal{D}_m = \{\}$, for all agents $m \in [n]$.
2: **for** $k = 1 \ldots T$ **do**
3:     Compute $(\theta_1^k, \theta_2^k, \ldots, \theta_n^k, \boldsymbol{\pi}^k) = \arg\max_{\hat{\theta}_1 \in \mathcal{B}_1^k, \hat{\theta}_2 \in \mathcal{B}_2^k, \ldots, \hat{\theta}_n \in \mathcal{B}_n^k, \boldsymbol{\pi}} V^{\boldsymbol{\pi}}(\hat{\boldsymbol{\theta}})$.
4:     **for** $i = 1 \ldots, n$ **do**
5:         **for** $h = 1, \ldots, H$ **do**
6:             Selects an action according to $a_{h,m}^k \sim \pi_{h,m}^k(\cdot \mid \tau_{h-1,m}, o_{h,m})$ for all $m \in [n]$.
7:             For agent $m \in [i]$, collect observation $o_{h+1,m}^k$ from the environment.
8:             For agent $m \in [n]\backslash[i]$, samples dummy state $s_{h+1,m} \sim \mathbb{T}_{h,m}^k(\cdot \mid s_{h,m}, a_{h,\overline{\mathsf{pa}}(m)})$.
9:             Collect observation $o_{h+1,m}^k \sim \mathbb{O}_{h+1,m}^k(\cdot \mid s_{h+1,m})$ for agent $m \in [n]\backslash[i]$.
10:        Add $(\pi_i^k, \tau_{\overline{\mathsf{pa}}(i)}^k)$ into $\mathcal{D}_i$, and then update $\mathcal{B}_i^{k+1}$ with eq. (2).

---

**Theorem 4.1.** *For all $m \in [n]$, we select bonus parameter as $\beta_m = H^2(S^2A^{|\overline{\mathsf{pa}}(m)|} + SO)\log(TSAOH) + \log(Tn/\delta)$ for some constant c. Then, with probability at least $1 - \delta$, Algorithm 4.2 guarantees that the following inequality holds.*

$$Regret(k) = \sum_{t=1}^k V^{\boldsymbol{\pi}^*} - V^{\boldsymbol{\pi}^t} \leq \tilde{\mathcal{O}}\Big(\alpha^{-2}S^2OA^{d+1}\sqrt{k(S^2A^{d+1} + SO)}\Big), \forall k \in [T], \quad (3)$$

*where we define $\boldsymbol{\pi}^* = \arg\max_{\boldsymbol{\pi}} V^{\boldsymbol{\pi}}$, and recall that $d$ denotes the maximum in-degree of $G$.*

**Remark 4.1.** *The term $\sqrt{S^2A^{d+1} + SO}$ in 3 arises from the model error, while the additional $OA^{d+1}$ terms result from the statistical error related to the eluder dimension. The model dimension of the factored model scales exponentially with $d$. Consequently, the regret also scales exponentially in $d$, as we incur a model estimation error of $\mathcal{O}(A^d)$. Notably, when $d = \mathcal{O}(1)$, the regret is bounded by $poly(S, A^{\mathcal{O}(1)}, O, H, \alpha^{-1}, \log(\delta^{-1}T))$.*

**Theorem 4.2.** *(Lower Bound) For any randomized or deterministic algorithms, there exists an instance of DEC-POMDP with a factorization structure such that the regret for achieving global optimal is at least $\Omega(\sqrt{A^{d+1}T})$.*

The regret scales as $\Omega(\sqrt{A^{d+1}})$ since the model dimension scales as $\Omega(\sqrt{A^{d+1}})$. Therefore, Theorem 4.2 demonstrates that the dependence on the model's dimension is unavoidable.

## 4.3 Achieving Local Optimality with Factored Structure

In this section, we derive theoretical guarantees for achieving local optimality within a factored model. Since local optimality is a specific case of global optimality, we can directly apply Algorithm 4.2 with minor modifications to achieve $\epsilon$-local optimality with a sample complexity of $K = \tilde{\mathcal{O}}(S^4A^{2d+2}(S^2A^{d+1} + SO) \cdot poly(H)/(\alpha^4\epsilon^3))$. However, we demonstrate that a more refined analysis is possible to further improve the sample complexity. We present algorithm which achieves local optimality with fewer samples compared to the direct application of Algorithm 4.2.

**Algorithm Description:** Due to space constraints, we refer the reader to Appendix D.1 for the complete description of Algorithm D.1. Here, we briefly outline the core idea of the algorithm and illustrate the key sub-routine, emphasizing the novel contributions of our approach. Our approach entails iteratively implementing the following procedure for each agent $m \in [n]$: we maintain the policies of agents $[n]\backslash\{m\}$ (referred to as $\pi_{-m}$) fixed and determine the policy $\pi_m^* = \arg\max_{\mu_m} V^{\mu_m,\pi_{-m}}$. If $V^{\pi_m^*,\pi_{-m}} - V^{\pi_m,\pi_{-m}} < \epsilon$, we terminate the entire algorithm and output the policy $\otimes_{m=1}^n \pi_m$. Otherwise, we replace the policy of agent $m$ with $\pi_m^*$ and continue the procedure. Consequently, we are tasked with developing a sample-efficient algorithm to obtain $\pi_m^* = \arg\max_{\mu_m} V^{\mu_m,\pi_{-m}}$. We present Algorithm D.1, which fulfills this task. The algorithm consists of two main steps, which we now explain in detail. For clarity in the presentation, we denote $\overline{\mathsf{ch}}(i) = \{l_1, \ldots, l_r\}$ with $l_r = i$.

---

**Algorithm 2** OMLE for Achieving Local Optimal Under Factored Model

---

1: **Initialize**: $\mathcal{B}_m^1 = \{\hat{\theta}_m \in \Theta_m : \min_h \sigma_S(\mathbb{O}_m(\hat{\theta}_m)) \geq \alpha\}$, $\mathcal{D}_m = \{\}$ for all $m \in \overline{\mathsf{ch}}(i)$,
   $\tilde{\mathcal{B}}^1 = \{\theta_{m \in \mathsf{nch}(i)} : \min_h \sigma_S(\mathbb{O}_m(\theta_m)) \geq \alpha, \forall m \notin \overline{\mathsf{ch}}(i)\}$, $\tilde{\mathcal{D}} = \{\}$, central agent $i$, policy of
   other agent $\pi_{-i}$.
2: **for** $k = 1 \dots T$ **do**
3:    Follow $\pi_{\mathsf{nch}(i)}$ to collect trajectories $\tau_{\overline{\mathsf{ch}}(i)}^k = \{o_{1,m}^k, \dots, a_{H,m}^k\}_{m \in \mathsf{nch}(i)}$.
4:    Add $\tau_{\mathsf{nch}(i)}^k$ into $\tilde{\mathcal{D}}$ and update confidence interval with eq. (4).

5: **for** $k = 1 \dots T$ **do**
6:    compute $(\boldsymbol{\theta}^k, \pi_i^k) = \arg\max_{\{\hat{\theta}_m \in \mathcal{B}_m^k\}_{m \in \mathsf{Chl}(i)}, \tilde{\theta}_i \in \tilde{\Theta}_i, \mu_i} V^{\mu_i, \pi_{-i}}(\hat{\boldsymbol{\theta}})$
7:    **for** $m = 1, 2, \dots, r$ **do**
8:       **for** $h = 1, \dots, H$ **do**
9:          Agent $l \in \mathsf{nch}(i)$ take action $a_{h,l}^T$.
10:         Select an action $a_{h,l_j}^k \sim \pi_{h,l_j}(\cdot \mid \tau_{h-1,l_j}, o_{h,l_j})$ for all $j \in [r-1]$.
11:         Select an action $a_{h,i}^k \sim \pi_{h,i}^k(\cdot \mid \tau_{h-1,i}, o_{h,i})$.
12:         For agent $l_j$ with $j \in [m]$, collect observation $o_{h+1,l_j}^k$ from the environment.
13:         For $j \in [r] \setminus [m]$, sample dummy state $s_{h+1,l_j} \sim \mathbb{T}_{h,l_j}^k(\cdot \mid s_{h,l_j}, a_{h,\overline{\mathsf{pa}}(l_j)})$.
14:         Collect observation $o_{h+1,l_j}^k \sim \mathbb{O}_{h+1,l_j}^k(\cdot \mid s_{h+1,l_j})$ for $j \in [r] \setminus [m]$.
15:      If $m \neq r$, add $(\pi_m, \tau_{\overline{\mathsf{pa}}(m) \cap \overline{\mathsf{ch}}(i)}^k, \tau_{\overline{\mathsf{pa}}(m) \setminus \overline{\mathsf{ch}}(i)}^T)$ to $\mathcal{D}_m$.
16:      Otherwise, add $(\pi_i^k, \tau_{\overline{\mathsf{pa}}(i) \cap \overline{\mathsf{ch}}(i)}^k, \tau_{\overline{\mathsf{pa}}(i) \setminus \overline{\mathsf{ch}}(i)}^T)$ to $\mathcal{D}_i$.
17:   Update confidence interval with eq. (5) for all $j \in \overline{\mathsf{ch}}(i)$.
18: **Output** $\hat{\pi}$ as uniform mixture of the policies $\pi_i^1, \pi_i^2, \dots, \pi_i^K$.

---

- *Estimate model parameters $\theta_{\mathsf{nch}(i)}$ (Lines 2-4)*: According to the factored structure, agent $m \notin \overline{\mathsf{ch}}(i)$ is not influenced by the actions of agents $m \in \overline{\mathsf{ch}}(i)$, and the policies of agents $m \notin \overline{\mathsf{ch}}(i)$ are predetermined. Based on this observation, the key idea of our algorithm is to estimate the model parameters $\theta_{\overline{\mathsf{ch}}(i)}$ and $\theta_{\mathsf{nch}(i)}$ separately in two loops over $T$ episodes. For the estimation of the model parameter $\theta_{\mathsf{nch}(i)}$, we construct $\tilde{\mathcal{B}}$ as the set of model parameters for agents $m \notin \overline{\mathsf{ch}}(i)$. We then iteratively follow this process for $T$ episodes: In the $k$-th episode, by executing policy $\pi_{\mathsf{nch}(i)}$, we collect trajectories $\tau_{\mathsf{nch}(i)}^k$. Subsequently, we update the confidence interval with:

$$\tilde{\mathcal{B}}^{k+1} = \Big\{\theta_{\mathsf{nch}(i)} \in \tilde{\mathcal{B}}^1 : \sum_{\tau_{\mathsf{nch}(i)} \in \tilde{\mathcal{D}}} \log f(\hat{\theta}_{\mathsf{nch}(i)}, \tau_{\mathsf{nch}(i)}) \geq \max_{\hat{\theta}_{\mathsf{nch}(i)}'} \sum_{\tau_{\mathsf{nch}(i)} \in \tilde{\mathcal{D}}} f(\hat{\theta}_{\mathsf{nch}(i)}', \tau_{\mathsf{nch}(i)}) - \tilde{\beta}\Big\}, \tag{4}$$

  where $\forall \theta_{\mathsf{nch}(i)}$, we define $f(\theta_{\mathsf{nch}(i)}, \tau_{\mathsf{nch}(i)}) = \prod_{m \notin \overline{\mathsf{ch}}(i)} \mathbb{P}_{\theta_m}^{\pi_m}(\tau_m \mid \tau_{\overline{\mathsf{pa}}(m)})$.

- *Estimate model parameters $\theta_{\overline{\mathsf{ch}}(i)}$ and update policy (Lines 9-15)*: We proceed with another $T$ episodes to estimate the remaining parameters and find the optimal policy. Similar to Algorithm 4.2, we construct separate confidence intervals to estimate each model parameter for the individual agents. At the beginning of the $k$-th episode, we select the parameter $\hat{\theta}_m^k$ from the confidence interval $\mathcal{B}_m^k$, the parameter $\theta_{\mathsf{nch}(i)}$ from $\tilde{B}^k$, and the policy for agent $i$ that optimizes the total value function. Next, for each $m \in [r]$, we collect a joint trajectory using a similar sampling procedure as in Algorithm 4.2. Specifically, at step $h$, agent $l \in \mathsf{nch}(i)$ takes action $a_{h,l}^T$, while agent $l_j$ (with $j \in [m]$) samples actions and observations from the true environment, and the remaining agents sample from the model corresponding to $\theta^k$. Eventually, we add a sample to each individual sample set and update the confidence intervals with:

$$\mathcal{B}_j^{k+1} = \Big\{\hat{\theta}_j \in \mathcal{B}_j^1 : \sum_{(\pi_j, \tau_{\overline{\mathsf{pa}}(j)}) \in \mathcal{D}_j} \log \mathbb{P}_{\hat{\theta}_j}^{\pi_j}(\tau_j \mid \tau_{\mathsf{pa}(j)}) \geq \max_{\theta_j' \in \Theta_j} \sum_{(\pi_j, \tau_{\overline{\mathsf{pa}}(j)}) \in \mathcal{D}_j} \log \mathbb{P}_{\theta_j'}^{\pi_j}(\tau_j \mid \tau_{\mathsf{pa}(j)}) - \beta_j\Big\}. \tag{5}$$

**Theorem 4.3.** *We define bonus parameter in eq.* (15). *Then, with probability at least* $1 - \delta$, *Algorithm D.1 terminates within* $4H/\epsilon$ *steps of the while loop, and outputs an* $\epsilon-$*approximate local optimal*

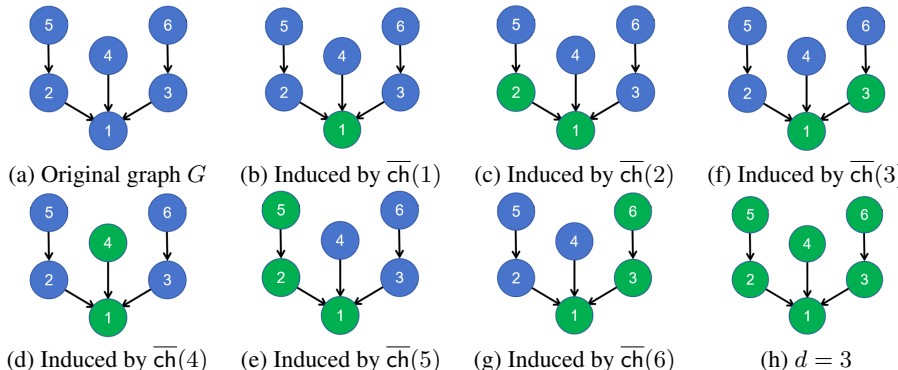

(a) Original graph $G$    (b) Induced by $\overline{\mathsf{ch}}(1)$    (c) Induced by $\overline{\mathsf{ch}}(2)$    (f) Induced by $\overline{\mathsf{ch}}(3)$

(d) Induced by $\overline{\mathsf{ch}}(4)$    (e) Induced by $\overline{\mathsf{ch}}(5)$    (g) Induced by $\overline{\mathsf{ch}}(6)$    (h) $d = 3$

Figure 1: An example illustrating the sample complexity of Algorithm D.1. For each $m \in \mathcal{V}$, we highlight the subgraph induced by $\overline{\mathsf{ch}}(m)$ in green. It can be observed that $\max_{m \in [n]} d_m = 1$.

*policy. The total number of episodes is at most*

$$K = \tilde{\mathcal{O}}\left( \sum_{m=1}^{n} S^4 O^2 A^{2d_m+2}(S^2 A^{d+1} + SO) \cdot \mathrm{poly}(H)/(\alpha^4 \epsilon^3) \right).$$

*where we use $d_m$ to denote the maximum in-degree of the sub-graph induced by $\overline{\mathsf{ch}}(m)$, and we naturally have $d_m \leq d$.*

**Technical Insights:** The novelty of Algorithm D.1 compared to Algorithm 4.2 lies in its careful utilization of the structural properties. Specifically, since only the policy of the central agent (denoted as $i$) varies across episodes, and the trajectory probability of agents $m \notin \overline{\mathsf{ch}}(i)$ remains unaffected by this variation, we can pre-estimate the model parameters $\theta_{\mathsf{nch}(i)}$. Finally, we proceed to estimate $\theta_{\overline{\mathsf{ch}}(i)}$, where we adopt a similar sampling method as in Algorithm 4.2 and achieve a sample complexity that is exponential only in $d_m$, rather than in $d$. If we were to directly apply Algorithm 4.2 by adjusting the parameter and policy selection to $(\theta_1^k, \theta_2^k, \ldots, \theta_n^k, \pi_m^k) = \arg\max_{\hat{\theta}_1 \in \mathcal{B}_1^k, \hat{\theta}_2 \in \mathcal{B}_2^k, \ldots, \hat{\theta}_n \in \mathcal{B}_n^k, \pi_m} V^{\pi_m, \pi_{-m}}(\hat{\boldsymbol{\theta}})$, while keeping the other proceduresd unchanged, the total number of required episodes would be $K = \tilde{\mathcal{O}}(S^4 A^{2d+2}(S^2 A^{d+1} + SO) \cdot \mathrm{poly}(H)/(\alpha^4 \epsilon^3))$. Thus, Algorithm D.1 significantly reduces the sample complexity needed to achieve an $\epsilon$-approximate locally optimal policy. To illustrate this improvement, we provide an example in Figure 1. In this example, Algorithm D.1 requires a sample complexity of $\mathcal{O}(A^8)$, whereas applying Algorithm 4.2 directly would result in a sample complexity of $\mathcal{O}(A^{12})$.

### 4.4 APPLICATION: POMDP WITH KNAPSACK CONSTRAINTS

As a minor extension, we demonstrate the applicability of our approach to a specific problem domain: POMDP with knapsack constraints, akin to the example in (Chen et al., 2020). We consider a POMDP with a budget $\mathbf{M} \in \mathbb{R}^d$. At each time step $h$, the agent incurs a cost vector $\mathbf{C}_h$, and the total budget updates to $\mathbf{M}_{h+1} = \mathbf{M}_h - \mathbf{C}_h$. We model the transition of each budget component $i$ as $\mathbf{M}_{h+1,i} \sim \mathbb{T}_{h,\mathbf{M}}(\cdot \mid \mathbf{M}_{h,i}, o_h, a_h)$. The episode terminates when any budget component reaches 0.

We formulate this problem as a factored DEC-POMDP with $d+1$ agents, treating the budgets as observations of $d$ dummy agents. Consequently, Algorithm 4.2 can be directly applied. However, since the budgets are directly observed, there are still opportunities for improvement. We estimate the transition $\mathbb{T}_{h,\mathbf{B}}$ using a confidence interval approach akin to UCB-VI (Azar et al., 2017), and we can achieve a sharper bound. Due to the space limit, we defer the complete discussion to Appendix F.

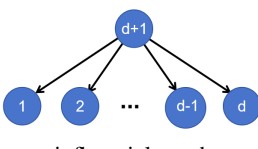

influential graph

## 5 LEARNING DEC-POMDP WITH MEMORYLESS POLICY

In this section, we focus on the setting where agents adopt memoryless policies. Namely, each agent select her action base solely on her current observation. We define the policy class as $\{\{\otimes_{m=1}^{n} \pi_{h,m}\}_{h \in [H]} \mid \pi_{h,m} : \mathcal{O}_m \to \Delta(\mathcal{A}_m)\}$. Notably, our results can be readily extended to settings where agents consider observations and actions from the preceding $L$ steps. In such cases, the policy class broadens to $\{\{\otimes_{m=1}^{n} \pi_{h,m}\}_{h \in [H]} \mid \pi_{h,m} : (\mathcal{O}_m \times \mathcal{A}_m)^{\min(h,L)-1} \times \mathcal{O}_m \to \Delta(\mathcal{A}_m)\}$.

## 5.1 Achieving Local Optimality with Memoryless Policy

We utilize a framework similar to that described in Section 4.3. Specifically, we iteratively update the policy of the $m^{th}$ agent using $\pi_m^* = \arg\max_{\mu_m} V^{\mu_m, \pi_{-m}}$ and terminate the procedure when further updates no longer produce a significant increase in the total value function. Our remaining task is to develop a sample-efficient method to obtain $\pi_m^* = \arg\max_{\mu_m} V^{\mu_m, \pi_{-m}}$ given $\pi_{-m}$, for all $m \in [n]$. We present Algorithm 5.1, which addresses this task. Due to space constraints, a detailed description of the complete algorithm is provided in Appendix E.1 (Algorithm E.1).

**Algorithm Description:** For each agent $m \in [n]$, the parameter set $\theta_m$ represents the model parameters of the joint probability distribution of trajectories for the $m$-th agent and the $i$-th agent. We denote $\mathbb{P}_{\theta_i}^{\pi_i}(\tau_i)$ as the probability of the trajectory for the $i$-th agent and $\mathbb{P}_{\theta_m}^{\pi_i, \pi_m}(\tau_i, \tau_m)$ as the joint probability of trajectories for the $i$-th and $m$-th agents, given the underlying DEC-POMDP with parameters $\boldsymbol{\theta} = (\theta_1, \ldots, \theta_n)$. The formal definition is provided in Appendix E.1 (Equation equation 18). With these definitions, we now proceed to explain Algorithm 5.1 in detail.

- *Update policy and parameters (Lines 3-4)*: We denote the value functions $V_i^{\pi_i, \pi_{-i}}(\theta_i)$ and $V_m^{\pi_i, \pi_{-i}}(\theta_m)$ for all $m \in [n] \setminus \{i\}$ as:

$$V_i^{\pi_i, \pi_{-i}}(\theta_i) = \sum_{\tau_i} \mathbb{P}_{\theta_i}^{\pi_i}(\tau_i)\big(\sum_{h=1}^H r_{h,i}(o_{h,i})\big),$$
$$V_m^{\pi_i, \pi_{-i}}(\theta_m) = \sum_{\tau_m, \tau_i} \mathbb{P}_{\theta_m}^{\pi_i, \pi_m}(\tau_m, \tau_i)\big(\sum_{h=1}^H r_{h,m}(o_{h,m})\big).$$

The observation probabilities $\{\mathbb{O}_{h,m}\}_{h=1}^H$ and the transition probabilities $\{\mathbb{T}_{h,m}\}_{h=1}^H$ corresponding to the real model $\theta_m = \theta_m^*$ are defined as per Equations equation 19 and equation 20 in Appendix E.1. In this context, the value function can be decomposed as $V^{\boldsymbol{\pi}} = \sum_{m=1}^n V_m^{\pi_i, \pi_{-i}}(\theta_m^*)$. Thus, we decompose the value function into $n$ distinct terms, each depending solely on the parameter $\theta_m$. For each $m \in [n]$, we select $\theta_m^k \in \mathcal{B}_m^k$ as the optimal parameter that maximizes $V_m^{\pi_i, \pi_{-i}}(\theta_m)$. We then determine the policy $\pi_i^k$ as the optimal policy that maximizes the total value function. Subsequently, we use the policy $\boldsymbol{\pi}^k = (\pi_i^k, \pi_{-i})$ to collect a trajectory $\boldsymbol{\tau}^k = (\mathbf{o}_1^k, \mathbf{a}_1^k, \ldots, \mathbf{o}_H^k, \mathbf{a}_H^k)$.
- *Construct Confidence Intervals (Lines 5-6)*: We construct $n$ different confidence intervals to estimate the $n$ model parameters separately. For each agent $m \in [n] \setminus \{i\}$, we add the newly collected policy-trajectory pair $(\pi_i^k, \tau_i^k, \tau_m^k)$ to the dataset $\mathcal{D}_m$. Similarly, for agent $i$, we add the policy-trajectory pair $(\pi_i^k, \tau_i^k)$ to the dataset $\mathcal{D}_i$. Subsequently, we update each of the $n$ confidence intervals separately according to the following equations for agent $i$ and $m \in [n] \setminus \{i\}$:

$$\mathcal{B}_i^{k+1} = \Big\{\hat{\theta}_i \in \mathcal{B}_i^1 : \sum_{(\pi_i, \tau_i) \in \mathcal{D}_i} \log \mathbb{P}_{\hat{\theta}_i}^{\pi_i}(\tau_i) \geq \max_{\theta_i' \in \Theta_i} \sum_{(\pi_i, \tau_i) \in \mathcal{D}_i} \log \mathbb{P}_{\theta_i'}^{\pi_i}(\tau_i) - \beta_i\Big\}$$

$$\mathcal{B}_m^{k+1} = \Big\{\hat{\theta}_m \in \mathcal{B}_m^1 : \sum_{(\pi_i, \tau_i, \tau_m) \in \mathcal{D}_m} \log \mathbb{P}_{\hat{\theta}_m}^{\pi_i, \pi_m}(\tau_i, \tau_m) \geq \max_{\theta_m' \in \Theta_m} \sum_{(\pi_i, \tau_i, \tau_m) \in \mathcal{D}_m} \log \mathbb{P}_{\theta_m'}^{\pi_i, \pi_m}(\tau_i, \tau_m) - \beta_m\Big\}$$

$$(6)$$

**Technical Challenge and Novelty:** To showcase our novel approach, we highlight the challenges in developing a sample-efficient algorithm to find $\pi_m^* = \arg\max_{\mu_m} V^{\mu_m, \pi_{-m}}$ with given $\pi_{-m}$, along with our solutions to these challenges:

1. In DEC-MDPs, given $\pi_{-m}$, the model reduces to a single-agent problem with action space $\mathcal{A}_m$. However, this reduction does not apply to DEC-POMDPs, precluding the use of single-agent algorithms for sample-efficient guarantees as in MDP setting.
2. Another challenge arises from the exponential growth of the joint action and observation space with the number of agents, resulting in a model dimension that scales as $\mathcal{O}(A^n O^n)$. Consequently, constructing a single confidence interval to estimate the model parameters leads to a sample complexity of $\mathcal{O}(A^n O^n)$.

We overcome the technical challenges by assigning a parameter for the trajectory probability of each agent, subsequently estimating and updating these parameters with separate confidence intervals. Since the dimension of parameter $\theta_m$ $(m \in [n])$ is at most $H(S^2 A^2 O^2 + SO^2) + S$, we achieve an $\epsilon$-approximate local optimum with a sample complexity that avoids exponential scaling in $n$.

**Theorem 5.1.** *Let the central agent for Algorithm 5.1 be agent $i$. We define the bonus parameter as eq. (17). Then, with a probability of at least $1 - \delta$, Algorithm E.1 terminates within $4H/\epsilon$ steps of the while loop and outputs an $\epsilon$-approximate local optimal policy. The total number of episodes*

---

**Algorithm 3** OMLE for memoryless policy

---

1: **Input**: Central agent $i$, and the policy for agent $[n] \setminus \{i\}$, $\pi_1, \pi_2, \ldots, \pi_{i-1}, \pi_{i+1}, \ldots, \pi_n$.
2: **Initialize**: $\mathcal{B}_i^1 = \{\hat{\theta}_i \in \Theta_i : \min_h \sigma_S(\mathbb{O}_i(\hat{\theta}_i) \geq \alpha)\}$, $\mathcal{B}_m^1 = \{\hat{\theta}_m \in \Theta_m : \min_h \sigma_S(\mathbb{O}_m(\hat{\theta}_m) \geq \alpha \setminus \sqrt{O})\}$ for all $m \in [n] \setminus \{i\}$. Set $\mathcal{D}_m = \{\}$, for all agents $m \in [n]$.
3: **for** $k = 1 \ldots T$ **do**
4:     Compute $(\theta_1^k, \theta_2^k, \ldots, \theta_n^k, \pi_i^k) = \arg\max_{\hat{\theta}_1 \in \mathcal{B}_1^k, \hat{\theta}_2 \in \mathcal{B}_2^k, \ldots, \hat{\theta}_n \in \mathcal{B}_n^k, \pi_i} \sum_{m=1}^n V_m^{\pi_i, \pi_{-i}}(\hat{\theta}_m)$.
5:     Follow $\pi^k$ to collect a trajectory $\boldsymbol{\tau}^k = (\mathbf{o}_1^k, \mathbf{a}_1^k, \ldots, \mathbf{o}_H^k, \mathbf{a}_H^k)$.
6:     Add $(\pi_i^k, \tau_i^k, \tau_m^k)$ into $\mathcal{D}_m$ for $m \in [n] \setminus \{i\}$ and add $(\pi_i^k, \tau_i^k)$ into $\mathcal{D}_i$.
7:     Update $\mathcal{B}_i^{k+1}$ and $\mathcal{B}_m^{k+1}$ for all $m \in [n] \setminus \{i\}$ with eq. (6).
8: **Output** $\hat{\pi}$, which is selected uniformly from the policies $\pi_i^1, \pi_i^2, \ldots, \pi_i^T$.

---

*played by Algorithm E.1 is at most*

$$K = \tilde{\mathcal{O}}\big(S^4 O^4 A^4 (S^2 A^2 O^2 + SO^2) \cdot \text{poly}(H)/(\alpha^4 \epsilon^3)\big).$$

**Remark 5.1.** *In a commonly studied model (where agents adopt memoryless policies), we derive an algorithm capable of achieving an $\epsilon$-approximate local optimal policy for DEC-POMDPs. Importantly, the sample complexity of this algorithm does not scale exponentially with $n$.*

*Moreover, since DEC-POMDPs can be seen as a special case of a partially observable version of a Markov potential game, our framework extends to the analysis of achieving Nash equilibrium within this partially observable version.*

## 5.2 HARDNESS RESULT FOR ACHIEVING GLOBAL OPTIMALITY

In addition to local optimality, we now explore the attainment of global optimality with memoryless policies. However, the following theorem reveals that, without additional assumptions on the model, deriving an algorithm to achieve global optimality with regret not exponential in $n$ is unattainable.

**Theorem 5.2.** *For any randomized or deterministic algorithms, there exists an instance of DEC-MDP with horizon $H = 2$ wherein the regret scales at least as $\mathcal{O}(\sqrt{A^n T})$. This result underscores the limitation of achieving sample efficiency in algorithms for DEC-POMDP without imposing assumptions on the transition model, either when agents adopt memoryless policies.*

## 6 CONCLUSION AND DISCUSSION

**Conclusion and Summary**: This work introduces a sample-efficient algorithm and provides theoretical guarantees for DEC-POMDPs. Theorem 5.2 highlights the challenges associated with developing sample-efficient algorithms for DEC-POMDPs without making any assumptions about the model. Consequently, our focus shifts towards identifying such algorithms under specific conditions rather than for the general model. Initially, we present a sample-efficient algorithm for a commonly studied scenario where agents utilize memoryless policies. Furthermore, inspired by empirical methods that leverage value decomposition to address exponential complexity, we propose a factored structural model as a sufficient condition for value decomposition and derive a sample-efficient algorithm based on this assumption. This analysis provides a theoretical foundation for the empirical strategies currently employed.

**Open Directions**: One open question is whether it is possible to derive a sample-efficient algorithm that achieves local optimality without imposing any assumptions on the model. When Algorithm 5.1 is applied to a full-memory setting, the sample complexity upper bound scales as $\mathcal{O}(A^H)$. In contrast, applying the vanilla OMLE algorithm from Liu et al. (2022a) results in a sample complexity of $\mathcal{O}(A^n)$. Therefore, it remains an open problem to determine whether a lower bound on sample complexity can scale as $\mathcal{O}(A^{\min\{H,n\}})$, or if it is possible to overcome the multi-agent curse without imposing assumptions on the model. Another avenue for future research is to explore whether additional reasonable assumptions about the model could facilitate the development of sample-efficient algorithms. We leave these directions for future investigation.

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

# Appendix

## Table of Contents

# A ADDITIONAL RELATED WORK

**Learning POMDPs** Learning partially observable Markov decision processes (POMDPs) presents significant challenges due to the lack of the Markov property in observations and the dependence of policies on the full observation history. This complexity is underscored by lower bounds, such as those established by Mossel and Roch (2005) and Krishnamurthy et al. (2016), which show exponential complexity in the horizon for learning near-optimal policies in POMDPs. Given the difficulty of learning POMDPs in the general case, recent research has explored learning under various structural conditions. Some works, like Jin et al. (2020) and Liu et al. (2022a), have investigated weakly revealing conditions, while others, such as Cai et al. (2022) and Wang et al. (2022), have focused on low-rank POMDPs. Efroni et al. (2022) and Zhang et al. (2023) have delved into learning under decodable conditions, while Uehara et al. (2022) and Uehara et al. (2023) have proposed algorithms for learning with memoryless policies and deterministic transition models, respectively. Chen et al. (2022) have introduced the B-stability condition as a comprehensive framework that encompasses previous structural conditions. In our work, we demonstrate our results under a weakly revealing condition akin to that of Jin et al. (2020) and Liu et al. (2022a). However, it's important to note that our framework can be extended to incorporate other conditions proposed in previous works, such as the B-stability condition introduced by Chen et al. (2022). This flexibility underscores the applicability and generality of our approach within the broader landscape of learning POMDPs.

**Learning POMGs** Liu et al. (2022b) present the OMLE algorithm for finding approximate Nash equilibria, correlated equilibria, as well as coarse correlated equilibrium of weakly revealing POMGs in a polynomial number of samples, particularly when the number of agents is small. On a related note, Liu and Zhang (2023) develop a partially observable multi-agent reinforcement learning (MARL) algorithm that is both statistically and computationally quasi-efficient, incorporating information sharing under the general framework of partially observable stochastic games.

**Learning MDPs and POMDPs with Specific Structures** We consider a factorized structure model in this work. In the context of factored MDPs, Osband and Van Roy (2014) first proposed the factored MDP model and introduced PSRL and UCRL-style algorithms with near-optimal Bayesian and frequentist regret bounds. Xu and Tewari (2020) extended the results of Osband and Van Roy (2014) to the infinite horizon setting. Tian et al. (2020) applied the UCBVI algorithm (Azar et al., 2017) to the factored MDP framework, while Chen et al. (2020) further refined the approach by applying the UCB-VI algorithm and developing the FMDP-BF algorithm, which achieves a sharper bound compared to Tian et al. (2020). Additionally, Chen et al. (2020) introduced reinforcement learning with knapsack constraints as an example of factored MDPs. Diuk et al. (2009) proposes an algorithmic framework based on the KWIK principle for learning probabilistic concepts, and applies this framework to reinforcement learning in factored models. The authors provide empirical insights that suggest more efficient algorithms can be derived when restricted to factored structure models. Strehl et al. (2007) addresses the reinforcement learning problem in factored MDPs and proposes an efficient algorithm leveraging dynamic Bayesian networks (DBNs). Chakraborty and Stone (2011) studies factored-state MDPs and aims to develop an algorithm that guarantees a return close to the optimal in factored MDPs. Since factored-state MDPs are a special case of Markov chains, they utilize the properties of ergodic stochastic processes to analyze factored MDPs.

In terms of POMDPs, several prior studies have explored POMDPs with specific factored structures. For example, Katt et al. (2018) introduced the Factored Bayes-Adaptive POMDP model, along with a method to learn both the factorization and the model parameters simultaneously. Guestrin et al. (2001) demonstrated that, for factored POMDPs, the value function can be represented as a linear combination of basis functions, enabling the derivation of an efficient algorithm by leveraging the decomposition of the value function. Similar to our setting, several previous studies have examined factor structures in DEC-POMDPs. For instance, Oliehoek et al. (2008b) analyzed general factored DEC-POMDPs, focusing on the model's dependencies over space and time, and formulated decomposable value functions.

The aforementioned works primarily focus on analyzing the state structure of POMDPs. Beyond considering POMDPs with specific state structures, Altabaa and Yang (2024) investigated the role of information structure, which describes how events in the system occurring at different points in time influence each other. Altabaa and Yang (2024) also provided an upper bound on the sample

complexity for learning a general sequential decision-making problem with a directed acyclic graph (DAG) information structure.

**Learning with Memoryless Policies**    Since learning POMDPs is known to be PSPACE-complete, many works focus on developing algorithms to learn optimal memoryless policies, which can be viewed as a special case of general POMDPs. Kara and Yuksel (2022) studied learning optimal memoryless policies for POMDPs by approximating the belief model through discretizing the belief space. Kara and Yüksel (2023) provided convergence analysis for a Q-learning algorithm tailored to POMDPs with memoryless policies.

**Learning Multi-agent System:**    In multi-agent reinforcement learning (MARL), the action space grows exponentially with the number of agents, making it crucial to derive algorithms whose sample complexity is not exponential in the number of agents—a challenge commonly referred to as breaking the curse of multi-agency. Daskalakis et al. (2023); Jin et al. (2021) derive sample-efficient algorithms with non-exponential sample complexity, while Wang et al. (2023); Cui et al. (2023) further generalize this approach to settings with linear function approximation. Another method to address the exponential growth of the action space is the mean-field approach, which assumes that each agent's decision is influenced by the mean field (i.e., the average behavior of other agents) rather than by the individual actions of each agent. Previous work utilizing the mean-field method to tackle exponential growth in multi-agent RL includes Yang et al. (2018); Pasztor et al. (2021); Qiu et al. (2022). Gu et al. (2021) demonstrates that if all agents are homogeneous and exchangeable, mean-field control can provide a good approximation to an $N$-agent problem. Similarly, Mondal et al. (2022) provides a comparable approximation for a $K$-class of heterogeneous agents. In contrast to the mean-field method, we adopt a similar approach of optimizing the performance of a single agent while considering the overall effect of others. This allows the optimization problem to be approximated as a single-agent problem, thereby mitigating the exponential growth of the action space in the environment.

## B    PROOF OF THEOREM 3.1

**Theorem B.1.** *For any randomized or deterministic algorithms, there exists an instance of DEC-MDP wherein the regret scales at least as* $\Omega(\sqrt{A^n T})$.

*Proof.* The proof for Theorem 3.1 proceeds straightforwardly. We consider a two-step DEC-MDP, commencing from an initial state $s_1$. For all $s \in \mathcal{S}$, we assume the reward function satisfies $r_{h,1}(s) = r_{h,2}(s) = \cdots = r_{h,n}(s)$ for all $h \in [2]$. Consequently, the entire DEC-MDP reduces to a multi-armed bandit problem. By leveraging a classic result on the lower bound of regret for the multi-armed bandit problem (Mannor and Tsitsiklis, 2004), it follows that for any randomized or deterministic algorithm, there exists an instance of the multi-arm bandit problem such that the regret is at least $\mathcal{O}(\sqrt{\tilde{A}T})$, where $\tilde{A}$ denotes the number of arms. Consequently, for any randomized or deterministic algorithm, there exists an instance of DEC-MDP such that the regret is at least $\mathcal{O}(\sqrt{A^n T})$. □

## C    SUPPLEMENTARY DETAILS FOR SECTION 4.2

### C.1    PROOF FOR THEOREM 4.1

In this section, we present the proof of Theorem 4.1. To ensure clarity, we begin by defining several notations that will be useful throughout the proof.

**Definition C.1.** *For all* $(m, i) \in [n] \times [T]$, $\theta_m \in \Theta_m$, *and any policy* $\pi_m$ *of agent* $m$, *we denote* $f_m(\theta_m, \pi_m)$ *as* $f_m(\theta_m, \pi_m) = \mathbb{P}_{\theta_m}^{\pi_m}(\tau_m \mid \{\tau_r\}_{r \in \mathsf{pa}(m)})$. *Additionally, we use* $\tilde{f}_m^i(\theta_m, \pi_m)$ *to denote* $\tilde{f}_m^i(\theta_m, \pi_m) = \mathbb{P}_{\theta_m, m}^{\pi_m}\left(\tau_m^i \mid \{\tau_r^i\}_{r \in \mathsf{pa}(m)}\right)$.

**Lemma C.1.** *For all $(\theta_m, t) \in \Theta_m \times [T]$ and agent $m \in [n]$, the following inequality holds with probability at least $1 - \delta$:*

$$\sum_{i=1}^{t} \log \left( \tilde{f}_m^i(\theta_m, \pi_m) / \tilde{f}_m^i(\theta_m^\star, \pi_m) \right) \leq \beta_m,$$

*where we define bonus term $\beta_m = c(H^2(S^2 A^{|\mathsf{pa}(m)|+1} + SO) \log(TSAOH) + \log(Tn/\delta))$ for some absolute constant $c$.*

*Proof.* Initially, we can view $\Theta_m$ as a subset of $\mathbb{R}^{d_m}$ with $d_m = H(S^2 A^{|\mathsf{pa}(m)|} + SO) + S$. We denote $\bar{\theta}_m$ as the optimistic $\epsilon-$discretelization of $\theta_m$, so that $\bar{\theta}_{m,i} = \lceil \theta_{m,i}/\epsilon \rceil \times \epsilon$ for all coordinates $i$. Selecting $\epsilon \leq 1/(c(S + O + A)HTO^H A^H)$, we obtain the following relationship:

$$f_m(\bar{\theta}_m, \pi_m) \geq f_m(\theta_m, \pi_m), \quad \left| f_m(\bar{\theta}_m, \pi_m) - f_m(\theta_m, \pi_m) \right| \leq 1/(TO^H A^H).$$

The inequalities holds for all trajectories $\tau_{H,m} \in (\mathcal{O}_m \times \mathcal{A}_m)^H$. We use $\bar{\Theta}_m$ to represent the collections of all such $\bar{\theta}_m$, then, the log-cardinality of $\bar{\Theta}_m$ is bounded by

$$\log \left| \bar{\Theta}_m \right| \leq \mathcal{O} \left( H^2 \left( S^2 A^{|\mathsf{pa}(m)|+1} + SO \right) \log(TSAOH) \right).$$

In the following step, we aim to apply Markov inequality to bound the following expectation: $\mathbb{E}[\exp(\sum_{i=1}^{t} \log(\tilde{f}_m^i(\bar{\theta}_m, \pi_m)/\tilde{f}_m^i(\theta_m^\star, \pi_m)))]$. We denote $\mathbb{E}_t[\cdot] = \mathbb{E}\left[ \cdot \mid \{\pi^i, \tau^i\}_{i=1}^{t-1} \cup \{\pi^t\} \right]$. We then have

$$\mathbb{E}\left[ \exp \left( \sum_{i=1}^{t} \log \left( \tilde{f}_m^i \left( \bar{\theta}_m, \pi_m^i \right) / \tilde{f}_m^i \left( \theta_m^\star, \pi_m^i \right) \right) \right) \right]$$

$$= \mathbb{E}\left[ \exp \left( \sum_{i=1}^{t-1} \log \left( \tilde{f}_m^i \left( \bar{\theta}_m, \pi_m^i \right) / \tilde{f}_m^i \left( \theta_m^\star, \pi_m^i \right) \right) \right) \cdot \mathbb{E}_t \left[ \exp \left( \log \left( \tilde{f}_m^t(\bar{\theta}_m, \pi_m^t)/\tilde{f}_m^t(\theta_m^\star, \pi_m^t) \right) \right) \right] \right]$$

$$= \mathbb{E}\left[ \exp \left( \sum_{i=1}^{t-1} \log \left( \tilde{f}_m^i \left( \bar{\theta}_m, \pi_m^i \right) / \tilde{f}_m^i \left( \theta_m^\star, \pi_m^i \right) \right) \right) \cdot \mathbb{E}_t \left( \tilde{f}_m^t(\bar{\theta}_m, \pi_m^t)/\tilde{f}_m^t(\theta_m^\star, \pi_m^t) \right) \right] \quad (7)$$

According to the Definition C.1, we further have

$$\mathbb{E}_t \left( \frac{\tilde{f}_m^t(\bar{\theta}_m, \pi_m^t)}{\tilde{f}_m^t(\theta_m^\star, \pi_m^t)} \right) = \mathbb{E}_t \left[ \sum_{\boldsymbol{\tau}_H} f_m \left( \bar{\theta}_m, \pi_m^t \right) \left( \prod_{j=1}^{m-1} f_j \left( \theta_j^\star, \pi_j^t \right) \right) \left( \prod_{j=m+1}^{n} f_j \left( \theta_j^t, \pi_j^t \right) \right) \right] \leq \left( 1 + \frac{1}{T} \right). \quad (8)$$

We insert eq. (8) back into eq. (7), and we can obtain that

$$\mathbb{E}\left[ \exp \left( \sum_{i=1}^{t} \log \left( \tilde{f}_m^i \left( \bar{\theta}_m, \pi_m \right) / \tilde{f}_m^i \left( \theta_m^\star, \pi_m \right) \right) \right) \right] \leq e.$$

We then use Markov inequality and take a union bound for all $(\bar{\theta}_m, t) \in \bar{\Theta}_m \times [T]$ and $m \in [n]$, and we can conclude that the following event holds with probability at least $1 - \delta$ for all $m \in [n]$:

$$\max_{(\bar{\theta}_m, t) \in \bar{\Theta}_m \times [T]} \sum_{i=1}^{t} \log \left( \tilde{f}_m^i(\bar{\theta}_m, \pi_m) / \tilde{f}_m^i(\theta_m^\star, \pi_m) \right) \leq \beta_m,$$

According to the definition of optimistic discretization, we obtain that the following inequality holds with probability at least $1 - \delta$ for all $m \in [n]$:

$$\max_{(\theta_m, t) \in \bar{\Theta}_m \times [T]} \sum_{i=1}^{t} \log \left( \tilde{f}_m^i(\theta_m, \pi_m) / \tilde{f}_m^i(\theta_m^\star, \pi_m) \right) \leq \beta_m,$$

$\square$

**Lemma C.2.** *There exists a universal constant $c$ such that for any $\delta \in (0,1]$, with probability at least for all $t \in [T]$ and all $\theta_m \in \Theta_m$, $m \in [n]$, it holds that*

$$
\sum_{i=1}^{t} \left( \sum_{\tau} \left| f_m\left(\theta_m, \pi_m^i\right) - f_m\left(\theta_m^\star, \pi_m^i\right) \right| \left[ \prod_{l=1}^{m-1} f_l\left(\theta_l^\star, \pi_l^i\right) \right] \left[ \prod_{j=m+1}^{n} f_l\left(\theta_j^i, \pi_j^i\right) \right] \right)^2
$$

$$
\lesssim \left( \sum_{i=1}^{t} \log\left( \tilde{f}_m^i\left(\theta_m^\star, \pi_m^i\right) / \tilde{f}_m^i\left(\theta_m, \pi_m^i\right) \right) + \beta_m \right)
$$

*Proof.* We define tangent trajectory sample $\widehat{\tau}^i$ that satisfies $\widehat{\tau}^i \sim \prod_{i=1}^{m} \mathbb{P}_{\theta_*,i}^{\pi_i^k} \prod_{j=m+1}^{n} \mathbb{P}_{\theta_j^k,j}^{\pi_j^k}$ but are independent with $\tau^i$. With similar analysis as Lemma 15 of Liu et al. (2022a), we obtain that

$$
\mathbb{E}\left[ \exp\left( \sum_{i=1}^{t} \frac{1}{2} \log\left( \frac{\tilde{f}_m^i\left(\bar{\theta}_m, \pi_m^i\right)}{\tilde{f}_m^i\left(\theta_m^\star, \pi_m^i\right)} \right) - \log \mathbb{E}\left[ \exp\left( \frac{1}{2} \log\left( \frac{\widehat{f}_m^i\left(\bar{\theta}_m, \pi_m^i\right)}{\widehat{f}_m^i\left(\theta_m^\star, \pi_m^i\right)} \right) \right) \Big| \mathcal{E}_m \right] \right) \right] = 1,
$$

where for all $(\theta_m, m) \in \Theta_m \times [n]$, we denote $\widehat{f}_m^i(\theta_m, \pi_m^i)$ as $\widehat{f}_m^i(\theta_m, \pi_m^i) = \mathbb{P}_{\theta_m,m}^{\pi_m^i}(\tau_m^i \mid \{\tau_r^i\}_{r \in \mathsf{pa}(m)})$, and we denote $\mathcal{E}_m, \widehat{\mathcal{E}}_m$ as $\mathcal{E}_m = \{(\pi^i, \tau^i)\}_{i=1}^{t}$, $\widehat{\mathcal{E}}_m = \{(\pi^i, \widehat{\tau}^i)\}_{i=1}^{t}$. With Chernoff's method, we can obtain that with probability at least $1 - \delta$, for all $\bar{\theta}_m \in \bar{\Theta}_m$ we have

$$
- \log \mathbb{E}_{\widehat{\mathcal{E}}_m}\left[ \exp\left( \sum_{i=1}^{t} \frac{1}{2} \log\left( \frac{\widehat{f}_m^i\left(\bar{\theta}_m, \pi_m^i\right)}{\widehat{f}_m^i\left(\theta_m^\star, \pi_m^i\right)} \right) \right) \Big| \mathcal{E}_m \right] \leq - \sum_{i=1}^{t} \frac{1}{2} \log\left( \frac{\widehat{f}_m^i\left(\bar{\theta}_m, \pi_m^i\right)}{\widehat{f}_m^i\left(\theta_m^\star, \pi_m^i\right)} \right) + \beta_m.
$$

(9)

Then, we apply elementary inequality $- \log x \geq 1 - x$, and we can obtain that

$$
- \log \mathbb{E}_{\widehat{\mathcal{E}}_m}\left[ \exp\left( \sum_{i=1}^{t} \frac{1}{2} \log\left( \frac{\widehat{f}_m^i\left(\bar{\theta}_m, \pi_m^i\right)}{\widehat{f}_m^i\left(\theta_m^\star, \pi_m^i\right)} \right) \right) \Big| \mathcal{E}_m \right]
$$

$$
= - \sum_{i=1}^{t} \log \mathbb{E}_{\tau \sim \left( \prod_{l=1}^{m} \mathbb{P}_{\theta^*,l}^{\pi_l^i} \right)\left( \prod_{j=m+1}^{n} \mathbb{P}_{\theta_j^i,j}^{\pi_j^i} \right)} \left[ \sqrt{\frac{f_m\left(\bar{\theta}_m, \pi_m^i\right)}{f_m\left(\theta_m^\star, \pi_m^i\right)}} \right]
$$

$$
\geq \sum_{i=1}^{t} \left( 1 - \mathbb{E}_{\tau \sim \left( \prod_{l=1}^{m} \mathbb{P}_{\theta^*,l}^{\pi_l^i} \right)\left( \prod_{j=m+1}^{n} \mathbb{P}_{\theta_j^i,j}^{\pi_j^i} \right)} \left[ \sqrt{\frac{f_m\left(\bar{\theta}_m, \pi_m^i\right)}{f_m\left(\theta_m^\star, \pi_m^i\right)}} \right] \right)
$$

$$
= \sum_{i=1}^{t} \left( 1 - \sum_{\tau} \sqrt{f_m\left(\bar{\theta}_m, \pi_m^i\right) \cdot f_m\left(\theta_m^\star, \pi_m^i\right)} \left[ \prod_{l=1}^{m-1} f_l\left(\theta_l^\star, \pi_l^i\right) \right] \left[ \prod_{j=m+1}^{n} f_j\left(\theta_j^i, \pi_j^i\right) \right] \right).
$$

To continue, we aim to achieve the lower bound for the following term of interest:

$$
- \log \mathbb{E}_{\widehat{\mathcal{E}}_m}\left[ \exp\left( \sum_{i=1}^{t} \frac{1}{2} \log\left( \frac{\widehat{f}_m^i\left(\bar{\theta}_m, \pi_m^i\right)}{\widehat{f}_m^i\left(\theta_m^\star, \pi_m^i\right)} \right) \right) \Big| \mathcal{E}_m \right] + \frac{1}{2}.
$$

We have the folllowing inequalities:

$$
- \log \mathbb{E}_{\widehat{\mathcal{E}}_m}\left[ \exp\left( \sum_{i=1}^{t} \frac{1}{2} \log\left( \frac{\widehat{f}_m^i\left(\bar{\theta}_m, \pi_m^i\right)}{\widehat{f}_m^i\left(\theta_m^\star, \pi_m^i\right)} \right) \right) \Big| \mathcal{E}_m \right] + \frac{1}{2}
$$

$$
\geq \sum_{i=1}^{t} \left( 1 - \sum_{\tau} \sqrt{f_m\left(\bar{\theta}_m, \pi_m^i\right) \cdot f_m\left(\theta_m^\star, \pi_m^i\right)} \left[ \prod_{l=1}^{m-1} f_l\left(\theta_l^\star, \pi_l^i\right) \right] \left[ \prod_{j=m+1}^{n} f_j\left(\theta_j^i, \pi_j^i\right) \right] \right) + \frac{1}{2}
$$

$$
\geq \frac{1}{2} \sum_{i=1}^{t} \sum_{\tau} \left( \sqrt{\left[ \prod_{l=1}^{m-1} f_l\left(\theta_l^\star, \pi_l^i\right) \right] f_m\left(\bar{\theta}_m, \pi_m^i\right) \left[ \prod_{j=m+1}^{n} f_j\left(\theta_j^i, \pi_j^i\right) \right]} - \sqrt{\left[ \prod_{l=1}^{m} f_l\left(\theta_l^\star, \pi_l^i\right) \right] \left[ \prod_{j=m+1}^{n} f_j\left(\theta_j^i, \pi_j^i\right) \right]} \right)^2.
$$

With elementary calculation, we have

$$
-\log \mathbb{E}_{\widehat{\mathcal{E}}_m}\left[\exp\left(\sum_{i=1}^{t}\frac{1}{2}\log\left(\frac{\widehat{f}_m^i\left(\bar{\theta}_m,\pi_m^i\right)}{\widehat{f}_m^i\left(\theta_m^\star,\pi_m^i\right)}\right)\right)\Bigg|\,\mathcal{E}_m\right]+\frac{1}{2}
$$

$$
\geq\frac{1}{12}\sum_{i=1}^{t}\left[\sum_{\boldsymbol{\tau}}\left(\sqrt{\left[\prod_{l=1}^{m-1}f_l\left(\theta_l^\star,\pi_l^i\right)\right]f_m\left(\bar{\theta}_m,\pi_m^i\right)\left[\prod_{j=m+1}^{n}f_j\left(\theta_j^i,\pi_j^i\right)\right]}-\sqrt{\left[\prod_{l=1}^{m}f_l\left(\theta_l^\star,\pi_l^i\right)\right]\left[\prod_{j=m+1}^{n}f_j\left(\theta_j^i,\pi_j^i\right)\right]}\right)^2\right]
$$

$$
\cdot\left[\sum_{\boldsymbol{\tau}}\left(\sqrt{\left[\prod_{l=1}^{m-1}f_l\left(\theta_l^\star,\pi_l^i\right)\right]f_m\left(\bar{\theta}_m,\pi_m^i\right)\left[\prod_{j=m+1}^{n}f_j\left(\theta_j^i,\pi_j^i\right)\right]}+\sqrt{\left[\prod_{l=1}^{m}f_l\left(\theta_l^\star,\pi_l^i\right)\right]\left[\prod_{j=m+1}^{n}f_j\left(\theta_j^i,\pi_j^i\right)\right]}\right)^2\right]
$$

We then apply Cauchy-Schawarz inequality, and we arrive at

$$
-\log \mathbb{E}_{\widehat{\mathcal{E}}_m}\left[\exp\left(\sum_{i=1}^{t}\frac{1}{2}\log\left(\frac{\widehat{f}_m^i\left(\bar{\theta}_m,\pi_m^i\right)}{\widehat{f}_m^i\left(\theta_m^\star,\pi_m^i\right)}\right)\right)\Bigg|\,\mathcal{E}_m\right]+\frac{1}{2}
$$

$$
\geq\frac{1}{12}\sum_{i=1}^{t}\left(\sum_{\boldsymbol{\tau}}\left|f_m\left(\bar{\theta}_m,\pi_m^i\right)-f_m\left(\theta_m^\star,\pi_m^i\right)\right|\left[\prod_{l=1}^{m-1}f_l\left(\theta_l^\star,\pi_l^i\right)\right]\left[\prod_{j=m+1}^{n}f_j\left(\theta_j^i,\pi_j^i\right)\right]\right)^2-\frac{1}{2}
$$

$$\tag{10}$$

We insert eq. (10) back into eq. (9), and we obtain that there exist a universal constant $c$ such that for any $\delta\in(0,1]$, with probability at least $1-\delta$ for all $t\in[T]$, $m\in[n]$, and all $\theta_m\in\Theta_m$, it holds that

$$
\sum_{i=1}^{t}\left(\sum_{\boldsymbol{\tau}}\left|f_m\left(\theta_m,\pi_m^i\right)-f_m\left(\theta_m^\star,\pi_m^i\right)\right|\left[\prod_{l=1}^{m-1}f_l\left(\theta_l^\star,\pi_l^i\right)\right]\left[\prod_{j=m+1}^{n}f_l\left(\theta_j^i,\pi_j^i\right)\right]\right)^2
$$

$$
\lesssim\left(\sum_{i=1}^{t}\log\left(\tilde{f}_m^i\left(\theta_m^\star,\pi_m^i\right)/\tilde{f}_m^i\left(\theta_m,\pi_m^i\right)\right)+\beta_m\right)
$$

$\square$

We combine this result with the update rule of Algorithm 4.2.

**Corollary C.1.** *With probability at least $1-\delta$, for all $k\in[K]$, $m\in[n]$, the following inequality holds.*

$$
\sum_{t=1}^{k-1}\sum_{\{\tau_{H,r}\}_{r\in[n]}}\left|f_m\left(\theta_m^k,\pi_m^t\right)-f_m\left(\theta_m^\star,\pi_m^t\right)\right|\left[\prod_{l=1}^{m-1}f_l\left(\theta_l^\star,\pi_l^t\right)\right]\left[\prod_{j=m+1}^{n}f_j\left(\theta_j^t,\pi_j^t\right)\right]\lesssim\sqrt{\beta_m k}.
$$

According to Lemma C.2 and Lemma D.1, with probability at least $1-\delta$, for all $k\in[K]$, $m\in[n]$, the following inequality holds.

$$
\sum_{t=1}^{k-1}\left(\sum_{\{\tau_{H,r}\}_{r\in[n]}}\left|f_m\left(\theta_m^k,\pi_m^t\right)-f_m\left(\theta_m^\star,\pi_m^t\right)\right|\left[\prod_{l=1}^{m-1}f_l\left(\theta_l^\star,\pi_l^t\right)\right]\left[\prod_{j=m+1}^{n}f_j\left(\theta_j^t,\pi_j^t\right)\right]\right)^2\lesssim\beta_m.
$$

We apply Cauchy-Schawarz inequality, and we can obtain that

$$
\sum_{t=1}^{k-1}\sum_{\{\tau_{H,r}\}_{r\in[n]}}\left|f_m\left(\theta_m^k,\pi_m^t\right)-f_m\left(\theta_m^\star,\pi_m^t\right)\right|\left[\prod_{l=1}^{m-1}f_l\left(\theta_l^\star,\pi_l^t\right)\right]\left[\prod_{j=m+1}^{n}f_j\left(\theta_j^t,\pi_j^t\right)\right]\lesssim\sqrt{\beta_m k}.
$$

**Definition C.2.** *For all* $(m, h, k) \in [n] \times [H] \times [T]$, *we define the matrix notations* $\mathbb{M}_{h,m}^\star \in \mathbb{R}^{O \times O}$ *and* $\mathbb{M}_{h,m}^k \in \mathbb{R}^{O \times O}$ *as follows:*

$$\mathbb{M}_{0,m}^\star = \mathbb{O}_{1,m}^\star \mu_m^\star \in \mathbb{R}^O, \quad \mathbb{M}_{0,m}^k = \mathbb{O}_{1,m}^k \mu_m^k \in \mathbb{R}^O,$$

$$\mathbb{M}_{h,m}^\star(o_m, a_m, \{a_r\}_{r \in \mathsf{pa}(m)}) = \mathbb{O}_{h+1,m}^\star \mathbb{T}_{h,m,a_m,\{a_r\}_{r \in \mathsf{pa}(m)}}^\star \cdot diag\left(\mathbb{O}_{h,m}^\star(o_m \mid \cdot)\right) \left(\mathbb{O}_{h,m}^\star\right)^\dagger \in \mathbb{R}^{O \times O},$$

$$\mathbb{M}_{h,m}^k(o_m, a_m, \{a_r\}_{r \in \mathsf{pa}(m)}) = \mathbb{O}_{h+1,m}^k \mathbb{T}_{h,m,a_m,\{a_r\}_{r \in \mathsf{pa}(m)}}^k \cdot diag\left(\mathbb{O}_{h,m}^k(o_m \mid \cdot)\right) \left(\mathbb{O}_{h,m}^k\right)^\dagger \in \mathbb{R}^{O \times O},$$

*where* $\{\mathbb{O}_{h,m}^\star\}_{(h,m) \in [H] \times [n]}$ *and* $\{\mathbb{T}_{h,m,a_m,\{a_r\}_{r \in \mathsf{pa}(m)}}^\star\}_{(h,m) \in [H] \times [n]}$ *denote the observation and transition matrices corresponding to the true transition model, and* $\{\mathbb{O}_{h,m}^k\}_{(h,m) \in [H] \times [n]}$ *and* $\{\mathbb{T}_{h,m,a_m,\{a_r\}_{r \in \mathsf{pa}(m)}}^k\}_{(h,m) \in [H] \times [n]}$ *denote the observation and transition matrices corresponding to model parameter* $\theta_m^k$ *for all* $k \in [T]$. *When no confusion arises, we simplify the notation by using* $\mathbb{M}_{h,m}^\star$ *to represent* $\mathbb{M}_{h,m}^\star(o_m, a_m, \{a_r\}_{r \in \mathsf{pa}(m)})$ *and* $\mathbb{M}_{h,m}^k$ *to represent* $\mathbb{M}_{h,m}^k(o_m, a_m, \{a_r\}_{r \in \mathsf{pa}(m)})$.

Since marginalizing two distribution will not increase their TV distance, so for all $(k, h) \in [K] \times [H-1]$, we have the following corollary.

**Corollary C.2.** *With probability at least* $1 - \delta$, *for all* $(k, h) \in [T] \times [H-1]$, *the following inequality holds true.*

$$\sum_{t=1}^{k-1} \sum_{\boldsymbol{\tau}_h} \pi_m^t(\tau_{h,m}) \left\| \left[\prod_{h'=0}^{h} \mathbb{M}_{h',m}^k\right] - \left[\prod_{h'=0}^{h} \mathbb{M}_{h',m}^\star\right] \right\|_1 \left[\prod_{l=1}^{m-1} \left\| \prod_{h'=0}^{h} \mathbb{M}_{h',l}^\star \right\|_1 \pi_l^t(\tau_{h,l}) \right]$$

$$\cdot \left[\prod_{j=m+1}^{n} \left\| \prod_{h'=0}^{h} \mathbb{M}_{h',j}^t \right\|_1 \pi_j^t(\tau_{h,j}) \right] \lesssim \sqrt{k\beta_m}.$$

**Lemma C.3.** *With probability at least* $1 - \delta$, *for all* $(k, h, m) \in [T] \times [H-1] \times [n]$,

$$\sum_{t=1}^{k-1} \sum_{\boldsymbol{\tau}_h} \pi_m^t(\tau_{h,m}) \cdot \left\| \left(\mathbb{M}_{h,m}^k - \mathbb{M}_{h,m}^\star\right) \left[\prod_{h'=0}^{h-1} \mathbb{M}_{h',m}^\star\right] \right\|_1 \cdot \left[\prod_{l=1}^{m-1} \left\| \prod_{h'=0}^{h} \mathbb{M}_{h',l}^\star \right\|_1 \cdot \pi_l^t(\tau_{h,l}) \right]$$

$$\cdot \left[\prod_{j=m+1}^{n} \left\| \prod_{h'=0}^{h} \mathbb{M}_{h',j}^\star \right\|_1 \cdot \pi_j^t(\tau_{h,j}) \right] \lesssim \sqrt{Sk\beta_m}/\alpha.$$

*Proof.* We intend to bound the following term:

$$\sum_{t=1}^{k-1} \sum_{\boldsymbol{\tau}_h} \pi_m^t(\tau_{h,m}) \left\| \left(\mathbb{M}_{h,m}^k - \mathbb{M}_{h,m}^\star\right) \left[\prod_{h'=0}^{h-1} \mathbb{M}_{h',m}^\star\right] \right\|_1 \left[\prod_{l=1}^{m-1} \left\| \prod_{h'=0}^{h} \mathbb{M}_{h',l}^\star \right\|_1 \pi_l^t(\tau_{h,l}) \right] \left[\prod_{j=m+1}^{n} \left\| \prod_{h'=0}^{h} \mathbb{M}_{h',j}^\star \right\|_1 \pi_j^t(\tau_{h,j}) \right]$$

We initially have the following decomposition:

$$\left\| \left(\mathbb{M}_{h,m}^k - \mathbb{M}_{h,m}^\star\right) \left[\prod_{h'=0}^{h-1} \mathbb{M}_{h',m}^\star\right] \right\|_1$$

$$\leq \left\| \mathbb{M}_{h,m}^k \left[\prod_{h'=0}^{h-1} \mathbb{M}_{h',m}^k\right] - \mathbb{M}_{h,m}^\star \left[\prod_{h'=0}^{h-1} \mathbb{M}_{h',m}^\star\right] \right\|_1 + \left\| \mathbb{M}_{h,m}^k \left( \left[\prod_{h'=0}^{h-1} \mathbb{M}_{h',m}^k\right] - \left[\prod_{h'=0}^{h-1} \mathbb{M}_{h',m}^\star\right] \right) \right\|_1$$

According to the result in Corollary C.2, we obtain that

$$\sum_{t=1}^{k-1} \sum_{\boldsymbol{\tau}_h} \pi_m^t(\tau_{h,m}) \left\| \left[\prod_{h'=0}^{h} \mathbb{M}_{h',m}^k\right] - \left[\prod_{h'=0}^{h} \mathbb{M}_{h',m}^\star\right] \right\|_1$$

$$\cdot \left[\prod_{l=1}^{m-1} \left\| \prod_{h'=0}^{h} \mathbb{M}_{h',l}^\star \right\|_1 \pi_l^t(\tau_{h,l}) \right] \left[\prod_{j=m+1}^{n} \left\| \prod_{h'=0}^{h} \mathbb{M}_{h',j}^\star \right\|_1 \pi_j^t(\tau_{h,j}) \right] \lesssim \sqrt{k\beta_m}. \tag{11}$$

According to the definition of matrix operator, we have

$$
\sum_{t=1}^{k-1} \sum_{\boldsymbol{\tau}_h} \pi_m^t(\tau_{h,m}) \left\| \mathbb{M}_{h,m}^k \left( \left[ \prod_{h'=0}^{h-1} \mathbb{M}_{h',m}^k \right] - \left[ \prod_{h'=0}^{h-1} \mathbb{M}_{h',m}^\star \right] \right) \right\|_1
$$
$$
\cdot \left[ \prod_{l=1}^{m-1} \left\| \prod_{h'=0}^{h} \mathbb{M}_{h',l}^\star \right\|_1 \pi_l^t(\tau_{h,l}) \right] \left[ \prod_{j=m+1}^{n} \left\| \prod_{h'=0}^{h} \mathbb{M}_{h',j}^\star \right\|_1 \pi_j^t(\tau_{h,j}) \right] \lesssim \sqrt{Sk\beta_m}/\alpha. \tag{12}
$$

We combine eq. (11) and eq. (12), and we enventually arrive at:

$$
\sum_{t=1}^{k-1} \sum_{\boldsymbol{\tau}_h} \pi_m^t(\tau_{h,m}) \cdot \left\| \left( \mathbb{M}_{h,m}^k - \mathbb{M}_{h,m}^\star \right) \left[ \prod_{h'=0}^{h-1} \mathbb{M}_{h',m}^\star \right] \right\|_1 \cdot \left[ \prod_{l=1}^{m-1} \left\| \prod_{h'=0}^{h} \mathbb{M}_{h',l}^\star \right\|_1 \cdot \pi_l^t(\tau_{h,l}) \right]
$$
$$
\cdot \left[ \prod_{j=m+1}^{n} \left\| \prod_{h'=0}^{h} \mathbb{M}_{h',j}^\star \right\|_1 \cdot \pi_j^t(\tau_{h,j}) \right] \lesssim \sqrt{Sk\beta_m}/\alpha,
$$

holds with probability at least $1 - \delta$ for all $(k,h) \in [T] \times [H-1]$. □

**Lemma C.4.** *The regret is bounded by the following inequality:*

$$
Regret(k) = \sum_{t=1}^{k} V^{\boldsymbol{\pi}^*} - V^{\boldsymbol{\pi}^t} \leq nH \sum_{t=1}^{k} \sum_{\boldsymbol{\tau}_H} \left| \mathbb{P}_{\boldsymbol{\theta}^t}^{\boldsymbol{\pi}^t}(\boldsymbol{\tau}_H) - \mathbb{P}_{\boldsymbol{\theta}^*}^{\boldsymbol{\pi}^t}(\boldsymbol{\tau}_H) \right|,
$$

*where we define*

$$
\mathbb{P}_{\boldsymbol{\theta}^t}^{\boldsymbol{\pi}^t}(\boldsymbol{\tau}_H) = \prod_{m=1}^{n} \mathbb{P}_{\theta_m^t}^{\pi_m^t}(\tau_{H,m} \mid \{\tau_{H,r}\}_{r \in \mathsf{pa}(m)}), \quad \mathbb{P}_{\boldsymbol{\theta}^*}^{\boldsymbol{\pi}^t}(\boldsymbol{\tau}_H) = \prod_{m=1}^{n} \mathbb{P}_{\theta_m^*}^{\pi_m^t}(\tau_{H,m} \mid \{\tau_{H,r}\}_{r \in \mathsf{pa}(m)}).
$$

*Proof.* We can strightforwardly achieve this result according to the definition of value function and regret. □

**Lemma C.5.** *The regret is bounded by the following inequality:*

$$
Regret(k) = \sum_{t=1}^{k} V^{\boldsymbol{\pi}^*} - V^{\boldsymbol{\pi}^t}
$$
$$
\leq nH \sum_{t=1}^{k} \sum_{m=1}^{n} \sum_{h=1}^{H-1} \sum_{\{\tau_{h,r}\}_{r \in [n]}} \frac{\sqrt{S}}{\alpha} \cdot \pi_m^t(\tau_{h,m}) \cdot \left\| \left( \mathbb{M}_{h,m}^t - \mathbb{M}_{h,m}^\star \right) \left[ \prod_{h'=0}^{h} \mathbb{M}_{h',m}^\star \right] \right\|_1
$$
$$
\cdot \left[ \prod_{l=1}^{m-1} \left\| \prod_{h'=0}^{h} \mathbb{M}_{h',l}^\star \right\|_1 \cdot \pi_l^t(\tau_{h,l}) \right] \cdot \left[ \prod_{j=m+1}^{n} \left\| \prod_{h'=0}^{h} \mathbb{M}_{h',j}^t \right\|_1 \cdot \pi_j^t(\tau_{h,j}) \right]
$$

*Proof.* According to the definition of transition model factorization, we have

$$
\sum_{t=1}^{k} \sum_{\boldsymbol{\tau}_H} \left| \mathbb{P}_{\boldsymbol{\theta}^t}^{\boldsymbol{\pi}^t}(\boldsymbol{\tau}_H) - \mathbb{P}_{\boldsymbol{\theta}^*}^{\boldsymbol{\pi}^t}(\boldsymbol{\tau}_H) \right|
$$
$$
= \sum_{t=1}^{k} \sum_{\boldsymbol{\tau}_H} \left| \prod_{m=1}^{n} \mathbb{P}_{\theta_m^t}^{\pi_m^t} \left( \tau_{H,m} \mid \{\tau_{H,r}\}_{r \in \mathsf{pa}(m)} \right) - \prod_{m=1}^{n} \mathbb{P}_{\theta_m^*}^{\pi_m^t} (\tau_{H,m} \left( \tau_{H,m} \mid \{\tau_{H,r}\}_{r \in \mathsf{pa}(m)} \right)) \right|
$$

Moreover, we have the following inequality of difference between transition probability measure.

$$
\sum_{t=1}^{k} \sum_{\boldsymbol{\tau}_H} \left| \mathbb{P}_{\boldsymbol{\theta}^t}^{\boldsymbol{\pi}^t}(\boldsymbol{\tau}_H) - \mathbb{P}_{\boldsymbol{\theta}^*}^{\boldsymbol{\pi}^t}(\boldsymbol{\tau}_H) \right|
$$
$$
= \sum_{t=1}^{k} \sum_{\boldsymbol{\tau}_H} \sum_{m=1}^{n} \left[ \prod_{j=1}^{m-1} f_j \left( \theta_j^\star, \pi_j^t \right) \right] \left| f_m \left( \theta_m^t, \pi_m^t \right) - f_m \left( \theta_m^\star, \pi_m^t \right) \right| \left[ \prod_{l=m+1}^{n} f_l \left( \theta_l^t, \pi_l^t \right) \right] \tag{13}
$$

Moreover, according to the definition of matrix notation, we can rewrite the term $|f_m(\theta_m^t, \pi_m^t) - f_m(\theta_m^\star, \pi_m^t)|$ as

$$\left|f_m\left(\theta_m^t, \pi_m^t\right) - f_m\left(\theta_m^\star, \pi_m^t\right)\right| = \left\|\left[\prod_{h'=0}^{H-1} \mathbb{M}_{h,m}^t\right] - \left[\prod_{h'=0}^{H-1} \mathbb{M}_{h,m}^\star\right]\right\|_1 \pi_m^t(\tau_{H,m})$$

Then we have the following inequality:

$$\left\|\left[\prod_{h'=0}^{h} \mathbb{M}_{h,m}^t\right] - \left[\prod_{h'=0}^{h} \mathbb{M}_{h,m}^\star\right]\right\|_1 \pi_m^t(\tau_{H,m}) \leq \sum_{j=1}^{H-1}\left\|\left[\prod_{h=j+1}^{H-1} \mathbb{M}_{h,m}^t\right]\left(\mathbb{M}_{j,m}^t - \mathbb{M}_{j,m}^\star\right)\left[\prod_{h=0}^{j-1} \mathbb{M}_{h,m}^\star\right]\right\|_1 \cdot \pi_m^t(\tau_{H,m})$$
(14)

We insert eq. (14) back into eq. (13), and we can obtain that

$$\sum_{\boldsymbol{\tau}_H}\left[\prod_{j=1}^{m-1} f_j\left(\theta_j^\star, \pi_j^t\right)\right]\left|f_m\left(\theta_m^t, \pi_m^t\right) - f_m\left(\theta_m^\star, \pi_m^t\right)\right|\left[\prod_{l=m+1}^{n} f_l\left(\theta_l^t, \pi_l^t\right)\right]$$

$$\leq \sum_{\boldsymbol{\tau}_H}\sum_{j=1}^{H-1}\left\|\left[\prod_{h=j+1}^{H-1} \mathbb{M}_{h,m}^t\right]\left(\mathbb{M}_{j,m}^t - \mathbb{M}_{j,m}^\star\right)\left[\prod_{h=0}^{j-1} \mathbb{M}_{h,m}^\star\right]\right\|_1 \pi_m^t(\tau_{H,m})\left[\prod_{j=1}^{m-1} f_j\left(\theta_j^\star, \pi_j^t\right)\right]\left[\prod_{l=m+1}^{n} f_l\left(\theta_l^t, \pi_l^t\right)\right]$$

$$\leq \sum_{h=1}^{H-1}\sum_{\boldsymbol{\tau}_h}\frac{\sqrt{S}}{\alpha}\left\|\left(\mathbb{M}_{h,m}^t - \mathbb{M}_{h,m}^\star\right)\left[\prod_{h'}^{h-1} \mathbb{M}_{h',m}^\star\right]\right\|_1 \cdot \pi_m^t(\tau_{H,m})$$

$$\cdot \left[\prod_{j=1}^{m-1}\left[\prod_{h'=0}^{h} \mathbb{M}_{j,m}^\star\right]\pi_j^t(\tau_{h,j})\right] \cdot \left[\prod_{l=m+1}^{n}\left[\prod_{h'=0}^{h} \mathbb{M}_{l,m}^t\right]\pi_l^t(\tau_{h,l})\right]$$

Eventually, the target regret can be bounded with

$$\text{Regret}(k) \leq \sum_{t=1}^{k}\sum_{\boldsymbol{\tau}_H}\left|\mathbb{P}_{\boldsymbol{\theta}^t}^{\boldsymbol{\pi}^t}(\boldsymbol{\tau}_H) - \mathbb{P}_{\boldsymbol{\theta}^*}^{\boldsymbol{\pi}^t}(\boldsymbol{\tau}_H)\right|$$

$$= \sum_{t=1}^{k}\sum_{\boldsymbol{\tau}_H}\left|\prod_{m=1}^{n}\mathbb{P}_{\theta_m^t}^{\pi_m^t}(\tau_{H,m}|\tau_{H,m-1}) - \prod_{m=1}^{n}\mathbb{P}_{\theta_m^*}^{\pi_m^t}(\tau_{H,m}|\tau_{H,m-1})\right|$$

$$\leq \sum_{t=1}^{k}\sum_{\boldsymbol{\tau}_H}\sum_{m=1}^{n}\left[\prod_{j=1}^{m-1} f_j\left(\theta_j^\star, \pi_j^t\right)\right]\left|f_m\left(\theta_m^t, \pi_m^t\right) - f_m\left(\theta_m^\star, \pi_m^t\right)\right|\left[\prod_{l=m+1}^{n} f_l\left(\theta_l^t, \pi_l^t\right)\right]$$

$$\leq nH\sum_{t=1}^{k}\sum_{m=1}^{n}\sum_{h=1}^{H-1}\sum_{\{\tau_{h,r}\}_{r\in[n]}}\frac{\sqrt{S}}{\alpha}\cdot\pi_m^t(\tau_{h,m})\cdot\left\|\left(\mathbb{M}_{h,m}^t - \mathbb{M}_{h,m}^\star\right)\left[\prod_{h'=0}^{h} \mathbb{M}_{h',m}^\star\right]\right\|_1$$

$$\cdot\left[\prod_{l=1}^{m-1}\left\|\prod_{h'=0}^{h} \mathbb{M}_{h',l}^\star\right\|_1\cdot\pi_l^t(\tau_{h,l})\right]\cdot\left[\prod_{j=m+1}^{n}\left\|\prod_{h'=0}^{h} \mathbb{M}_{h',j}^t\right\|_1\cdot\pi_j^t(\tau_{h,j})\right]$$

$\square$

**Proof for Theorem 4.1**   With the Lemmas provided above, we now present the proof for Theorem 4.1, We first restate the theorem as follows:

**Theorem C.1.** *We select bonus parameter as* $\beta_m = H^2(S^2 A^{|\mathsf{pa}(m)|+1} + SO)\log(TSAOH) + \log(Tn/\delta)$ *for some constant c. Then with probability at least* $1 - \delta$*, Algorithm 4.2 guarantees that the following inequality holds true.*

$$\text{Regret}(k) = \sum_{t=1}^{k} V^{\boldsymbol{\pi}^*} - V^{\boldsymbol{\pi}^t} \leq \tilde{\mathcal{O}}\left(\sum_{m=1}^{n}\frac{S^2 O A^{|\mathsf{pa}(m)|+1}}{\alpha^2}\sqrt{k(S^2 A^{|\mathsf{pa}(m)|+1} + SO)}\right),$$

*where we define* $\boldsymbol{\pi}^*$ *as* $\boldsymbol{\pi}^* = \arg\max_{\boldsymbol{\pi}} V^{\boldsymbol{\pi}}$*.*

*Proof.* According to Lemma C.5, it is sufficient to obtain an upper bound for the following term:

$$\sum_{t=1}^{k}\sum_{h=1}^{H-1}\sum_{\boldsymbol{\tau}_h}\pi_m^t(\tau_{h,m})\left\|(\mathbb{M}_{h,m}^t-\mathbb{M}_{h,m}^\star)\left[\prod_{h'=0}^{h}\mathbb{M}_{h',m}^\star\right]\right\|_1$$

$$\cdot\left[\prod_{l=1}^{m-1}\left\|\prod_{h'=0}^{h}\mathbb{M}_{h',l}^\star\right\|_1\cdot\pi_l^t(\tau_{h,l})\right]\cdot\left[\prod_{j=m+1}^{n}\left\|\prod_{h'=0}^{h}\mathbb{M}_{h',j}^t\right\|_1\cdot\pi_j^t(\tau_{h,j})\right]$$

For probability at least $1-\delta$, we have

$$\sum_{t=1}^{k}\sum_{h=1}^{H-1}\sum_{\boldsymbol{\tau}_h}\pi_m^t(\tau_{h,m})\left\|(\mathbb{M}_{h,m}^k-\mathbb{M}_{h,m}^\star)\left[\prod_{h'=0}^{h}\mathbb{M}_{h',m}^\star\right]\right\|_1\cdot\left[\prod_{l=1}^{m-1}\left\|\prod_{h'=0}^{h}\mathbb{M}_{h',l}^\star\right\|_1\cdot\pi_l^t(\tau_{h,l})\right]$$

$$\cdot\left[\prod_{j=m+1}^{n}\left\|\prod_{h'=0}^{h}\mathbb{M}_{h',j}^t\right\|_1\cdot\pi_j^t(\tau_{h,j})\right]\lesssim\sqrt{Sk\beta_m}/\alpha$$

For $m\in[n]$, we fix $(o,a,\{a_r\}_{r\in\mathsf{pa}(m)})\in\mathcal{O}_m\times\mathcal{A}_m\times\left(\times_{r\in\mathsf{pa}(m)}\mathcal{A}_r\right)$. We define the set of trajectories $\{\tau_{h,r}\}_{r\in[n]}$, denoted by $\mathcal{C}_m$, as:

$$\mathcal{C}_m=\left\{\boldsymbol{\tau}_h\mid\{\tau_{h,r}\}_{r\in[n]}:(o_{h,m},a_{h,m},\{a_{h,r}\}_{r\in\mathsf{pa}(m)})=(o,a,\{a_r\}_{r\in\mathsf{pa}(m)})\right\}.$$

Then, we have the following condition:

$$\sum_{t=1}^{k}\sum_{h=1}^{H-1}\sum_{\boldsymbol{\tau}_h}\pi_m^t(\tau_{h,m})\left\|\left[(\mathbb{M}_{h,m}^k-\mathbb{M}_{h,m}^\star)\,\mathbb{O}_{h,m}^\star\right](\mathbb{O}_{h,m}^\star)^\dagger\left[\prod_{h'=0}^{h}\mathbb{M}_{h',m}^\star\right]\right\|_1$$

$$\cdot\left[\prod_{l=1}^{m-1}\left\|\prod_{h'=0}^{h}\mathbb{M}_{h',l}^\star\right\|_1\cdot\pi_l^t(\tau_{h,l})\right]\cdot\left[\prod_{j=m+1}^{n}\left\|\prod_{h'=0}^{h}\mathbb{M}_{h',j}^t\right\|_1\cdot\pi_j^t(\tau_{h,j})\right]\lesssim\sqrt{Sk\beta_m}/\alpha$$

We define $\{w_{t,l}\}_{(t,l)\in[T]\times[O]}$ that satisfies:

$$w_{t,l}=\left[(\mathbb{M}_{h,m}^t(o,a,\{a_r\}_{r\in\mathsf{pa}(m)})-\mathbb{M}_{h,m}(o,a,\{a_r\}_{r\in\mathsf{pa}(m)}))\mathbb{O}_{h,m}\right]_l.$$

We denote the sequence

$$\pi_m^t(\tau_{h,m})\,(\mathbb{O}_{h,m}^\star)^\dagger\left[\prod_{h'=0}^{h}\mathbb{M}_{h',m}^\star\right]\cdot\left[\prod_{l=1}^{m-1}\left\|\prod_{h'=0}^{h}\mathbb{M}_{h',l}^\star\right\|_1\cdot\pi_l^t(\tau_{h,l})\right]\cdot\left[\prod_{j=m+1}^{n}\left\|\prod_{h'=0}^{h}\mathbb{M}_{h',j}^t\right\|_1\cdot\pi_j^t(\tau_{h,j})\right]$$

for all $\boldsymbol{\tau}_h:(o_{h,m},a_{h,m},\{a_{h,r}\}_{r\in\mathsf{pa}(m)})=(o,a,\{a_r\}_{r\in\mathsf{pa}(m)})$ by $x_{t,1},x_{t,2},\ldots,x_{t,N}$, where

$$N=\left|\left\{\boldsymbol{\tau}_h\mid\boldsymbol{\tau}_h:(o_{h,m},a_{h,m},\{a_{h,r}\}_{r\in\mathsf{pa}(m)})=(o,a,\{a_r\}_{r\in\mathsf{pa}(m)})\right\}\right|.$$

Then we have two observations about the $x,w$ sequence:

The vector sequence $\{x_{t,i}\}_{i=1}^{N}$ satisfies $\sum_{i=1}^{N}\|x_{t,i}\|_1\le 1$ for all $t$ because

$$\pi_m^t(\tau_{h,m})\left\|(\mathbb{O}_{h,m}^\star)^\dagger\left[\prod_{h'=0}^{h}\mathbb{M}_{h',m}^\star\right]\right\|_1\cdot\left[\prod_{l=1}^{m-1}\left\|\prod_{h'=0}^{h}\mathbb{M}_{h',l}^\star\right\|_1\cdot\pi_l^t(\tau_{h,l})\right]\cdot\left[\prod_{j=m+1}^{n}\left\|\prod_{h'=0}^{h}\mathbb{M}_{h',j}^t\right\|_1\cdot\pi_j^t(\tau_{h,j})\right]$$

$$\le\sum_{\boldsymbol{\tau}_h:(o_{h,m},a_{h,m})=(o,a)}\pi_m^t(\tau_{h,m})\left\|(\mathbb{O}_{h,m}^\star)^\dagger\left[\prod_{h'=0}^{h}\mathbb{M}_{h',m}^\star\right]\right\|_1\left[\prod_{l=1}^{m-1}\left\|\prod_{h'=0}^{h}\mathbb{M}_{h',l}^\star\right\|_1\pi_l^t(\tau_{h,l})\right]\left[\prod_{j=m+1}^{n}\left\|\prod_{h'=0}^{h}\mathbb{M}_{h',j}^t\right\|_1\pi_j^t(\tau_{h,j})\right]$$

$$\le\sum_{\{\tau_{h,r}\}_{r\ne m},\tau_{h-1,m}}\pi_m^t(\tau_{h,m})\left\|(\mathbb{O}_{h,m}^\star)^\dagger\left[\prod_{h'=0}^{h}\mathbb{M}_{h',m}^\star\right]\right\|_1\cdot\left[\prod_{l=1}^{m-1}\left\|\prod_{h'=0}^{h}\mathbb{M}_{h',l}^\star\right\|_1\cdot\pi_l^t(\tau_{h,l})\right]$$

$$\cdot\left[\prod_{j=m+1}^{n}\left\|\prod_{h'=0}^{h}\mathbb{M}_{h',j}^t\right\|_1\cdot\pi_j^t(\tau_{h,j})\right]=1.$$

The vectors $\{w_{t,l}\}_{l=1}^{O}$ satisfy $\sum_{l=1}^{O}\|w_{t,l}\|_1 \leq 2S^{1.5}/\alpha$ for all $t$, since we have

$$\sum_{l=1}^{O}\|w_{t,l}\|_1 = \left\|\left(\mathbb{M}_{h,m}^t - \mathbb{M}_{h,m}^\star\right)\mathbb{O}_{h,m}^\star\right\|_1 \leq S\left(\left\|\mathbb{M}_{h,m}^t\right\|_1 + \left\|\mathbb{M}_{h,m}^\star\right\|_1\right) \leq 2S^{1.5}/\alpha.$$

Using the notation of $\{x_{t,i}\}_{i=1}^{n}$ and $\{w_{t,l}\}_{l=1}^{O}$, we have

$$\sum_{t=1}^{k-1}\sum_{l=1}^{O}\sum_{i=1}^{n}|w_{k,l}^T x_{t,i}| = \mathcal{O}\left(\frac{\sqrt{S}}{\alpha}\sqrt{k\beta_m}\right).$$

Therefore, we can bind the target term with Eluder-Dimension lemma (Proposition 22 of Liu et al. (2022a)). We have the following result.

$$\sum_{t=1}^{k}\sum_{l=1}^{O}\sum_{i=1}^{n}|w_{t,l}^T x_{t,i}| = \mathcal{O}\left(\frac{S^{1.5}H^2}{\alpha}\sqrt{k\beta_m}\right).$$

The equation holds for all $k \in [T]$. We represent the result with matrix operator, and we arrive at

$$\sum_{t=1}^{k}\sum_{h=1}^{H-1}\sum_{\boldsymbol{\tau}_h \in \mathcal{C}_m}\pi_m^t(\tau_{h,m})\left\|\left(\mathbb{M}_{h,m}^k - \mathbb{M}_{h,m}^\star\right)\left[\prod_{h'=0}^{h}\mathbb{M}_{h',m}^\star\right]\right\|_1 \cdot \left[\prod_{l=1}^{m-1}\left\|\prod_{h'=0}^{h}\mathbb{M}_{h',l}^\star\right\|_1 \cdot \pi_l^t(\tau_{h,l})\right]$$

$$\cdot \left[\prod_{j=m+1}^{n}\left\|\prod_{h'=0}^{h}\mathbb{M}_{h',j}^t\right\|_1 \cdot \pi_j^t(\tau_{h,j})\right] \lesssim \sqrt{S^3H^4 k\beta_m}/\alpha$$

We sum up both the left-hand side and the right-hand side for all $\left(o, a, \{a_r\}_{r\in\mathsf{pa}(m)}\right)$, and we can obtain that

$$\sum_{t=1}^{k}\sum_{h=1}^{H-1}\sum_{\boldsymbol{\tau}_h}\pi_m^t(\tau_{h,m})\left\|\left(\mathbb{M}_{h,m}^k - \mathbb{M}_{h,m}^\star\right)\left[\prod_{h'=0}^{h}\mathbb{M}_{h',m}^\star\right]\right\|_1 \cdot \left[\prod_{l=1}^{m-1}\left\|\prod_{h'=0}^{h}\mathbb{M}_{h',l}^\star\right\|_1 \cdot \pi_l^t(\tau_{h,l})\right]$$

$$\cdot \left[\prod_{j=m+1}^{n}\left\|\prod_{h'=0}^{h}\mathbb{M}_{h',j}^t\right\|_1 \cdot \pi_j^t(\tau_{h,j})\right] \lesssim \frac{S^{1.5}H^2OA^{\mathsf{pa}(m)+1}}{\alpha}\sqrt{k\beta_m}.$$

$\square$

## C.2 PROOF FOR THEOREM 4.2

**Theorem C.2.** *For both randomized and deterministic algorithms, there exists an instance of DEC-POMDP with factorization such that the regret is at least $\mathcal{O}(\sqrt{A^{ind(G)+1}T})$.*

*Proof.* We consider the scenario where the state is directly observable to the agents, namely, a DEC-MDP under a factored structure model. We further assume that the transition probability satisfies the following structure: for all $\mathbf{s}', \mathbf{s} \in \mathcal{S}$, $\mathbf{a} \in \mathcal{A}$, and $h \in [H]$,

$$\mathbb{T}_{h,\mathbf{a}}(\mathbf{s}' \mid \mathbf{s}) = \left[\prod_{m=1}^{n-1}\mathbb{T}_{h,m}(s_m' \mid s_m, a_m)\right]\mathbb{T}_{h,n}(s_n' \mid s_n, \{a_r\}_{r\in[n]}).$$

We further assume that the episode length $H = 2$, and the reward function satisfies $r_{h,m}(s_m) = 0$ for all $h \in [H]$ and $m \in [n-1]$. Thus, the entire model is equivalent to an $A^n$-armed bandit problem. By leveraging a classic result on the lower bound of regret for the multi-armed bandit problem (Mannor and Tsitsiklis, 2004), it follows that for any randomized or deterministic algorithm, there exists an instance of the multi-armed bandit problem such that the regret is at least $\mathcal{O}(\sqrt{\tilde{A}T})$, where $\tilde{A}$ denotes the number of arms. Consequently, for any randomized or deterministic algorithm, there exists an instance of a factored DEC-POMDP such that the regret for achieving the global optimum is at least $\mathcal{O}(\sqrt{A^nT}) = \mathcal{O}(\sqrt{A^{\text{ind}(G)+1}T})$. $\square$

# D  SUPPLEMENTARY DETAILS FOR SECTION 4.3

## D.1  COMPLETE ALGORITHM FOR ACHIEVING LOCAL OPTIMAL UNDER FACTORED STRUCTURE MODEL

We present Algorithm D.1 as the complete algorithm for achieving local optimal under factored structure model.

---

**Algorithm 4** NASH-CA for Achieving Local Optimal Under Factored Structure Model

---

1: **Initialize** $\pi = \{\pi_i\}_{i\in[n]}$, where $\pi_i = \{\pi_{h,i}\}_{(h,i)\in[H]\times[m]}$.
2: **while** true **do**
3:     Execute policy $\pi$ for $N = \frac{CH^2}{\epsilon^2} \log\left((nHSK \max_{i\in[n]} A_i)/(\epsilon\delta)\right)$ episodes and obtain $\hat{V}_{1,i}(\pi)$ which is the empirical average of the total return under policy $\pi$.
4:     **for** agent $i = 1, \ldots, m$ **do**
5:         Appoint agent $i$ as the central agent and fix $\pi_{-i}$ to run Algorithm **??** for $K_i = \tilde{\mathcal{O}}(S^4 A^{2\cdot\mathsf{ind}(G[\mathsf{ch}(i)\cup\{i\}])}(S^2 A^{\mathsf{ind}(G)} + SO) \cdot \mathsf{poly}(H)/(\alpha^4\epsilon^2))$ episodes and get a new policy $\hat{\pi}_i$.
6:         Execute policy $(\hat{\pi}_i, \pi_{-i})$ for $N = \frac{CH^2}{\epsilon^2} \log\left((nHSK \max_{i\in[n]} A_i)/(\epsilon\delta)\right)$ episodes and obtain $\hat{V}(\hat{\pi}_i, \pi_{-i})$ which is the empirical average of the total return under policy $(\hat{\pi}_i, \pi_{-i})$.
7:         Set $\Delta_i \leftarrow \hat{V}(\hat{\pi}_i, \pi_{-i}) - \hat{V}(\pi)$.
8:     **if** $\max_{i\in[n]} \Delta_i > \epsilon/2$ **then**
9:         Update $\pi_j \leftarrow \hat{\pi}_j$ where $j = \arg\max_{i\in[n]} \Delta_i$.
10:    **else**
11:        **return** $\pi$

---

**Algorithm 5** OMLE for Achieving Local Optimal Under Factored Model

---

1: **Initialize**: $\mathcal{B}_m^1 = \{\hat{\theta}_m \in \Theta_m : \min_h \sigma_S(\mathbb{O}_m(\hat{\theta}_m)) \geq \alpha\}$, $\mathcal{D}_m = \{\}$ for all $m \in \overline{\mathsf{ch}}(i)$, $\tilde{\mathcal{B}}^1 = \{\theta_{m\in\mathsf{nch}(i)} : \min_h \sigma_S(\mathbb{O}_m(\theta_m)) \geq \alpha, \forall m \notin \overline{\mathsf{ch}}(i)\}$, $\tilde{\mathcal{D}} = \{\}$, central agent $i$, policy of other agent $\pi_{-i}$.
2: **for** $k = 1 \ldots T$ **do**
3:     Follow $\pi_{\mathsf{nch}(i)}$ to collect trajectories $\tau_{\overline{\mathsf{ch}}(i)}^k = \{o_{1,m}^k, \ldots, a_{H,m}^k\}_{m\in\mathsf{nch}(i)}$.
4:     Add $\tau_{\mathsf{nch}(i)}^k$ into $\tilde{\mathcal{D}}$ and update confidence interval with eq. (4).
5: **for** $k = 1 \ldots T$ **do**
6:     compute $(\boldsymbol{\theta}^k, \pi_i^k) = \arg\max_{\{\hat{\theta}_m \in \mathcal{B}_m^k\}_{m\in\mathsf{Chl}(i)}, \tilde{\boldsymbol{\theta}}_i \in \tilde{\Theta}_i, \mu_i} V^{\mu_i, \pi_{-i}}(\hat{\boldsymbol{\theta}})$
7:     **for** $m = 1, 2, \ldots, r$ **do**
8:         **for** $h = 1, \ldots, H$ **do**
9:             Agent $l \in \mathsf{nch}(i)$ take action $a_{h,l}^T$.
10:            Select an action $a_{h,l_j}^k \sim \pi_{h,l_j}(\cdot \mid \tau_{h-1,l_j}, o_{h,l_j})$ for all $j \in [r-1]$.
11:            Select an action $a_{h,i}^k \sim \pi_{h,i}^k(\cdot \mid \tau_{h-1,i}, o_{h,i})$.
12:            For agent $l_j$ with $j \in [m]$, collect observation $o_{h+1,l_j}^k$ from the environment.
13:            For $j \in [r] \setminus [m]$, sample dummy state $s_{h+1,l_j} \sim \mathbb{T}_{h,l_j}^k(\cdot \mid s_{h,l_j}, a_{h,\overline{\mathsf{pa}}(l_j)})$.
14:            Collect observation $o_{h+1,l_j}^k \sim \mathbb{O}_{h+1,l_j}^k(\cdot \mid s_{h+1,l_j})$ for $j \in [r] \setminus [m]$.
15:        If $m \neq r$, add $(\pi_m, \tau_{\overline{\mathsf{pa}}(m)\cap\overline{\mathsf{ch}}(i)}^k, \tau_{\overline{\mathsf{pa}}(m)\setminus\mathsf{ch}(i)}^T)$ to $\mathcal{D}_m$ for $m \neq r$.
16:        Otherwise, add $(\pi_i^k, \tau_{\overline{\mathsf{pa}}(i)\cap\overline{\mathsf{ch}}(i)}^k, \tau_{\overline{\mathsf{pa}}(i)\setminus\mathsf{ch}(i)}^T)$ to $\mathcal{D}_i$.
17:    Update confidence interval with eq. (5) for all $m \in \overline{\mathsf{pa}}(i)$.
18: **Output** $\hat{\pi}$ as uniform mixture of the policies $\pi_i^1, \pi_i^2, \ldots, \pi_i^K$.

---

**Bonums Term** : For $m \in [n]$, the bonus term $\beta_m$ and $\tilde{\beta}$ is defined as

$$\beta_m = c(H^2(S^2 A^{|\mathsf{pa}(m)|+1} + SO)\log(TSAOH) + \log(Tn/\delta)),$$

$$\tilde{\beta} = c\left(\left(\sum_{r \notin \overline{\mathsf{ch}}(n)} H(S^2 A^{|\mathsf{pa}(r)|+1} + SO)\log(TSAOH)\right) + \log(Tn/\delta)\right). \tag{15}$$

## D.2 PROOF FOR THEOREM 4.3

We first proof the following Theorem D.1, and then we will prove Theorem 4.3.

**Theorem D.1.** *If Algorithm 5.1 take agent $i$ as central agent and take policies $\{\pi_m\}_{m \in [n]/\{i\}}$ as input, then Algorithm 5.1 guarantees that with probability at least $1 - \delta$ for all $k \in [T]$*

$$\left[\sum_{t=1}^{k} V^{\pi_i^*, \boldsymbol{\pi}_{-i}} - V^{\pi_i^k, \boldsymbol{\pi}_{-i}}\right] \leq \tilde{\mathcal{O}}\left(\frac{S^2 O A^{d_i+1}}{\alpha^2}\sqrt{K\left(S^2 A^{d+1} + SO\right) \cdot poly(H)}\right),$$

*where $\pi_i^*$ is defined as $\pi_i^* = \arg\max_{\pi_i} V^{\pi_i, \boldsymbol{\pi}_{-i}}$, and recall that we use $d_i$ to denote the maximum indegree of the subgraph induced by $\overline{\mathsf{ch}}(i)$*

### D.2.1 PROOF FOR THEOREM D.1

Without loss of generosity, we consider the case where the central agent is agent $n$.

**Lemma D.1.** *There exists an absolute constant $c$ such that for any $\delta \in (0, 1]$, with probability at least $1 - \delta$: the following inequality holds true.*

$$\max_{(\theta_m, t) \in \Theta_m \times [T]} \sum_{i=1}^{t} \log\left(\frac{\mathbb{P}_{\theta_m}^{\pi_m}\left(\tau_m^i \mid \{\tau_r^i\}_{r \in \mathsf{pa}(m) \cap \overline{\mathsf{ch}}(n)}, \{\tau_r^T\}_{r \in \mathsf{pa}(m)}^{r \notin \overline{\mathsf{ch}}(n)}\right)}{\mathbb{P}_{\theta_m^*}^{\pi_m}\left(\tau_m^i \mid \{\tau_r^i\}_{r \in \mathsf{pa}(m) \cap \overline{\mathsf{ch}}(n)}, \{\tau_r^T\}_{r \in \mathsf{pa}(m)}^{r \notin \overline{\mathsf{ch}}(n)}\right)}\right) < \beta_m, m \in \overline{\mathsf{ch}}(n),$$

$$\max_{(\theta_n, t) \in \Theta_n \times [T]} \sum_{i=1}^{t} \log\left(\frac{\mathbb{P}_{\theta_n}^{\pi_n^i}\left(\tau_n^i \mid \{\tau_r^i\}_{r \in \mathsf{pa}(n) \cap \overline{\mathsf{ch}}(n)}, \{\tau_r^T\}_{r \in \mathsf{pa}(n)}^{r \notin \overline{\mathsf{ch}}(n)}\right)}{\mathbb{P}_{\theta_n^*}^{\pi_n^i}\left(\tau_n^i \mid \{\tau_r^i\}_{r \in \mathsf{pa}(n) \cap \overline{\mathsf{ch}}(n)}, \{\tau_r^T\}_{r \in \mathsf{pa}(n)}^{r \notin \overline{\mathsf{ch}}(n)}\right)}\right) < \beta_n,$$

$$\max_{(\times_{j \notin \overline{\mathsf{ch}}(n)} \theta_j, t) \in \times_{j \notin \overline{\mathsf{ch}}(n)} \Theta_j \times [T]} \sum_{i=1}^{t} \log\left(\frac{\prod_{j \notin \overline{\mathsf{ch}}(n)} \mathbb{P}_{\theta_j}^{\pi_j}(\tau_j^i \mid \{\tau_r^i\}_{r \in \mathsf{pa}(n)})}{\prod_{j \notin \overline{\mathsf{ch}}(n)} \mathbb{P}_{\theta_j^*}^{\pi_j}(\tau_j^i \mid \{\tau_r^i\}_{r \in \mathsf{pa}(n)})}\right) < \tilde{\beta}.$$

*Proof.* The proof of the lemma is similar to the proof for Lemma D.1, so we omit it here for clarity. □

**Lemma D.2.** *There exists a universal constant $c$ such that for any $\delta \in (0, 1]$, with probability at least for all $t \in [T]$ and all $\theta_m \in \Theta_m$, $m \in |\overline{\mathsf{ch}}(n)| - 1$, the following inequalities hold true. Initially, for agent $m \in [|ch(n)|]$, we have*

$$\sum_{i=1}^{t}\left(\sum_{\{\tau_r\}_{r \in \overline{\mathsf{ch}}(n)}} \left|\mathbb{P}_{\theta_{c_{n,m}}}^{\pi_{c_{n,m}}} - \mathbb{P}_{\theta_{c_{n,m}}^*}^{\pi_{c_{n,m}}}\right|\left[\prod_{l=1}^{m-1} \mathbb{P}_{\theta_{c_{n,l}^*}}^{\pi_{c_{n,l}}}\right]\left[\prod_{j=m+1}^{|\overline{\mathsf{ch}}(n)|-1} \mathbb{P}_{\theta_{c_{n,j}}}^{\pi_{c_{n,j}}}\right]\mathbb{P}_{\theta_n^t}^{\pi_n^t}\right)^2$$

$$\leq c\left(\sum_{i=1}^{t} \log\left(\frac{\mathbb{P}_{\theta_{c_{n,m}}^*}^{\pi_{c_{n,m}}}\left(\tau_{c_{n,m}}^i \mid \{\tau_r^i\}_{r \in \mathsf{pa}(c_{n,m}) \cap \overline{\mathsf{ch}}(n)}, \{\tau_r^T\}_{r \in \mathsf{pa}(c_{n,m})}^{r \notin \overline{\mathsf{ch}}(n)}\right)}{\mathbb{P}_{\theta_{c_{n,m}}}^{\pi_{c_{n,m}}}\left(\tau_{c_{n,m}}^i \mid \{\tau_r^i\}_{r \in \mathsf{pa}(c_{n,m}) \cap \overline{\mathsf{ch}}(n)}, \{\tau_r^T\}_{r \in \mathsf{pa}(c_{n,m})}^{r \notin \overline{\mathsf{ch}}(n)}\right)}\right) + \beta_{c_{n,m}}\right).$$

*For central agent $n$, we have*

$$\sum_{i=1}^{t}\left(\sum_{\{\tau_r\}_{r \in \overline{\mathsf{ch}}(n)}} \left|\mathbb{P}_{\theta_n}^{\pi_n^i} - \mathbb{P}_{\theta_n^*}^{\pi_n^i}\right|\left[\prod_{l \in \mathsf{ch}(n)} \mathbb{P}_{\theta_l^*}^{\pi_l}\right]\right)^2$$

$$\leq c\left(\sum_{i=1}^{t} \log\left(\frac{\mathbb{P}_{\theta_n^*}^{\pi_n^i}\left(\tau_n^i \mid \{\tau_r^i\}_{r \in \mathsf{pa}(n) \cap \overline{\mathsf{ch}}(n)}, \{\tau_r^T\}_{r \in \mathsf{pa}(n)}^{r \notin \overline{\mathsf{ch}}(n)}\right)}{\mathbb{P}_{\theta_n}^{\pi_n^i}\left(\tau_n^i \mid \{\tau_r^i\}_{r \in \mathsf{pa}(n) \cap \overline{\mathsf{ch}}(n)}, \{\tau_r^T\}_{r \in \mathsf{pa}(n)}^{r \notin \overline{\mathsf{ch}}(n)}\right)}\right) + \beta_n\right).$$

*For estimation error of $\{\theta_l\}_{l\notin\overline{\mathsf{ch}}(n)}$, we have*

$$\sum_{i=1}^{t}\left(\sum_{\{\tau_r\}_{r\notin\overline{\mathsf{ch}}(n)}}\left|\prod_{l\notin\overline{\mathsf{ch}}(n)}\mathbb{P}_{\theta_l}^{\pi_l}\left(\tau_l\mid\{\tau_r\}_{r\in\mathsf{pa}(l)}\right)-\prod_{l\notin\overline{\mathsf{ch}}(n)}\mathbb{P}_{\theta_l^*}^{\pi_l}\left(\tau_l\mid\{\tau_r\}_{r\in\mathsf{pa}(l)}\right)\right|\right)^2$$

$$\leq c\left(\sum_{i=1}^{t}\log\left(\frac{\prod_{l\notin\overline{\mathsf{ch}}(n)}\mathbb{P}_{\theta_l^*}^{\pi_l}\left(\tau_l^i\mid\{\tau_r^i\}_{r\in\mathsf{pa}(l)}\right)}{\prod_{l\notin\overline{\mathsf{ch}}(n)}\mathbb{P}_{\theta_l}^{\pi_l}\left(\tau_l^i\mid\{\tau_r^i\}_{r\in\mathsf{pa}(l)}\right)}\right)+\tilde{\beta}\right),$$

*where we for all $m\in|\overline{\mathsf{ch}}(n)|-1$, $\theta_m\in\Theta_m$, and policy $\pi_m$, we denote $\mathbb{P}_{\theta_m}^{\pi_m}$ as $\mathbb{P}_{\theta_m}^{\pi_m}=$*
$\mathbb{P}_{\theta_m}^{\pi_m}\left(\tau_m\mid\{\tau_r\}_{r\in\mathsf{pa}(m)\cap\overline{\mathsf{ch}}(n)},\{\tau_r^T\}_{r\in\mathsf{pa}(m)}^{r\notin\overline{\mathsf{ch}}(n)}\right).$

*Proof.* The proof of the lemma is similar to the proof for Lemma C.2, so we omit it here for clarity.
□

**Definition D.1.** *For all $(m,h,k)\in[n]\times[H]\times[T]$, we define the matrix notations $\mathbb{M}_{h,m}^{\star}\in\mathbb{R}^{O\times O}$ and $\mathbb{M}_{h,m}^{k}\in\mathbb{R}^{O\times O}$ as follows:*

$\mathbb{M}_{0,m}^{\star}=\mathbb{O}_{1,m}^{\star}\mu_m^{\star}\in\mathbb{R}^{O},\quad\mathbb{M}_{0,m}^{k}=\mathbb{O}_{1,m}^{k}\mu_m^{k}\in\mathbb{R}^{O},$

$\mathbb{M}_{h,m}^{\star}(o_m,a_m,\{a_r\}_{r\in\mathsf{pa}(m)})=\mathbb{O}_{h+1,m}^{\star}\mathbb{T}_{h,m,a_m,\{a_r\}_{r\in\mathsf{pa}(m)}}^{\star}\cdot diag\left(\mathbb{O}_{h,m}^{\star}(o_m\mid\cdot)\right)\left(\mathbb{O}_{h,m}^{\star}\right)^{\dagger}\in\mathbb{R}^{O\times O},$

$\mathbb{M}_{h,m}^{k}(o_m,a_m,\{a_r\}_{r\in\mathsf{pa}(m)})=\mathbb{O}_{h+1,m}^{k}\mathbb{T}_{h,m,a_m,\{a_r\}_{r\in\mathsf{pa}(m)}}^{k}\cdot diag\left(\mathbb{O}_{h,m}^{k}(o_m\mid\cdot)\right)\left(\mathbb{O}_{h,m}^{k}\right)^{\dagger}\in\mathbb{R}^{O\times O},$

*where $\{\mathbb{O}_{h,m}^{\star}\}_{(h,m)\in[H]\times[n]}$ and $\{\mathbb{T}_{h,m,a_m,\{a_r\}_{r\in\mathsf{pa}(m)}}^{\star}\}_{(h,m)\in[H]\times[n]}$ denote the observation and transition matrices corresponding to the true transition model, and $\{\mathbb{O}_{h,m}^{k}\}_{(h,m)\in[H]\times[n]}$ and $\{\mathbb{T}_{h,m,a_m,\{a_r\}_{r\in\mathsf{pa}(m)}}^{k}\}_{(h,m)\in[H]\times[n]}$ denote the observation and transition matrices corresponding to model parameter $\theta_k$ for all $k\in[T]$. When no confusion arises, we simplify the notation by using $\mathbb{M}_{h,m}^{\star}(\{a_{h,r}\}_{r\in\mathsf{pa}(m)}^{r\notin\overline{\mathsf{ch}}(n)})$ to represent $\mathbb{M}_{h,m}^{\star}(o_m,a_m,\{a_r\}_{r\in\mathsf{pa}(m)})$ and $\mathbb{M}_{h,m}^{k}(\{a_{h,r}\}_{r\in\mathsf{pa}(m)}^{r\notin\overline{\mathsf{ch}}(n)})$ to represent $\mathbb{M}_{h,m}^{k}(o_m,a_m,\{a_r\}_{r\in\mathsf{pa}(m)})$.*

According to the Definition D.1, we can directly achieve the following result:

**Lemma D.3.** *With probability at least $1-\delta$, for all $(k,h,m)\in[K]\times[H-1]\times|\overline{\mathsf{ch}}(n)|-1$, the following probability holds true.*

$$\sum_{t=1}^{k-1}\sum_{\{\tau_r\}_{r\in\overline{\mathsf{ch}}(n)}}\left\|\left(\mathbb{M}_{h,c_{n,m}}^{k}\left(\{a_{h,r}^T\}_{r\in\mathsf{pa}(c_{n,m})}^{r\notin\overline{\mathsf{ch}}(n)}\right)-\mathbb{M}_{h,c_{n,m}}^{\star}\left(\{a_{h,r}^T\}_{r\in\mathsf{pa}(c_{n,m})}^{r\notin\overline{\mathsf{ch}}(n)}\right)\right)\left[\prod_{h'=0}^{h-1}\mathbb{M}_{h',c_{n,m}}^{\star}\left(\{a_{h',r}^T\}_{r\in\mathsf{pa}(c_{n,m})}^{r\notin\overline{\mathsf{ch}}(n)}\right)\right]\right\|_1$$

$$\cdot\pi_{c_{n,m}}(\tau_{h,c_{n,m}})\left[\prod_{j=m+1}^{n-1}\left\|\mathbf{m}_{h,c_{n,j}}^t\right\|_1\pi_{c_{n,j}}(\tau_{h,c_{n,j}})\right]\left[\prod_{j=1}^{m-1}\left\|\mathbf{m}_{h,c_{n,j}}^t\right\|_1\pi_{c_{n,j}}(\tau_{h,c_{n,j}})\right]\|\mathbf{m}_{h,n}\|_1\pi_n^t(\tau_{h,n})\lesssim\frac{\sqrt{Sk\beta_{c_{n,m}}}}{\alpha}$$

$$\sum_{t=1}^{k-1}\sum_{\{\tau_r\}_{r\in\overline{\mathsf{ch}}(n)}}\pi_n^t(\tau_{h,n})\left\|\left(\mathbb{M}_{h,n}^{k}\left(\{a_{h,r}^T\}_{r\in\mathsf{pa}(n)}^{r\notin\overline{\mathsf{ch}}(n)}\right)-\mathbb{M}_{h,n}^{\star}\left(\{a_{h,r}^T\}_{r\in\mathsf{pa}(n)}^{r\notin\overline{\mathsf{ch}}(n)}\right)\right)\mathbf{m}_{h-1,n}\right\|_1$$

$$\cdot\left[\prod_{l=1}^{n-1}\|\mathbf{m}_{h,c_{n,l}}\|_1\pi_{c_{n,l}}(\tau_{h,c_{n,l}})\right]\lesssim\sqrt{Sk\beta_n}/\alpha$$

$$\sum_{t=1}^{k-1}\sum_{\{\tau_r\}_{r\notin\overline{\mathsf{ch}}(n)}}\left|\prod_{l\notin\overline{\mathsf{ch}}(n)}\mathbb{P}_{\theta_l^k}^{\pi_l}\left(\tau_l\mid\{\tau_r\}_{r\in\mathsf{pa}(l)}\right)-\prod_{l\notin\overline{\mathsf{ch}}(n)}\mathbb{P}_{\theta_l^*}^{\pi_l}\left(\tau_l\mid\{\tau_r\}_{r\in\mathsf{pa}(l)}\right)\right|=\mathcal{O}\left(\sqrt{k\tilde{\beta}}\right).$$

*where for all $m\in\overline{\mathsf{ch}}(n)$, $h\in[H]$, $t\in[T]$, we define $\mathbf{m}_{h,m}$ and $\mathbf{m}_{h,m}^t$ as*

$$\mathbf{m}_{h,m}=\prod_{h'=0}^{h}\mathbb{M}_{h,m}^{\star}\left(\{a_{h,r}^T\}_{r\in\mathsf{pa}(m)}^{r\notin\overline{\mathsf{ch}}(n)}\right),\quad\mathbf{m}_{h,m}^t=\prod_{h'=0}^{h}\mathbb{M}_{h,m}^t\left(\{a_{h,r}^T\}_{r\in\mathsf{pa}(m)}^{r\notin\overline{\mathsf{ch}}(n)}\right).$$

According to the definition of regret, we can bound the regret with trajectory probability in the following way:

**Lemma D.4.** *The regret is bounded by the following inequality:*

$$Regret(k) \leq nH \sum_{t=1}^{k} \sum_{\boldsymbol{\tau}_H} \left| \mathbb{P}_{\boldsymbol{\theta}^t}^{\boldsymbol{\pi}_{-n}, \pi_n^t}(\boldsymbol{\tau}_H) - \mathbb{P}_{\boldsymbol{\theta}^*}^{\boldsymbol{\pi}_{-n}, \pi_n^t}(\boldsymbol{\tau}_H) \right|,$$

*where we define*

$$\mathbb{P}_{\boldsymbol{\theta}^t}^{\boldsymbol{\pi}_{-n}, \pi_n^t}(\boldsymbol{\tau}_H) = \left[ \prod_{m=1}^{n-1} \mathbb{P}_{\theta_m^t}^{\pi_m}(\tau_{H,m} \mid \{\tau_{H,r}\}_{r \in \mathsf{pa}(m)}) \right] \mathbb{P}_{\theta_n^t}^{\pi_n^t}\left(\tau_{H,n} \mid \{\tau_{H,r}\}_{r \in \mathsf{pa}(n)}\right),$$

$$\mathbb{P}_{\boldsymbol{\theta}^*}^{\boldsymbol{\pi}_{-n}, \pi_n^t}(\boldsymbol{\tau}_H) = \left[ \prod_{m=1}^{n-1} \mathbb{P}_{\theta_m^*}^{\pi_m}(\tau_{H,m} \mid \{\tau_{H,r}\}_{r \in \mathsf{pa}(m)}) \right] \mathbb{P}_{\theta_n^*}^{\pi_n^t}\left(\tau_{H,n} \mid \{\tau_{H,r}\}_{r \in \mathsf{pa}(n)}\right).$$

**Lemma D.5.** *The regret is bounded by the following inequality:*

$$Regret(k) \leq nH \sum_{t=1}^{k} \sum_{\boldsymbol{\tau}_H} \left| \left[ \prod_{l \in \overline{\mathsf{ch}}(n)} \mathbb{P}_{\theta_l^t}^{\pi_l} \right] \mathbb{P}_{\theta_n^t}^{\pi_n^t} - \left[ \prod_{l \in ch(n)} \mathbb{P}_{\theta_l^*}^{\pi_l} \right] \mathbb{P}_{\theta_n^*}^{\pi_n^t} \right| \left[ \prod_{l \notin \mathsf{ch}(n)} \mathbb{P}_{\theta_l^*}^{\pi_l} \right]$$

$$+ nH \sum_{t=1}^{k} \sum_{\{\tau_{H,r}\}_{r \notin \overline{\mathsf{ch}}(n)}} \left| \prod_{l \notin \overline{\mathsf{ch}}(n)} \mathbb{P}_{\theta_l^t}^{\pi_l}\left(\tau_{H,l} \mid \{\tau_{H,r}\}_{r \in \mathsf{pa}(l)}\right) - \prod_{l \notin \overline{\mathsf{ch}}(n)} \mathbb{P}_{\theta_l^*}^{\pi_l}\left(\tau_{H,l} \mid \{\tau_{H,r}\}_{r \in \mathsf{pa}(l)}\right) \right|,$$

*where for any* $m \in \overline{\mathsf{ch}}(n)$, $\theta_m \in \Theta_m$, *any policy* $\pi_m$, *we denote* $\mathbb{P}_{\theta_m}^{\pi_m}$ *as* $\mathbb{P}_{\theta_m}^{\pi_m} = \mathbb{P}_{\theta_m}^{\pi_m}\left(\tau_{H,m} \mid \{\tau_{H,r}\}_{r \in \mathsf{pa}(m)}\right).$

*Proof.* According to the factorization of trajectory probability, we can bound $\sum_{\boldsymbol{\tau}_H} \left| \mathbb{P}_{\boldsymbol{\theta}^t}^{\boldsymbol{\pi}_{-n}, \pi_n^t}(\boldsymbol{\tau}_H) - \mathbb{P}_{\boldsymbol{\theta}^*}^{\boldsymbol{\pi}_{-n}, \pi_n^t}(\boldsymbol{\tau}_H) \right|$ with the following inequalities.

$$\sum_{\boldsymbol{\tau}_H} \left| \mathbb{P}_{\boldsymbol{\theta}^t}^{\boldsymbol{\pi}_{-n}, \pi_n^t}(\boldsymbol{\tau}_H) - \mathbb{P}_{\boldsymbol{\theta}^*}^{\boldsymbol{\pi}_{-n}, \pi_n^t}(\boldsymbol{\tau}_H) \right|$$

$$\leq \sum_{\boldsymbol{\tau}_H} \left| \left[ \prod_{l \in \mathsf{ch}(n)} \mathbb{P}_{\theta_l^t, l}^{\pi_l} \right] \mathbb{P}_{\theta_n^t, n}^{\pi_n^t} - \left[ \prod_{l \in \mathsf{ch}(n)} \mathbb{P}_{\theta_l^*, l}^{\pi_l} \right] \mathbb{P}_{\theta_n^*, n}^{\pi_n^t} \right| \left[ \prod_{l \notin \overline{\mathsf{ch}}(n)} \mathbb{P}_{\theta_l^*, l}^{\pi_l} \right]$$

$$+ \sum_{t=1}^{k} \sum_{\boldsymbol{\tau}_H} \left| \prod_{l \notin \overline{\mathsf{ch}}(n)} \mathbb{P}_{\theta_l^t, l}^{\pi_l}\left(\tau_{H,l} \mid \{\tau_{H,r}\}_{r \in \mathsf{pa}(l)}\right) - \prod_{l \notin \overline{\mathsf{ch}}(n)} \mathbb{P}_{\theta_l^*, l}^{\pi_l}\left(\tau_{H,l} \mid \{\tau_{H,r}\}_{r \in \mathsf{pa}(l)}\right) \right| \left[ \prod_{l \in \mathsf{ch}(n)} \mathbb{P}_{\theta_l^t, l}^{\pi_l} \right] \mathbb{P}_{\theta_n, n}^{\pi_n^t}$$

$$\leq \sum_{t=1}^{k} \sum_{\boldsymbol{\tau}_H} \left| \left[ \prod_{l \in \mathsf{ch}(n)} \mathbb{P}_{\theta_l^t, l}^{\pi_l} \right] \mathbb{P}_{\theta_n^t, n}^{\pi_n^t} - \left[ \prod_{l \in \mathsf{ch}(n)} \mathbb{P}_{\theta_l^*, l}^{\pi_l} \right] \mathbb{P}_{\theta_n^*, n}^{\pi_n^t} \right| \left[ \prod_{l \notin \overline{\mathsf{ch}}(n)} \mathbb{P}_{\theta_l^*, l}^{\pi_l} \right]$$

$$+ \sum_{t=1}^{k} \sum_{\{\tau_{H,r}\}_{r \notin \overline{\mathsf{ch}}(n)}} \left| \prod_{l \notin \overline{\mathsf{ch}}(n)} \mathbb{P}_{\theta_l^t, l}^{\pi_l}\left(\tau_{H,l} \mid \{\tau_{H,r}\}_{r \in \mathsf{pa}(l)}\right) - \prod_{l \notin \overline{\mathsf{ch}}(n)} \mathbb{P}_{\theta_l^*, l}^{\pi_l}\left(\tau_{H,l} \mid \{\tau_{H,r}\}_{r \in \mathsf{pa}(l)}\right) \right|.$$

Thus, we finish the proof of the lemma. $\qquad\square$

For the clarity of presentation, we define the following notations:

**Definition D.2.** *We define* $R_1(k)$ *and* $R_2(k)$ *as*

$$R_1(k) = \sum_{t=1}^{k} \sum_{\boldsymbol{\tau}_H} \left| \left[ \prod_{l \in \mathsf{ch}(n)} \mathbb{P}_{\theta_l^t, l}^{\pi_l} \right] \mathbb{P}_{\theta_n^t}^{\pi_n^t} - \left[ \prod_{l \in \mathsf{ch}(n)} \mathbb{P}_{\theta_l^*}^{\pi_l} \right] \mathbb{P}_{\theta_n^*}^{\pi_n^t} \right| \left[ \prod_{l \notin \overline{\mathsf{ch}}(n)} \mathbb{P}_{\theta_l^*}^{\pi_l} \right],$$

$$R_2(k) = \sum_{t=1}^{k} \sum_{\{\tau_{H,r}\}_{r \notin \overline{\mathsf{ch}}(n)}} \left| \prod_{l \notin \overline{\mathsf{ch}}(n)} \mathbb{P}_{\theta_l^t}^{\pi_l}\left(\tau_{H,l} \mid \{\tau_{H,r}\}_{r \in \mathsf{pa}(l)}\right) - \prod_{l \notin \overline{\mathsf{ch}}(n)} \mathbb{P}_{\theta_l^*}^{\pi_l}\left(\tau_{H,l} \mid \{\tau_{H,r}\}_{r \in \mathsf{pa}(l)}\right) \right|.$$

We then have

$$\text{Regret}(k) \le H \cdot (R_1(k) + R_2(k)).$$

**Lemma D.6.** *With probability at least $1 - \delta$, we have the following bound on $R_2(k)$, for all $k \in [T]$.*

$$R_2(k) \le \mathcal{O}\left(\sqrt{k\tilde{\beta}}\right).$$

*Proof.* According to Lemma D.3, we have with probability at least $1 - \delta$ for all $k \in [T]$,

$$\sum_{t=1}^{k-1} \sum_{\{\tau_r\}_{r \notin \overline{\mathsf{ch}}(n)}} \left| \prod_{l \notin \overline{\mathsf{ch}}(n)} \mathbb{P}_{\theta_l^k}^{\pi_l}\left(\tau_l \mid \{\tau_r\}_{r \in \mathsf{pa}(l)}\right) - \prod_{l \notin \overline{\mathsf{ch}}(n)} \mathbb{P}_{\theta_l^*}^{\pi_l}\left(\tau_l \mid \{\tau_r\}_{r \in \mathsf{pa}(l)}\right) \right| = \mathcal{O}\left(\sqrt{k\tilde{\beta}}\right).$$

Then we can straightforwardly obtain that $R_2(k) \le \mathcal{O}\left(\sqrt{k\tilde{\beta}}\right)$. $\qquad\square$

**Lemma D.7.** *$R_1(k)$ is bounded by the following inequality:*

$$R_1(k) \le \left[ \sum_{t=1}^{k} \sum_{m=1}^{\mathsf{chl}(n)-1} \sum_{h=1}^{H-1} \sum_{\boldsymbol{\tau}_h} \frac{\sqrt{S}}{\alpha} \left\| \left( \mathbb{M}_{h,c_{n,m}}^t \left( \{a_{h,r}^T\}_{r \in \mathsf{pa}(c_{n,m})}^{r \notin \overline{\mathsf{ch}}(n)} \right) - \mathbb{M}_{h,c_{n,m}}^\star \left( \{a_{h,r}^T\}_{r \in \mathsf{pa}(c_{n,m})}^{r \notin \overline{\mathsf{ch}}(n)} \right) \right) \mathbf{m}_{h-1,c_{n,m}} \right\|_1 \right.$$

$$\cdot \pi_{c_{n,m}}^t(\tau_{h,c_{n,m}}) \left[ \prod_{j=m+1}^{n-1} \left\| \mathbf{m}_{h,c_{n,j}}^t \right\|_1 \pi_{c_{n,j}}^t(\tau_{h,c_{n,j}}) \right] \left[ \prod_{j=1}^{m-1} \left\| \mathbf{m}_{h,c_{n,j}}^t \right\|_1 \pi_{c_{n,j}}^t(\tau_{h,c_{n,j}}) \right] \|\mathbf{m}_{h,n}\|_1 \pi_n^t(\tau_{h,n})$$

$$\left. + \frac{\sqrt{S}}{\alpha} \left\| \left( \mathbb{M}_{h,n}^t \left( \{a_{h,r}^T\}_{r \in \mathsf{pa}(n)}^{r \notin \overline{\mathsf{ch}}(n)} \right) - \mathbb{M}_{h,n}^\star \left( \{a_{h,r}^T\}_{r \in \mathsf{pa}(n)}^{r \notin \overline{\mathsf{ch}}(n)} \right) \right) \mathbf{m}_{h-1,n} \right\|_1 \pi_n^t(\tau_{h,n}) \cdot \left[ \prod_{l=1}^{n-1} \|\mathbf{m}_{h,c_{n,l}}\|_1 \pi_{c_{n,l}}^t(\tau_{h,c_{n,l}}) \right] \right],$$

*where for all $m \in \overline{\mathsf{ch}}(n)$, $h \in [H]$, $t \in [T]$, we define $\mathbf{m}_{h,m}$ and $\mathbf{m}_{h,m}^t$ as*

$$\mathbf{m}_{h,m} = \prod_{h'=0}^{h} \mathbb{M}_{h,m}^\star \left( \{a_{h,r}^T\}_{r \in \mathsf{pa}(m)}^{r \notin \overline{\mathsf{ch}}(n)} \right), \quad \mathbf{m}_{h,m}^t = \prod_{h'=0}^{h} \mathbb{M}_{h,m}^t \left( \{a_{h,r}^T\}_{r \in \mathsf{pa}(m)}^{r \notin \overline{\mathsf{ch}}(n)} \right).$$

*Proof.* According to the definition of $R_1(k)$, we can obtain the following bound on $R_1(k)$:

$$R_1(k) \le \sum_{t=1}^{k} \sum_{\boldsymbol{\tau}_H} \sum_{m=1}^{\mathsf{chl}(n)-1} \left[ \prod_{l=1}^{m-1} \mathbb{P}_{\theta_l^*}^{\pi_l} \right] \left| \mathbb{P}_{\theta_m^t}^{\pi_m} - \mathbb{P}_{\theta_m^*}^{\pi_m} \right| \left[ \prod_{j=m+1}^{n-1} \mathbb{P}_{\theta_j^t}^{\pi_j} \right] \mathbb{P}_{\theta_n^t}^{\pi_n^t} \left[ \prod_{l \notin \overline{\mathsf{ch}}(n)} \mathbb{P}_{\theta_l^t}^{\pi_l} \right]$$

$$+ \sum_{t=1}^{k} \sum_{\boldsymbol{\tau}_H} \left[ \prod_{l=1}^{n-1} \mathbb{P}_{\theta_l^*}^{\pi_l} \right] \left| \mathbb{P}_{\theta_n^t}^{\pi_n^t} - \mathbb{P}_{\theta_n^*}^{\pi_n^t} \right| \left[ \prod_{l \notin \overline{\mathsf{ch}}(n)} \mathbb{P}_{\theta_l^*}^{\pi_l} \right].$$

We can deduce the result of lemma by representing this inequality with matrix notations. $\qquad\square$

**Lemma D.8.** *With probability at least $1 - \delta$, for all $(k, h, m) \in [K] \times [H-1] \times |\overline{\mathsf{ch}}(n)| - 1$, the following inequality holds true.*

$$\sum_{t=1}^{k-1} \sum_{\{\tau_r\}_{r \in \overline{\mathsf{ch}}(n)}} \pi_{c_{n,m}}^t(\tau_{h,c_{n,m}}) \left\| \left( \mathbb{M}_{h,c_{n,m}}^t \left( \{a_{h,r}^T\}_{r \in \mathsf{pa}(c_{n,m})}^{r \notin \overline{\mathsf{ch}}(n)} \right) - \mathbb{M}_{h,c_{n,m}}^\star \left( \{a_{h,r}^T\}_{r \in \mathsf{pa}(c_{n,m})}^{r \notin \overline{\mathsf{ch}}(n)} \right) \right) \mathbf{m}_{h-1,c_{n,m}} \right\|_1$$

$$\cdot \left[ \prod_{j=m+1}^{n-1} \left\| \mathbf{m}_{h,c_{n,j}}^t \right\|_1 \pi_{c_{n,j}}(\tau_{h,c_{n,j}}^t) \right] \left[ \prod_{j=1}^{m-1} \left\| \mathbf{m}_{h,c_{n,j}}^t \right\|_1 \pi_{c_{n,j}}^t(\tau_{h,c_{n,j}}) \right] \|\mathbf{m}_{h,n}\|_1 \pi_n^t(\tau_{h,n})$$

$$\lesssim \frac{S^{1.5} O A^{|\mathsf{pa}(c_{n,m}) \cap \overline{\mathsf{ch}}(n)|+1} H^2}{\alpha} \sqrt{k \beta_{c_{n,m}}},$$

*and for agent $n$, the following inequality holds true:*

$$\sum_{t=1}^{k-1} \sum_{\{\tau_r\}_{r\in\overline{\mathsf{ch}}(n)}} \pi_n^t(\tau_{h,n}) \left\| \left( \mathbb{M}_{h,n}^t \left( \{a_{h,r}^T\}_{r\in\mathsf{pa}(n)}^{r\notin\overline{\mathsf{ch}}(n)} \right) - \mathbb{M}_{h,n}^\star \left( \{a_{h,r}^T\}_{r\in\mathsf{pa}(n)}^{r\notin\overline{\mathsf{ch}}(n)} \right) \right) \mathbf{m}_{h-1,n} \right\|_1 \left[ \prod_{l=1}^{n-1} \|\mathbf{m}_{h,c_{n,l}}\|_1 \pi_{c_{n,l}}^t(\tau_{h,c_{n,l}}) \right]$$

$$\lesssim \frac{S^{1.5} O A^{|\mathsf{pa}(n)\cap\overline{\mathsf{ch}}(n)|+1} H^2}{\alpha} \sqrt{k\beta_n},$$

*where for all $m \in \overline{\mathsf{ch}}(n)$, $h \in [H]$, $t \in [T]$, we define $\mathbf{m}_{h,m}$ and $\mathbf{m}_{h,m}^t$ as*

$$\mathbf{m}_{h,m} = \prod_{h'=0}^{h} \mathbb{M}_{h,m}^\star \left( \{a_{h,r}^T\}_{r\in\mathsf{pa}(m)}^{r\notin\overline{\mathsf{ch}}(n)} \right), \quad \mathbf{m}_{h,m}^t = \prod_{h'=0}^{h} \mathbb{M}_{h,m}^t \left( \{a_{h,r}^T\}_{r\in\mathsf{pa}(m)}^{r\notin\overline{\mathsf{ch}}(n)} \right).$$

*Proof.* Initially, for $m \in |\overline{\mathsf{ch}}(n)| - 1$, we fix $(o, a, \{a_r\}_{r\in\mathsf{pa}(c_{n,m})\cap\overline{\mathsf{ch}}(n)}) \in \mathcal{O}_{c_{n,m}} \times \mathcal{A}_{c_{n,m}} \times \left( \times_{j\in\mathsf{pa}(c_{n,m})\cap\overline{\mathsf{ch}}(n)} \mathcal{A}_j \right)$, we first define set $\mathrm{S}_{c_{n,m}}$ as

$$\mathcal{C}_{c_{n,m}} = \left\{ \{\tau_{h,r}\}_{r\in\overline{\mathsf{ch}}(n)} \mid \{\tau_{h,r}\}_{r\in\overline{\mathsf{ch}}(n)} : \left( o_{h,c_{n,m}}, a_{h,c_{n,m}}, \{a_{h,r}\}_{r\in\mathsf{pa}(c_{n,m})\cap\overline{\mathsf{ch}}(n)} \right) = \left( o, a, \{a_r\}_{r\in\mathsf{pa}(c_{n,m})\cap\overline{\mathsf{ch}}(n)} \right) \right\}.$$

$\square$

We assume that

$$w_{t,l} = \left[ \left( \tilde{\mathbb{M}}_{h,c_{n,m}}^t - \tilde{\mathbb{M}}_{h,c_{n,m}}^\star \right) \mathbb{O}_{h,c_{n,m}}^\star \right]_l.$$

where we denote $\tilde{\mathbb{M}}_{h,c_{n,m}}^t$ and $\tilde{\mathbb{M}}_{h,c_{n,m}}^\star$ as

$$\tilde{\mathbb{M}}_{h,c_{n,m}}^t = \mathbb{M}_{h,c_{n,m}}^t \left( o, a, \{a_r\}_{r\in\mathsf{pa}(c_{n,m})\cap\overline{\mathsf{ch}}(n)}, \{a_{h,r}^T\}_{r\in\mathsf{pa}(c_{n,m})}^{r\notin\overline{\mathsf{ch}}(n)} \right),$$

$$\tilde{\mathbb{M}}_{h,c_{n,m}}^\star = \mathbb{M}_{h,c_{n,m}}^\star \left( o, a, \{a_r\}_{r\in\mathsf{pa}(c_{n,m})\cap\overline{\mathsf{ch}}(n)}, \{a_{h,r}^T\}_{r\in\mathsf{pa}(c_{n,m})}^{r\notin\overline{\mathsf{ch}}(n)} \right).$$

We denote the sequence $\pi_{c_{n,m}}^t(\tau_{h,c_{n,m}})\mathbb{O}_{h,c_{n,m}}^\dagger \mathbf{m}_{h-1,c_{n,m}} \cdot [\prod_{l=1}^{m-1}\|\mathbf{m}_{h,c_{n,l}}\|_1\pi_{c_{n,l}}(\tau_{h,c_{n,l}})] \cdot [\prod_{j=m+1}^{n-1}\|\mathbf{m}_{h,c_{n,j}}\|_1\pi_{c_{n,j}}^t(\tau_{h,c_{n,j}})] \cdot \|\mathbf{m}_{h,n}^t\|_1\pi_n^t(\tau_{h,n})$ for all $\{\tau_{h,r}\}_{r\in\overline{\mathsf{ch}}(n)} \in \mathcal{C}_{c_{n,m}}$ by $x_{t,1}, x_{t,2}, \ldots, x_{t,N}$, where $N = |\mathcal{C}_{c_{n,m}}|$. Then we have two observations about the $x, w$ sequence: The vector sequence $\{x_{t,i}\}_{i=1}^N$ satisfies $\sum_{i=1}^N \|x_{t,i}\|_1 \le 1$ for all $t$ because

$$\sum_{\{\tau_{h,r}\}_{r\in\overline{\mathsf{ch}}(n)}\in\mathcal{C}_{c_{n,m}}} \pi_{c_{n,m}}(\tau_{h,c_{n,m}}) \left\| \mathbb{O}_{h,c_{n,m}}^\dagger \mathbf{m}_{h-1,c_{n,m}} \right\|_1 \left[ \prod_{l=1}^{m-1} \|\mathbf{m}_{h,c_{n,l}}\|_1 \pi_l(\tau_{h,c_{n,l}}) \right]$$

$$\cdot \left[ \prod_{j=m+1}^{n-1} \|\mathbf{m}_{h,c_{n,j}}^t\|_1 \pi_j(\tau_{h,c_{n,j}}) \right] \|\mathbf{m}_{h,n}^t\|_1 \pi_n^t(\tau_{h,n})$$

$$\le \sum_{\{\tau_{h,r}\}_{r\in\overline{\mathsf{ch}}(n)}:(o_{h,c_{n,m}},a_{h,c_{n,m}})=(o,a)} \pi_{c_{n,m}}(\tau_{h,c_{n,m}}) \left\| \mathbb{O}_{h,c_{n,m}}^\dagger \mathbf{m}_{h-1,c_{n,m}} \right\|_1 \left[ \prod_{l=1}^{m-1} \|\mathbf{m}_{h,c_{n,l}}\|_1 \pi_l(\tau_{h,c_{n,l}}) \right]$$

$$\cdot \left[ \prod_{j=m+1}^{n-1} \|\mathbf{m}_{h,c_{n,j}}^t\|_1 \pi_j(\tau_{h,c_{n,j}}) \right] \cdot \|\mathbf{m}_{h,n}^t\|_1 \pi_n^t(\tau_{h,n})$$

$$\le \sum_{\{\tau_{h,r}\}_{r\in\overline{\mathsf{ch}}(n)/\{c_{n,m}\},\tau_{h-1,c_{n,m}}}} \pi_{c_{n,m}}(\tau_{h-1,c_{n,m}}) \left\| \mathbb{O}_{h,c_{n,m}}^\dagger \mathbf{m}_{h-1,c_{n,m}} \right\|_1 \left[ \prod_{l=1}^{m-1} \|\mathbf{m}_{h,c_{n,l}}\|_1 \pi_l(\tau_{h,c_{n,l}}) \right]$$

$$\cdot \left[ \prod_{j=m+1}^{n-1} \|\mathbf{m}_{h,c_{n,j}}^t\|_1 \pi_j(\tau_{h,c_{n,j}}) \right] = 1.$$

The vectors $\{w_{t,l}\}_{l=1}^{O}$ satisfy $\sum_{l=1}^{O}\|w_{t,l}\|_1 \leq 2S^{1.5}/\alpha$ for all $t$, since we have

$$\sum_{l=1}^{O}\|w_{t,l}\|_1 = \left\|\left(\tilde{\mathbb{M}}_{h,c_{n,m}}^{t} - \tilde{\mathbb{M}}_{h,c_{n,m}}^{\star}\right)\mathbb{O}_{h,c_{n,m}^{\star}}\right\|_1 \leq S\left(\left\|\tilde{\mathbb{M}}_{h,c_{n,m}}^{t}\right\|_1 + \left\|\tilde{\mathbb{M}}_{h,c_{n,m}}^{\star}\right\|_1\right) \leq \frac{2S^{1.5}}{\alpha}.$$

Using the notation of $\{x_{t,i}\}_{i=1}^{n}$ and $\{w_{t,l}\}_{l=1}^{O}$, we have

$$\sum_{t=1}^{k-1}\sum_{l=1}^{O}\sum_{i=1}^{n}|w_{k,l}^{T}x_{t,i}| = \mathcal{O}\left(\frac{\sqrt{S}}{\alpha}\sqrt{k\beta_{c_{n,m}}}\right).$$

Therefore, we can bind the target term with Eluder-Dimension lemma (Proposition 22 of Liu et al. (2022a)). We have the following result.

$$\sum_{t=1}^{k}\sum_{l=1}^{O}\sum_{i=1}^{n}|w_{t,l}^{T}x_{t,i}| = \mathcal{O}\left(\frac{S^{1.5}H^2}{\alpha}\sqrt{k\beta_{c_{n,m}}}\right).$$

The equation holds for all $k \in [K]$. We represent it with matrix operator, and we have

$$\sum_{t=1}^{k}\sum_{\{\tau_{h,r}\}_{r\in\overline{\mathsf{ch}}(n)}\in\mathcal{C}_{c_{n,m}}}\left\|\left(\tilde{\mathbb{M}}_{h,c_{n,m}}^{t} - \tilde{\mathbb{M}}_{h,c_{n,m}}^{\star}\right)\mathbf{m}_{h-1,c_{n,m}}\right\|_1 \cdot \pi_{c_{n,m}}^{t}(\tau_{h,c_{n,m}})$$

$$\cdot\left[\prod_{l=1}^{m-1}\|\mathbf{m}_{h,c_{n,l}}\|_1\pi_{c_{n,l}}^{t}(\tau_{h,c_{n,l}})\right]\left[\prod_{j=m+1}^{n-1}\left\|\mathbf{m}_{h,c_{n,j}}^{t}\right\|_1\pi_{c_{n,j}}^{t}(\tau_{h,c_{n,j}})\right]\|\mathbf{m}_{h,n}^{t}\|_1\pi^{t}(\tau_{h,n}) = \mathcal{O}\left(\frac{S^{1.5}H^2}{\alpha}\sqrt{k\beta_{c_{n,m}}}\right).$$

We sum up both the left-hand side and the right-hand side for all $(o, a, \{a_r\}_{r\in\mathsf{pa}(c_{n,m})\cap\overline{\mathsf{ch}}(n)}) \in \mathcal{O}_{c_{n,m}} \times \mathcal{A}_{c_{n,m}} \times \left(\times_{j\in\mathsf{pa}(c_{n,m})\cap\overline{\mathsf{ch}}(n)}\mathcal{A}_j\right)$, and we can obtain that

$$\sum_{t=1}^{k}\sum_{\boldsymbol{\tau}_h}\pi_{c_{n,m}}^{t}(\tau_{h,c_{n,m}})\left\|\left(\mathbb{M}_{h,c_{n,m}}^{t}\left(\{a_{h,r}^{T}\}_{r\in\mathsf{pa}(c_{n,m})}^{r\notin\overline{\mathsf{ch}}(n)}\right) - \mathbb{M}_{h,c_{n,m}}^{\star}\left(\{a_{h,r}^{T}\}_{r\in\mathsf{pa}(c_{n,m})}^{r\notin\overline{\mathsf{ch}}(n)}\right)\right)\mathbf{m}_{h-1,c_{n,m}}\right\|_1$$

$$\cdot\left[\prod_{l=1}^{m-1}\|\mathbf{m}_{h,l}\|_1\pi_l^{t}(\tau_{h,l})\right]\cdot\left[\prod_{j=m+1}^{n-1}\left\|\mathbf{m}_{h,j}^{t}\right\|_1\pi_j^{t}(\tau_{h,j})\right]\|\mathbf{m}_{h,m}^{t}\|_1\pi_n^{t}(\tau_{h,n}) \qquad = \mathcal{O}\left(\frac{S^{1.5}H^2OA^{1+}}{}\right.$$

Then, with similar techniques, we consider the term

$$\sum_{t=1}^{k-1}\sum_{\{\tau_r\}_{r\in\overline{\mathsf{ch}}(n)}}\pi_n^{t}(\tau_{h,n})\left\|\left(\mathbb{M}_{h,n}^{t}\left(\{a_{h,r}^{T}\}_{r\in\mathsf{pa}(n)}^{r\notin\overline{\mathsf{ch}}(n)}\right) - \mathbb{M}_{h,n}^{\star}\left(\{a_{h,r}^{T}\}_{r\in\mathsf{pa}(n)}^{r\notin\overline{\mathsf{ch}}(n)}\right)\right)\mathbf{m}_{h-1,n}\right\|_1\left[\prod_{l=1}^{n-1}\left\|\mathbf{m}_{h,c_{n,l}}\right\|_1\pi_{c_{n,l}}^{t}(\tau_{h,c_{n,l}})\right].$$

We define set $\mathrm{S}_n$ as

$$\mathcal{C}_n = \left\{\{\tau_{h,r}\}_{r\in\overline{\mathsf{ch}}(n)}\mid\{\tau_{h,r}\}_{r\in\overline{\mathsf{ch}}(n)} : \left(o_{h,n}, a_{h,n}, \{a_{h,r}\}_{r\in\mathsf{pa}(n)\cap\overline{\mathsf{ch}}(n)}\right) = \left(o, a, \{a_r\}_{r\in\mathsf{pa}(n)\cap\overline{\mathsf{ch}}(n)}\right)\right\}.$$

We assume that

$$w_{t,l} = \left[\left(\tilde{\mathbb{M}}_{h,n}^{t} - \tilde{\mathbb{M}}_{h,n}^{\star}\right)\mathbb{O}_{h,n}^{\star}\right]_l.$$

where we denote $\tilde{\mathbb{M}}_{h,n}^{t}$ and $\tilde{\mathbb{M}}_{h,n}^{\star}$ as

$$\tilde{\mathbb{M}}_{h,n}^{t} = \mathbb{M}_{h,n}^{t}\left(o, a, \{a_r\}_{r\in\mathsf{pa}(n)\cap\overline{\mathsf{ch}}(n)}, \{a_{h,r}^{T}\}_{r\in\mathsf{pa}(n)}^{r\notin\overline{\mathsf{ch}}(n)}\right),$$

$$\tilde{\mathbb{M}}_{h,n}^{\star} = \mathbb{M}_{h,n}^{\star}\left(o, a, \{a_r\}_{r\in\mathsf{pa}(n)\cap\overline{\mathsf{ch}}(n)}, \{a_{h,r}^{T}\}_{r\in\mathsf{pa}(n)}^{r\notin\overline{\mathsf{ch}}(n)}\right).$$

We denote the target sequence $\pi_n^{t}(\tau_{h,n})\mathbb{O}_{h,n}^{\dagger}\mathbf{m}_{h-1,n} \cdot [\prod_{l=1}^{n-1}\|\mathbf{m}_{h,c_{n,l}}\|_1\pi_{c_{n,l}}(\tau_{h,c_{n,l}})]$ for all $\{\tau_{h,r}\}_{r\in\overline{\mathsf{ch}}(n)} \in \mathrm{S}_n$ by $x_{t,1}, x_{t,2}, \ldots, x_{t,N}$, where $N = |\mathcal{C}_n|$. Then we have two observations

about the $x, w$ sequence: The vector sequence $\{x_{t,i}\}_{i=1}^N$ satisfies $\sum_{i=1}^N \|x_{t,i}\|_1 \leq 1$ for all $t$ because

$$\sum_{\{\tau_{h,r}\}_{r \in \overline{\mathsf{ch}}(n)} \in \mathcal{C}_n} \pi_n^t(\tau_{h,n}) \left\| \mathbb{O}_{h,n}^\dagger \mathbf{m}_{h,n} \right\|_1 \left[ \prod_{l=1}^{n-1} \|\mathbf{m}_{h,c_{n,l}}\|_1 \pi_l^t(\tau_{h,c_{n,l}}) \right]$$

$$\leq \sum_{\{\tau_{h,r}\}_{r \in \overline{\mathsf{ch}}(n)}: (o_{h,n}, a_{h,n}) = (o,a)} \pi_n^t(\tau_{h,n}) \left\| \mathbb{O}_{h,n}^\dagger \mathbf{m}_{h-1,n} \right\|_1 \left[ \prod_{l=1}^{n-1} \|\mathbf{m}_{h,c_{n,l}}\|_1 \pi_l^t(\tau_{h,c_{n,l}}) \right]$$

$$\leq \sum_{\{\tau_{h,r}\}_{r \in \mathsf{ch}(n)}, \tau_{h-1,n}} \pi_n^t(\tau_{h-1,n}) \left\| \mathbb{O}_{h,n}^\dagger \mathbf{m}_{h-1,n} \right\|_1 \left[ \prod_{l=1}^{n-1} \|\mathbf{m}_{h,c_{n,l}}\|_1 \pi_l^t(\tau_{h,c_{n,l}}) \right] = 1,$$

The vectors $\{w_{t,l}\}_{l=1}^O$ satisfy $\sum_{l=1}^O \|w_{t,l}\|_1 \leq 2S^{1.5}/\alpha$ for all $t$, since we have

$$\sum_{l=1}^O \|w_{t,l}\|_1 = \left\| \left( \tilde{\mathbb{M}}_{h,n}^t - \tilde{\mathbb{M}}_{h,n}^\star \right) \mathbb{O}_{h,n}^\star \right\|_1 \leq S \left( \left\| \tilde{\mathbb{M}}_{h,n}^t \right\|_1 + \left\| \tilde{\mathbb{M}}_{h,n}^\star \right\|_1 \right) \leq 2S^{1.5}/\alpha.$$

Using the notation of $\{x_{t,i}\}_{i=1}^n$ and $\{w_{t,l}\}_{l=1}^O$, we have

$$\sum_{t=1}^{k-1} \sum_{l=1}^O \sum_{i=1}^n |w_{k,l}^T x_{t,i}| = \mathcal{O}\left( \frac{\sqrt{S}}{\alpha} \sqrt{k\beta_n} \right).$$

Therefore, we can bind the target term with Eluder-Dimension lemma (Proposition 22 of Liu et al. (2022a)). We have the following result.

$$\sum_{t=1}^k \sum_{l=1}^O \sum_{i=1}^n |w_{t,l}^T x_{t,i}| = \mathcal{O}\left( \frac{S^{1.5}H^2}{\alpha} \sqrt{k\beta_n} \right).$$

The equation holds for all $k \in [K]$. We represent it with $B-$operator, and we have

$$\sum_{t=1}^k \sum_{\{\tau_{h,r}\}_{r \in \overline{\mathsf{ch}}(n)} \in \mathcal{C}_n} \left\| \left( \tilde{\mathbb{M}}_{h,n}^t - \tilde{\mathbb{M}}_{h,n}^\star \right) \mathbf{m}_{h-1,n} \right\|_1 \cdot \pi_n^t(\tau_{h,n}) \left[ \prod_{l=1}^{n-1} \|\mathbf{m}_{h,c_{n,l}}\|_1 \pi_{c_{n,l}}^t(\tau_{h,c_{n,l}}) \right] = \mathcal{O}\left( \frac{S^{1.5}H^2}{\alpha} \sqrt{k\beta_{c_{n,m}}} \right),$$

We sum up both the left-hand side and the right-hand side for all $(o, a, \{a_r\}_{r \in \mathsf{pa}(c_{n,m}) \cap \overline{\mathsf{ch}}(n)}) \in \mathcal{O}_{c_{n,m}} \times \mathcal{A}_{c_{n,m}} \times \left( \times_{j \in \mathsf{pa}(c_{n,m}) \cap \overline{\mathsf{ch}}(n)} \mathcal{A}_j \right)$, and we can obtain that

$$\sum_{t=1}^k \sum_{\boldsymbol{\tau}_h} \pi_n^t(\tau_{h,n}) \left\| \left( \mathbb{M}_{h,n}^t \left( \{a_{h,r}^T\}_{r \in \mathsf{pa}(n)}^{r \notin \overline{\mathsf{ch}}(n)} \right) - \mathbb{M}_{h,n}^\star \left( \{a_{h,r}^T\}_{r \in \mathsf{pa}(n)}^{r \notin \overline{\mathsf{ch}}(n)} \right) \right) \mathbf{m}_{h-1,n} \right\|_1$$

$$\cdot \left[ \prod_{l=1}^{n-1} \|\mathbf{m}_{h,l}\|_1 \pi_l^t(\tau_{h,l}) \right] = \mathcal{O}\left( \frac{S^{1.5}H^2 OA^{1+|\mathsf{pa}(c_{n,m}) \cap \overline{\mathsf{ch}}(n)|}}{\alpha} \sqrt{k\beta_{c_{n,m}}} \right).$$

**Corollary D.1.** *With probability at least $1 - \delta$, for all $k \in [T]$, $R_1(k)$ is bounded by the following inequality:*

$$R_1(k) \leq \mathcal{O}\left( \frac{S^2 H^3 OA}{\alpha^2} \left[ \sum_{m \in \overline{\mathsf{ch}}(n)} A^{|\mathsf{pa}(m) \cap \overline{\mathsf{ch}}(n)|+1} \sqrt{k\beta_m} \right] \right).$$

*Proof.* We only need to combine the result in Lemma D.3 and Lemma D.7, and we can achieve the result. $\square$

**Corollary D.2.** *With probability at least $1 - \delta$, for all $k \in [T]$, $R_1(k)$ is bounded by the following inequality:*

$$R_1(k) \leq \mathcal{O}\left( \frac{S^2 H^3 OA}{\alpha^2} \left[ \sum_{m \in \overline{\mathsf{ch}}(n)} A^{|\mathsf{pa}(m) \cap \overline{\mathsf{ch}}(n)|+1} \sqrt{k\beta_m} \right] \right).$$

*Proof.* We only need to combine the result in Lemma D.3 and Lemma D.7, and we can achieve the result. $\square$

**Proof of Theorem D.1**   With the lemmas presented above, we are now ready to prove Theorem D.1.

**Theorem D.2.** *If Algorithm 5.1 take agent $i$ as central agent and take policies $\{\pi_m\}_{m\in[n]/\{i\}}$ as input, then Algorithm 5.1 guarantees that with probability at least $1 - \delta$ for all $k \in [T]$*

$$\left[ \sum_{t=1}^k V^{\pi_i^*, \boldsymbol{\pi}_{-i}} - V^{\pi_i^k, \boldsymbol{\pi}_{-i}} \right] \leq \tilde{\mathcal{O}} \left( \frac{S^2 O A^{d_i+1}}{\alpha^2} \sqrt{K \left( S^2 A^{\mathsf{ind}(G)+1} + SO \right) \cdot poly(H)} \right),$$

*where $\pi_i^*$ is defined as $\pi_i^* = \arg\max_{\pi_i} V^{\pi_i, \boldsymbol{\pi}_{-i}}$.*

*Proof.* According to Corollary D.2 and Lemma D.6, we can obtain that with probability at least $1 - \delta$, for all $k \in [T]$,

$$R^k \leq H \cdot (R_1(k) + R_2(k))$$

$$\leq \mathcal{O} \left( \frac{S^2 H^4 O A}{\alpha^2} \left[ \sum_{m \in \overline{\mathsf{ch}}(n)} A^{|\mathsf{pa}(m) \cap \overline{\mathsf{ch}}(n)|+1} \sqrt{k\beta_m} \right] + H\sqrt{k\tilde{\beta}} \right)$$

$$\leq \tilde{\mathcal{O}} \left( \frac{S^2 O A^{\mathsf{indeg}(F_n)+1}}{\alpha^2} \sqrt{K \left( S^2 A^m + SO \right) \cdot \mathrm{poly}(H)} \right).$$

$\square$

### D.2.2   Proof for Theorem 4.3

After proving Theorem D.1, we are ready to prove Theorem 4.3. For the readers' convenience, we first restate the theorem here.

**Theorem D.3.** *If central agent for Algorithm D.1 is $i$, we define bonus parameter as $\beta_m = c(H^2(S^2 A^{|\mathsf{pa}(m)|+1} + SO)\log(TSAOH) + \log(Tn/\delta))$, $\forall m \in [n]$, $\tilde{\beta} = c((\sum_{r \notin \mathsf{ch}(i)} H(S^2 A^{\mathsf{ind}(r)+1} + SO)\log(TSAOH) + \log(Tn/\delta)))$. Then, with probability at least $1 - \delta$, Algorithm D.1 terminates within $4H/\epsilon$ steps of the while loop, and outputs an $\epsilon-$approximate local optimal policy. The total episodes of play in Algorithm D.1 is at most*

$$K = \tilde{\mathcal{O}}\big( \sum_{m=1}^n S^4 O^2 A^{2 \cdot \mathsf{ind}(G[\mathsf{ch}(m)\cup\{m\}])+2}(S^2 A^{\mathsf{ind}(G)+1} + SO) \cdot poly(H)/(\alpha^4\epsilon^3)\big).$$

*Proof.* We use superscript $t$ to represent variables at the $t^{th}$ step (before $\pi$ is updated) of the while loop. We set $K_i = \tilde{\mathcal{O}}(S^4 A^{2 \cdot \mathsf{ind}(G[\mathsf{ch}(i)\cup\{i\}])}(S^2 A^{\mathsf{ind}(G)} + SO) \cdot \mathrm{poly}(H)/(\alpha^4\epsilon^2))$. According to Theorem D.1, for fixed $i$ and $t$, we have with probability at least $1 - \frac{\delta\epsilon}{8nH}$

$$\max_{\mu_i} V(\mu_i, \pi_{-i}^t) - V(\hat{\pi}_i^t, \pi_{-i}^t) \leq \frac{\epsilon}{4}.$$

We then take a union bound over all $t \leq 4H/\epsilon$ and for all $i \in [n]$, and we have with probability at least $1 - \delta$ the following inequality holds for all $i \in [n]$ and $t \leq 4H/\epsilon$

$$\max_{\mu_i} V(\mu_i, \pi_{-i}^t) - V(\hat{\pi}_i^t, \pi_{-i}^t) \leq \frac{\epsilon}{4}. \tag{16}$$

For the empirical estimator $\hat{V}^t$, it's bounded in $[0, H]$. Thus, by Hoeffding's inequality, for fixed $i \in [n]$ and $t$

$$\mathbb{P}\left( \left| \hat{V}^t - V^t \right| \geq \frac{\epsilon}{8} \right) \leq 2\exp\left( -\frac{N\epsilon^2}{32H^2} \right).$$

Choosing $N = \frac{CH^2}{\epsilon^2} \log\left( \frac{nHSK \max_{i\in[n]} A_i}{\epsilon\delta} \right)$ for some large constant $C$, we have

$$\mathbb{P}\left( \left| \hat{V}^t - V^t \right| \geq \frac{\epsilon}{8} \right) \leq \frac{\epsilon\delta}{16nH}.$$

Applying this inequality for $\hat{V}^t(\hat{\pi}_i^t, \pi_{-i}^t)$ and $\hat{V}^t(\pi^t)$ and taking a union bound over $i \in [n]$ and $t \le 4H/\epsilon$, we can achieve that

$$|\hat{V}(\hat{\pi}_i^t, \pi_{-i}^t) - V(\hat{\pi}_i^t, \pi_{-i}^t)| \le \epsilon/8, \quad |\hat{V}^t(\pi^t) - V(\pi^t)| \le \epsilon/8.$$

We combine this result with equation 22, and we can obtain that with probability at least $1 - \delta$

$$\max_{\mu_i} V(\mu_i, \pi_{-i}^t) - V(\hat{\pi}_i^t, \pi_{-i}^t) \le \frac{\epsilon}{4}, \quad |\hat{V}(\hat{\pi}_i^t, \pi_{-i}^t) - V(\hat{\pi}_i^t, \pi_{-i}^t)| \le \epsilon/8, \quad |\hat{V}^t(\pi^t) - V(\pi^t)| \le \epsilon/8$$

holds for all $i \in [n]$ and $t \le \frac{4H}{\epsilon}$. On this event,

$$\delta_i^t = \hat{V}^t(\hat{\pi}_i, \pi_{-i}) - \hat{V}^t(\pi^t) \le V(\hat{\pi}_i^t, \pi_{-i}^t) - V(\pi^t) + \epsilon/4.$$

If the while loop doesn't end after the $t-$th iteration and $t \le 4H/\epsilon$, there exists $j^t$ s.t. $\Delta_{j^t}^t \ge \epsilon/2$, so we have

$$V(\hat{\pi}_{j^t}^t, \pi_{-j^t}^t) - V(\pi^t) \ge \Delta_{j^t}^t - \epsilon/4 \ge \epsilon.$$

Since the value function is bound by $H$, so the while loop ends within $4H/epsilon$ steps. Therefore, the inequality above that holds for all $i \in [n]$ and $t \le 4H/epsilon$ holds for simultaneously before the end of the while loop with probability at least $1 - \delta$. Again, on this event, if the while loop stops at the end of $t^{th}$ step, we have $\max_{i \in [n]} \delta_i^t \le \epsilon/2$, then

$$\max_{\mu_i} V(\mu_i, \pi_{-i}^t) - V(\pi^t) = \max_{\mu_i} V(\mu_i, \pi_{-i}^t) - V(\hat{\pi}_i^t, \pi_{-i}^t) + V(\hat{\pi}_i^t, \pi_{-i}^t) - V(\pi^t)$$

$$\le \epsilon/4 + \hat{V}^t(\hat{\pi}_i^t, \pi_{-i}^t) - \hat{V}^t(\pi^t) + 2\epsilon/8$$

$$\le \epsilon/2 + \Delta_i^t$$

$$\le \epsilon.$$

So the returned policy $\pi^t$ is an $\epsilon-$approximate local optimal. Therefore, we can conclude that probability at least $1 - \delta$, within $4H/\epsilon$ steps of the while loop, Algorithm E.1 outputs an $\epsilon-$approximate local optimal policy.

Eventually, we compute the total number of episodes as the total sample complexity. According to the definition of $N$ and $K_i$ for all $i \in [n]$, we can obtain that

$$K = \frac{4H}{\epsilon}\left(N + \sum_{i=1}^n (K_i + N)\right) = \tilde{\mathcal{O}}\left(\sum_{i=1}^n \frac{S^4 O^2 A^{2 \cdot \mathrm{ind}(G[\mathrm{ch}(i) \cup \{i\}])+2}(S^2 A^{\mathrm{ind}(G)+1} + SO) \cdot \mathrm{poly}(H)}{\alpha^4 \epsilon^3}\right).$$

$\square$

# E SUPPLEMENTARY DETAILS FOR SECTION 5

## E.1 COMPLETE ALGORITHM FOR ACHIEVING LOCAL OPTIMAL WITH MEMORYLESS POLICIES

The complete Algorithm for achieving local optimal with memoryless policies is presented in Algorithm E.1. If the central agent of Algorithm 5.1 is $n$, then the bonus parameter $\beta_m$ for $m \in [n]$ is defined as

$$\beta_m = c\big(H(S^2 A^2 O^2 + O^2 S) \log(TSAOH) + \log(Tn/\delta)\big), m \in [n-1]$$
$$\beta_n = c\big(H(S^2 AO + OS) \log(TSAOH) + \log(Tn/\delta)\big). \tag{17}$$

**Trajectory Probability** $\forall m \in [n]$, we use $\tau_m = (o_{1,m}, a_{1,m}, \ldots, o_{H,m}, a_{H,m})$ to denote the trajectory of the $m^{th}$ agent, and we denote the parameter $\theta_m = (\mathbb{T}_m, \mathbb{O}_m, \mu_m)$ as the collection of parameters representing the joint probability of the $m^{th}$ and $i^{th}$ agent's trajectory. We define trajectory probability $\mathbb{P}_{\theta_i, i}^{\pi_i}(\tau_i)$ and $\mathbb{P}_{\theta_m, m}^{\pi_i, \pi_m}(\tau_i, \tau_m)$ for all $m \in [n] \setminus \{i\}$ as follows:

$$\mathbb{P}_{\theta_i, i}^{\pi_i}(\tau_i) = \sum_{s_1, \ldots, s_H} \mu(s_1) \mathbb{O}_{1,i} \pi_{1,i}\big[\prod_{h=1}^{H-1} \mathbb{T}_{h,i,a_{h,i}} \mathbb{O}_{h+1,i} \pi_{h+1,i}\big],$$

$$\mathbb{P}_{\theta_m, m}^{\pi_i, \pi_m}(\tau_i, \tau_m) = \sum_{s_1, \ldots, s_H} \mu(s_1) \mathbb{O}_{1,i} \pi_{1,i} \pi_{1,m}\big[\prod_{h=1}^{H-1} \mathbb{T}_{h,i,a_{h,i}, a_{h,m}} \mathbb{O}_{h+1,m} \pi_{h+1,i} \pi_{h+1,m}\big]. \tag{18}$$

---

**Algorithm 6** NASH-CA for Achieving Local Optimal with Memoryless Policies

---

1: **Initialize** $\pi = \{\pi_i\}_{i \in [n]}$, where $\pi_i = \{\pi_{h,i}\}_{(h,i) \in [H] \times [m]}$.

2: **while** true **do**

3:     Execute policy $\pi$ for $N = \frac{CH^2}{\epsilon^2} \log \left(nHSK \max_{i \in [n]} A_i/(\epsilon\delta)\right)$ episodes and obtain $\hat{V}_{1,i}(\pi)$ which is the empirical average of the total return under policy $\pi$.

4:     **for** agent $i = 1, \ldots, m$ **do**

5:         Fix $\boldsymbol{\pi}_{-i}$ let the $i^{th}$ agent be the central agent to run Algorithm E.1 for $K_i = \tilde{\mathcal{O}}(S^4 O^4 A^4 (S^2 A^2 O^2 + SO^2) \times \text{poly}(H)/(\alpha^4 \epsilon^2))$ episodes and get a new policy $\hat{\pi}_i$.

6:         Execute policy $(\hat{\pi}_i, \pi_{-i})$ for $N = \frac{CH^2}{\epsilon^2} \log(nHSK \max_{i \in [n]} A_i/(\epsilon\delta))$ episodes and obtain $\hat{V}(\hat{\pi}_i, \pi_{-i})$ which is the empirical average of the total return under policy $(\hat{\pi}_i, \pi_{-i})$.

7:         Set $\Delta_i \leftarrow \hat{V}(\hat{\pi}_i, \pi_{-i}) - \hat{V}(\pi)$.

8:     **if** $\max_{i \in [n]} \Delta_i > \epsilon/2$ **then**

9:         Update $\pi_j \leftarrow \hat{\pi}_j$ where $j = \arg\max_{i \in [n]} \Delta_i$.

10:     **else**

11:         **return** $\pi$

---

**Algorithm 7** OMLE for memoryless policy

---

1: **Input**: central agent $i$, and the policy for agent $[n]/\{i\}$, $\pi_1, \pi_2, \ldots, \pi_{i-1}, \pi_{i+1}, \ldots, \pi_n$.

2: **Initialize**: $\mathcal{B}_i^1 = \{\hat{\theta}_i \in \Theta_i : \min_h \sigma_S(\mathbb{O}_i(\hat{\theta}_i) \geq \alpha)\}$, $\mathcal{B}_m^1 = \{\hat{\theta}_m \in \Theta_m : \min_h \sigma_S(\mathbb{O}_m(\hat{\theta}_m) \geq \alpha/\sqrt{O})\}$ for all $m \in [n]/\{i\}$. Set $\mathcal{D}_m = \{\}$, for all agents $m \in [n]$.

3: **for** $k = 1 \ldots T$ **do**

4:     compute $(\theta_1^k, \theta_2^k, \ldots, \theta_n^k, \pi_i^k) = \arg\max_{\hat{\theta}_1 \in \mathcal{B}_1^k, \hat{\theta}_2 \in \mathcal{B}_2^k, \ldots, \hat{\theta}_n \in \mathcal{B}_n^k, \pi_i} \sum_{m=1}^n V_m^{\pi_i, \pi_{-i}}(\hat{\theta}_m)$

5:     follow $\pi^k$ to collect a trajectory $\boldsymbol{\tau}^k = (\mathbf{o}_1^k, \mathbf{a}_1^k, \ldots, \mathbf{o}_H^k, \mathbf{a}_H^k)$

6:     add $(\pi_i^k, \tau_i^k, \tau_m^k)$ into $\mathcal{D}_m$ for $m \in [n]/\{i\}$ and add $(\pi_i^k, \tau_i^k)$ into $\mathcal{D}_i$, and update $\mathcal{B}_i^{k+1}$ and $\mathcal{B}_m^{k+1}$ for $m \in [n]/\{i\}$ as follows:

$$\mathcal{B}_i^{k+1} = \left\{\hat{\theta}_i \in \mathcal{B}_i^1 : \sum_{(\pi_i, \tau_i) \in \mathcal{D}_i} \log \mathbb{P}_{\hat{\theta}_i, i}^{\pi_i}(\tau_i) \geq \max_{\theta_i' \in \Theta_i} \sum_{(\pi_i, \tau_i) \in \mathcal{D}_i} \log \mathbb{P}_{\theta_i', i}^{\pi_i}(\tau_i) - \beta_i\right\}$$

$$\mathcal{B}_m^{k+1} = \left\{\hat{\theta}_m \in \mathcal{B}_m^1 : \sum_{(\pi_i, \tau_i, \tau_m) \in \mathcal{D}_m} \log \mathbb{P}_{\hat{\theta}_m, m}^{\pi_i, \pi_m}(\tau_i, \tau_m) \geq \max_{\theta_m' \in \Theta_m} \sum_{(\pi_i, \tau_i, \tau_m) \in \mathcal{D}_m} \log \mathbb{P}_{\theta_m', m}^{\pi_i, \pi_m}(\tau_i, \tau_m) - \beta_m\right\}$$

7: **Output** $\hat{\pi}$ as uniform mixture of the policies $\pi_i^1, \pi_i^2, \ldots, \pi_i^K$.

---

$\forall h \in [H]$, the notation in the equations are defined as: for agent $i$, $\pi_{h,i} = \pi_{h,i}(a_{h,i} \mid o_{h,i})$, $\mathbb{T}_{h,i,a_{h,i}} = \mathbb{T}_{h,i,a_{h,i}}(s_{h+1} \mid s_h, o_{h,i})$, $\mathbb{O}_{h,i} = \mathbb{O}_{h,i}(o_{h,i} \mid s_h)$, and $\forall m \in [n] \setminus \{i\}$, $\pi_{h,m} = \pi_{h,m}(a_{h,m} \mid o_{h,m})$, $\mathbb{T}_{h,m,a_{h,m}} = \mathbb{T}_{h,m,a_{h,m},a_{h,i}}(s_{h+1} \mid s_h, o_{h,m}, o_{h,i})$, $\mathbb{O}_{h,m} = \mathbb{O}_{h,m}(o_{h,m}, o_{h,i} \mid s_h)$.

For the real transition model $\theta_i = \theta_i^*$, the observation probability $\{\mathbb{O}_{h,i}\}_{h \in [H]}$ and transition probability $\{\mathbb{T}_{h,i}\}_{h \in [H]}$ are defined as ($\forall s_h \in \mathcal{S}, s_{h+1} \in \mathcal{S}, o_{h,i} \in \mathcal{O}_i, a_{h,i} \in \mathcal{A}_i$.)

$$\mathbb{O}_{h,i}(o_{h,i} \mid s_h) = \sum_{\{o_{h,m}\}_{m \in [n]/\{i\}}} \mathbb{O}_h(\mathbf{o}_h \mid s_h)$$

$$\mathbb{T}_{h,i,a_{h,i}}(s_{h+1} \mid s_h, o_{h,i}) = \sum_{\{(o_{h,m},a_{h,m})\}_{m \in [n]/\{i\}}} \left[ \prod_{j \in [n]/\{i\}} \pi_{h,j}(a_{h,j} \mid o_{h,j}) \right] \frac{\mathbb{O}_h(\mathbf{o}_h \mid s_h)}{\mathbb{O}_{h,i}(o_{h,i} \mid s_h)} \mathbb{T}_{h,\mathbf{a}_h}(s_{h+1} \mid s_h)$$

$$\tag{19}$$

Similarly, for all $m \in [n]/\{i\}$, the real transition model $\theta_m = \theta_m^*$, the observation probability $\{\mathbb{O}_{h,m}\}_{h \in [H]}$ and transition probability $\{\mathbb{T}_{h,m}\}_{h \in [H]}$ are defined as ($\forall s_h \in \mathcal{S}, s_{h+1} \in \mathcal{S}, o_{h,m} \in \mathcal{O}_m, a_{h,m} \in \mathcal{A}_m$.)

$$\mathbb{O}_{h,m}(o_{h,i}, o_{h,m} \mid s_h) = \sum_{\{o_{h,r}\}_{r \in [n]/\{i,m\}}} \mathbb{O}_h(\mathbf{o}_h \mid s_h)$$

$$\mathbb{T}_{h,m,a_{h,i},a_{h,m}}(s_{h+1} \mid s_h, o_{h,i}, o_{h,m}) = \sum_{\{(o_{h,r},a_{h,r})\}_{m \in [n]/\{i,m\}}} \left[ \prod_{j \in [n]/\{i,m\}} \pi_{h,j}(a_{h,j} \mid o_{h,j}) \right] \frac{\mathbb{O}_h(\mathbf{o}_h \mid s_h)}{\mathbb{O}_{h,m}(o_{h,i}, o_{h,m} \mid s_h)} \mathbb{T},$$

$$\tag{20}$$

where we denote $\mathbb{T}_{h,\mathbf{a}_h}$ as $\mathbb{T}_{h,\mathbf{a}_h}(s_{h+1} \mid s_h)$.

### E.2 PROOF FOR THEOREM 5.1

We first proof the following theorem, which can be seen as the bound of regret for Algorithm E.1.

**Theorem E.1.** *If Algorithm 5.1 take agent $i$ as central agent and take policies $\{\pi_m\}_{m \in [n]/\{i\}}$ as input, then Algorithm 5.1 guarantees that with probability at least $1 - \delta$ for all $k \in [T]$*

$$\left[ \sum_{t=1}^{k} V^{\pi_i^*, \boldsymbol{\pi}_{-i}} - V^{\pi_i^k, \boldsymbol{\pi}_{-i}} \right] \leq \tilde{\mathcal{O}} \left( \frac{S^2 O^2 A^2}{\alpha^2} \sqrt{k(S^2 A^2 O^2 + SO^2)} \times poly(H) \right),$$

*where $\pi_i^*$ is defined as*

$$\pi_i^* = \arg \max_{\pi_i} V^{\pi_i, \boldsymbol{\pi}_{-i}}.$$

According to the symmetric principle, we assume the central agent in Algorithm E.1 is agent $n$ without loss of generosity.

**Definition E.1.** *For all $(m, i) \in [n-1] \times [T]$, $\theta_m \in \Theta_m$, and any policy $\pi_m$ of agent $m$, we define $f_m(\theta_m, \pi_n, \pi_m,)$ and $\tilde{f}_m^i(\theta_m, \pi_n, \pi_m)$ as:*

$$f_m(\theta_m, \pi_n, \pi_m) = \mathbb{P}_{\theta_m, m}^{\pi_n, \pi_m}(\tau_n, \tau_m), \quad \tilde{f}_m^i(\theta_m, \pi_n, \pi_m) = \mathbb{P}_{\theta_m, m}^{\pi_n, \pi_m}\left(\tau_n^i, \tau_m^i\right).$$

*For agent $n$, we define $f_n(\theta_n, \pi_n)$ and $\tilde{f}_n^i(\theta_n, \pi_n)$ as:*

$$f_n(\theta_n, \pi_n) = \mathbb{P}_{\theta_n, n}^{\pi_n}(\tau_n), \quad \tilde{f}_n^i(\theta_n, \pi_n) = \mathbb{P}_{\theta_n, n}^{\pi_n}\left(\tau_n^i\right).$$

**Lemma E.1.** *There exist an absolute constant $c$ such that for any $\delta \in (0, 1]$, with probability at least $1 - \delta$, the following inequality holds for all $t \in [T]$ and all $\theta_m \in \Theta_m$, $m \in [n]$.*

$$\sum_{i=1}^{t} \log \left( \tilde{f}_m^i\left(\theta_m, \pi_n^i, \pi_m\right) / \tilde{f}_m^i\left(\theta_m^\star, \pi_n^i, \pi_m\right) \right) \leq \beta_m, \quad \sum_{i=1}^{t} \log \left( \tilde{f}_n^i\left(\theta_n, \pi_n^i\right) / \tilde{f}_n^i\left(\theta_n^\star, \pi_n^i\right) \right) \leq \beta_n,$$

*where we define bonus term $\beta_n$ and $\beta_m$ for all $m \in [n-1]$ as:*

$$\beta_m = c\left(H(S^2 A^2 O^2 + O^2 S) \log(TSAOH) + \log(Tn/\delta)\right),$$

$$\beta_n = c\left(H(S^2 AO + OS) \log(TSAOH) + \log(Tn/\delta)\right).$$

*Proof.* We first prove the first inequality. We use $\theta_n = (\mathbb{T}_n, \mathbb{O}_n, \mu)$ to denote the ensemble of all the parameter of the probability of trajectory $\tau_n^i$. We use $\Theta_n$ to denote the collections of all such parameters $\theta_n$. We can view $\Theta_n$ as a subset of a $d_n = H(S^2AO + SO) + S$ dimension subspace. We denote $\bar{\theta}_n$ as the optimistic $\epsilon-$discretelization of $\theta_m$ so that $\bar{\theta}_{m,i} = \lceil \theta_{m,i}/\epsilon \rceil \times \epsilon$ for all coordinates $i$. We always have $f_n(\bar{\theta}_n, \pi_n) \geq f_n(\theta_n, \pi_n)$. We can choose $\epsilon \leq 1/(c(S + O + A)HT)$ such that $\sum_{\tau_n} |f_n(\bar{\theta}_n, \pi_n) - f_n(\theta_n, \pi_n)| \leq 1/T$. We use $\bar{\Theta}_n$ to represent the collections of all such $\bar{\theta}_n$, then, the log-cardinality of $\bar{\Theta}_n$ is bounded by

$$\log |\bar{\theta}_n| \leq \mathcal{O}\left(H(S^2AO + SO)\log(TSAOH)\right).$$

We denote $\mathbb{E}_t[\cdot] = \mathbb{E} = [\cdot | \{(\pi_n^i, \boldsymbol{\tau}^i)\}_{i=1}^{t-1} \cup \{\pi_n^i\}]$.

$$\mathbb{E}\left[\exp\left(\sum_{i=1}^{t} \log\left(\tilde{f}_n^i\left(\bar{\theta}_n, \pi_n^i\right)/\tilde{f}_n^i\left(\theta_n^\star, \pi_n^i\right)\right)\right)\right]$$

$$=\mathbb{E}\left[\exp\left(\sum_{i=1}^{t-1} \log\left(\tilde{f}_n^i\left(\bar{\theta}_n, \pi_n^i\right)/\tilde{f}_n^i\left(\theta_n^\star, \pi_n^i\right)\right)\right) \cdot \mathbb{E}_t\left[\exp\left(\log\left(\tilde{f}_n^t(\bar{\theta}_n, \pi_n^t)/\tilde{f}_n^t(\theta_n^\star, \pi_n^t)\right)\right)\right]\right]$$

$$=\mathbb{E}\left[\exp\left(\sum_{i=1}^{t-1} \log\left(\tilde{f}_n^i\left(\bar{\theta}_n, \pi_n^i\right)/\tilde{f}_n^i\left(\theta_n^\star, \pi_n^i\right)\right)\right) \cdot \mathbb{E}_t\left(\tilde{f}_n^t(\bar{\theta}_n, \pi_n^t)/\tilde{f}_n^t(\theta_n^\star, \pi_n^t)\right)\right] \tag{21}$$

Since we have

$$\mathbb{E}_t\left(\tilde{f}_n^t(\bar{\theta}_n, \pi_n^t)/\tilde{f}_n^t(\theta_n^\star, \pi_n^t)\right) = \sum_{\tau_n} \tilde{f}_n^t\left(\bar{\theta}_n, \pi_n^t\right) \leq (1 + 1/T),$$

we obtain that

$$\mathbb{E}\left[\exp\left(\sum_{i=1}^{t} \log\left(\tilde{f}_n^i\left(\bar{\theta}_n, \pi_n^i\right)/\tilde{f}_n^i\left(\theta_n^\star, \pi_n^i\right)\right)\right)\right] \leq e.$$

Therefore, by Markov's inequality, we have

$$\mathbb{P}\left(\sum_{i=1}^{t} \log\left(\frac{\tilde{f}_n^i\left(\bar{\theta}_n, \pi_n^i\right)}{\tilde{f}_n^i\left(\theta_n^\star, \pi_n^i\right)}\right) > \log(1/\delta)\right) \leq \mathbb{E}\left[\exp\sum_{i=1}^{t} \log\left(\frac{\tilde{f}_n^i\left(\bar{\theta}_n, \pi_n^i\right)}{\tilde{f}_n^i\left(\theta_n^\star, \pi_n^i\right)}\right)\right] \cdot \exp(-\log(1/\delta)) \leq e\delta.$$

We take a union bound over all $(\bar{\theta}_n, t) \in \bar{\Theta}_n \times [T]$ and rescaling $\delta$, we obtain

$$\mathbb{P}\left(\max_{(\bar{\theta}_n, t) \in \bar{\Theta}_n \times [T]} \sum_{i=1}^{t} \log\left(\frac{\tilde{f}_n^i\left(\bar{\theta}_n, \pi_n^i\right)}{\tilde{f}_n^i\left(\theta_n^\star, \pi_n^i\right)}\right) > c(H(S^2AO + SO)\log(TSAOH) + \log(T/\delta))\right) \leq \delta.$$

Since $\bar{\theta}_n$ is an optimistic discretization of $\theta_n$, which implies that $\mathbb{P}_{\theta_n,n}^{\pi_n}(\tau_n) \leq \mathbb{P}_{\bar{\theta}_n,n}^{\pi_n}(\tau_n)$ for all $\theta_n, \pi_n, \tau_n$. As a result, we obtain that

$$\mathbb{P}\left(\max_{(\theta_n, t) \in \Theta_n \times [T]} \sum_{i=1}^{t} \log\left(\frac{\tilde{f}_n^i\left(\theta_n, \pi_n^i\right)}{\tilde{f}_n^i\left(\theta_n^\star, \pi_n^i\right)}\right) > c(H(S^2AO + SO)\log(TSAOH) + \log(T/\delta))\right) \leq \delta.$$

We similarly consider $f_m(\theta_m, \pi_n, \pi_m)$ for all $m \in [n-1]$. We use a $d_m = H(S^2A^2O^2 + SO^2) + S$ dimension parameter $\theta_m = (\mathbb{T}_m, \mathbb{O}_m, \mu)$ to denote the ensemble of all the parameters of the probability of trajectories $(\tau_n^i, \tau_m^i)$. We denote $\Theta_m$ as the collection of all the $\epsilon-$optimistic discretelization, $\bar{\theta}_m$. We still choose $\epsilon \leq 1/(c(S + O + A)HT)$ for large constant $c$ so that $\sum_{\tau_n, \tau_m} |f_m\left(\bar{\theta}_m, \pi_n, \pi_m\right) - f_m\left(\theta_m, \pi_n, \pi_m\right)| \leq 1/T$. With similar analysis as above, we can derive the following inequality:

$$\mathbb{P}\left(\max_{(\theta_m, t) \in \Theta_m \times [T]} \sum_{i=1}^{t} \log\left(\tilde{f}_m^i\left(\theta_m, \pi_n^i, \pi_m\right)/\tilde{f}_m^i\left(\theta_m^\star, \pi_n^i, \pi_m\right)\right) > \beta_m\right) \leq \delta.$$

Eventually, we can obtain that with probability at least $1 - \delta$, the following events hold true:

$$\sum_{i=1}^{t} \log\left(\tilde{f}_m^i\left(\theta_m, \pi_n^i, \pi_m\right)/\tilde{f}_m^i\left(\theta_m^\star, \pi_n^i, \pi_m\right)\right) \leq \beta_m, \quad \sum_{i=1}^{t} \log\left(\tilde{f}_n^i\left(\theta_n, \pi_n^i\right)/\tilde{f}_n^i\left(\theta_n^\star, \pi_n^i\right)\right) \leq \beta_n,$$

$\square$

**Lemma E.2.** *There exists a universal constant $c$ such that for any $\delta \in (0,1]$, with probability at least $1 - \delta$, for all $t \in [T]$ and all $\theta_m \in \Theta_m$, $m \in [n-1]$, it holds that*

$$\sum_{t=1}^{k} \left( \sum_{\tau_n \in (\mathcal{O}_n \times \mathcal{A}_n)^H, \tau_m \in (\mathcal{O}_m \times \mathcal{A}_m)^H} \left| f_m \left( \theta_m, \pi_n^i, \pi_m \right) - f_m \left( \theta_m^\star, \pi_n^i, \pi_m \right) \right| \right)^2$$

$$\leq c \left( \sum_{t=1}^{k} \log \left( \tilde{f}_m^i \left( \theta_m^\star, \pi_n^i, \pi_m \right) / \tilde{f}_m^i \left( \theta_m, \pi_n^i, \pi_m \right) \right) + \beta_m \right),$$

*and for $\theta_n \in \Theta_n$, it holds that*

$$\sum_{t=1}^{k} \left( \sum_{\tau_n \in (\mathcal{O}_n \times \mathcal{A}_n)^H} \left| f_n \left( \theta_n, \pi_n^i \right) - f_n \left( \theta_n^\star, \pi_n^i \right) \right| \right)^2 \leq c \left( \sum_{i=1}^{t} \log \left( \frac{\tilde{f}_n^i \left( \theta_n^\star, \pi_n^i \right)}{\tilde{f}_n^i \left( \theta_n, \pi_n^i \right)} \right) + \beta_n \right)$$

*Proof.* The proof of this lemma is very similar to the proof for Lemma C.2, so we omit it here for clarity. $\square$

**Lemma E.3.** *We have the following bound on the regret of Algorithm 5.1.*

$$\sum_{k=1}^{K} \left[ V^{\pi_n^*, \boldsymbol{\pi}_{-n}} - V^{\pi_n^k, \boldsymbol{\pi}_{-n}} \right]$$

$$\leq H \cdot \sum_{t=1}^{K} \sum_{\tau_{H,n}} \left| \mathbb{P}_{\theta_n^t, n}^{\pi_n^t} (\tau_{H,n}) - \mathbb{P}_{\theta_n^*, n}^{\pi_n^t} (\tau_{H,n}) \right| + H \cdot \sum_{m=1}^{n-1} \sum_{t=1}^{K} \sum_{\tau_{H,n}, \tau_{H,m}} \left| \mathbb{P}_{\theta_m^t, m}^{\pi_n^t, \pi_m} (\tau_{H,n}, \tau_{H,m}) - \mathbb{P}_{\theta_m^*, m}^{\pi_n^t, \pi_m} (\tau_{H,n}, \tau_{H,m}) \right|,$$

*where we define $\pi_i^*$ as $\pi_i^* = \arg \max_{\pi_i} V^{\pi_i, \boldsymbol{\pi}_{-i}}$.*

*Proof.* For any policy $\boldsymbol{\pi} = (\pi_1, \pi_2, \ldots, \pi_n)$, we can decompose the value function $V^{\boldsymbol{\pi}}$ as follows:

$$V^{\boldsymbol{\pi}} = \mathbb{E}_{\boldsymbol{\pi}} \left[ \sum_{m=1}^{n} \sum_{h=1}^{H} r_h(o_{h,m}) \right] = \sum_{m=1}^{n} \mathbb{E}_{\boldsymbol{\pi}} \left[ \sum_{h=1}^{H} r_h(o_{h,m}) \right]$$

$$= \sum_{m=1}^{n} \sum_{\{\tau_{H,j}\}_{j=1}^n} \mathbb{P}^{\boldsymbol{\pi}}(\boldsymbol{\tau}_H) \cdot \left( \sum_{h=1}^{H} r_h(o_{h,m}) \right)$$

$$= \sum_{m=1}^{n-1} \sum_{\tau_{H,m}, \tau_{H,n}} \mathbb{P}_{\theta_m^*, m}^{\pi_n, \pi_m} (\tau_{H,m}, \tau_{H,n}) \cdot \left( \sum_{h=1}^{H} r_h(o_{h,m}) \right) + \sum_{\tau_{H,n}} \mathbb{P}_{\theta_n^*, n}^{\pi_n} (\tau_{H,n}) \cdot \left( \sum_{h=1}^{H} r_h(o_{h,n}) \right).$$

For $m \in [n-1]$, we further define

$$V_n^{\pi_n, \boldsymbol{\pi}_{-n}}(\theta_n) = \sum_{\tau_{H,n}} \mathbb{P}_{\theta_n, n}^{\pi_n} (\tau_{H,n}) \cdot \left( \sum_{h=1}^{H} r_h(o_{h,n}) \right), \qquad V_m^{\pi_n, \boldsymbol{\pi}_{-n}}(\theta_m) \sum_{\tau_{H,m}, \tau_{H,n}} \mathbb{P}_{\theta_m^*, m}^{\pi_n, \pi_m} (\tau_{H,m}, \tau_{H,n}) \cdot \left( \sum_{h=1}^{H} r_h(o_{h,m}) \right).$$

Then we can decompose the value function as

$$V^{\pi_n^*, \boldsymbol{\pi}_{-n}} = \sum_{m=1}^{n} V_m^{\pi_n^t, \boldsymbol{\pi}_{-n}}(\theta_m^*), \quad V^{\pi_n^k, \boldsymbol{\pi}_{-n}} = \sum_{m=1}^{n} V_m^{\pi_n^*, \boldsymbol{\pi}_{-n}}(\theta_m^*).$$

According to Lemma E.1 and Lemma E.2, we can deduce that $\theta_m^* \in \cap_{t \in [K]} \mathcal{B}_m^t$, holds for all $m \in [n]$ with probability at least $1 - \delta$. In the following analysis, we assume that $\theta_m^* \in \cap_{t \in [K]} \mathcal{B}_m^t$. On this event, according to the optimism of $\{\theta_m\}_{m \in [n]}$ and $\pi_n^t$ for $t \in [K]$, we can obtain that

$$V^{\pi_n^*, \boldsymbol{\pi}_{-n}} - V^{\pi_n^k, \boldsymbol{\pi}_{-n}} = \sum_{m=1}^{n} V_m^{\pi_n^*, \boldsymbol{\pi}_{-n}}(\theta_m^*) - \sum_{m=1}^{n} V_m^{\pi_n^t, \boldsymbol{\pi}_{-n}}(\theta_m^*) \leq \sum_{m=1}^{n} V_m^{\pi_n^t, \boldsymbol{\pi}_{-n}}(\theta_m^t) - \sum_{m=1}^{n} V_m^{\pi_n^t, \boldsymbol{\pi}_{-n}}(\theta_m^*).$$

According to the definition of $\left\{V_m^{\pi_n^t, \boldsymbol{\pi}-n}(\theta_m^*)\right\}_{m \in [n]}$ and $\left\{V_m^{\pi_n^t, \boldsymbol{\pi}-n}(\theta_m^t)\right\}_{m \in [n]}$, we further have the following bound on the regret.

$$
\sum_{t=1}^{K}\left[V^{\pi_n^*, \boldsymbol{\pi}-n} - V^{\pi_n^k, \boldsymbol{\pi}-n}\right]
$$

$$
\leq \sum_{t=1}^{K}\left[\sum_{m=1}^{n} V_m^{\pi_n^t, \boldsymbol{\pi}-n}(\theta_m^t) - \sum_{m=1}^{n} V_m^{\pi_n^t, \boldsymbol{\pi}-n}(\theta_m^*)\right]
$$

$$
\leq H \cdot \sum_{t=1}^{K} \sum_{\tau_{H,n}} \left|\mathbb{P}_{\theta_n^t}^{\pi_n^t}(\tau_{H,n}) - \mathbb{P}_{\theta_n^*}^{\pi_n^t}(\tau_{H,n})\right| + H \cdot \sum_{m=1}^{n-1} \sum_{t=1}^{K} \sum_{\tau_{H,n}, \tau_{H,m}} \left|\mathbb{P}_{\theta_m^t}^{\pi_n^t, \pi_m}(\tau_{H,n}, \tau_{H,m}) - \mathbb{P}_{\theta_m^*}^{\pi_n^t, \pi_m}(\tau_{H,n}, \tau_{H,m})\right|.
$$

$\square$

**Definition E.2.** *For all* $(m, h, k) \in [n-1] \times [H] \times [T]$*, we define the matrix notations* $\mathbb{M}_{h,m}^\star \in \mathbb{R}^{O^2 \times O^2}$ *and* $\mathbb{M}_{h,m}^k \in \mathbb{R}^{O^2 \times O^2}$ *as follows:*

$$
\mathbb{M}_{0,m}^\star = \mathbb{O}_{1,m}^\star \mu_m^\star \in \mathbb{R}^O, \quad \mathbb{M}_{0,m}^k = \mathbb{O}_{1,m}^k \mu_m^k \in \mathbb{R}^O,
$$

$$
\mathbb{M}_{h,m}^\star(o_m, a_m, a_n) = \mathbb{O}_{h+1,m}^\star \mathbb{T}_{h,m,a_m,a_n}^\star \cdot \textit{diag}\left(\mathbb{O}_{h,m}^\star(o_m, o_n \mid \cdot)\right) \left(\mathbb{O}_{h,m}^\star\right)^\dagger \in \mathbb{R}^{O^2 \times O^2},
$$

$$
\mathbb{M}_{h,m}^k(o_m, a_m, a_n) = \mathbb{O}_{h+1,m}^k \mathbb{T}_{h,m,a_m,a_n}^k \cdot \textit{diag}\left(\mathbb{O}_{h,m}^k(o_m, o_n \mid \cdot)\right) \left(\mathbb{O}_{h,m}^k\right)^\dagger \in \mathbb{R}^{O^2 \times O^2},
$$

*and for agent* $n$*, we define matrix notations* $\mathbb{M}_{h,n}^\star \in \mathbb{R}^{O \times O}$ *and* $\mathbb{M}_{h,n}^k \in \mathbb{R}^{O \times O}$ *as follows for all* $(h, k) \in [H] \times [T]$

$$
\mathbb{M}_{0,n}^\star = \mathbb{O}_{1,n}^\star \mu_n^\star \in \mathbb{R}^O, \quad \mathbb{M}_{0,m}^k = \mathbb{O}_{1,n}^k \mu_n^k \in \mathbb{R}^O,
$$

$$
\mathbb{M}_{h,n}^\star(o_n, a_n) = \mathbb{O}_{h+1,n}^\star \mathbb{T}_{h,n,a_n}^\star \cdot \textit{diag}\left(\mathbb{O}_{h,n}^\star(o_n \mid \cdot)\right) \left(\mathbb{O}_{h,n}^\star\right)^\dagger \in \mathbb{R}^{O \times O},
$$

$$
\mathbb{M}_{h,n}^k(o_n, a_n) = \mathbb{O}_{h+1,n}^k \mathbb{T}_{h,n,a_n}^k \cdot \textit{diag}\left(\mathbb{O}_{h,n}^k(o_n \mid \cdot)\right) \left(\mathbb{O}_{h,n}^k\right)^\dagger \in \mathbb{R}^{O \times O},
$$

*where* $\{\mathbb{O}_{h,m}^\star\}_{(h,m) \in [H] \times [n]}$ *and* $\{\mathbb{T}_{h,m}^\star\}_{(h,m) \in [H] \times [n]}$ *denote the observation and transition matrices corresponding to the true transition model, and* $\{\mathbb{O}_{h,m}^k\}_{(h,m) \in [H] \times [n]}$ *and* $\{\mathbb{T}_{h,m}^k\}_{(h,m) \in [H] \times [n]}$ *denote the observation and transition matrices corresponding to model parameter* $\theta_m^k$ *for all* $k \in [T]$*. When no confusion arises, we simplify the notation by using* $\mathbb{M}_{h,m}^\star$ *to represent* $\mathbb{M}_{h,m}^\star(o_m, a_m, a_n)$ *and* $\mathbb{M}_{h,m}^k$ *to represent* $\mathbb{M}_{h,m}^k(o_m, a_m, a_n)$ *for* $m \in [n-1]$*. We also simplify by using* $\mathbb{M}_{h,n}^\star$ *to represent* $\mathbb{M}_{h,n}^\star(o_n, a_n)$ *and* $\mathbb{M}_{h,n}^k$ *to represent* $\mathbb{M}_{h,n}^k(o_n, a_n)$*.*

**Lemma E.4.** *Given* $O \times S$ *matrix* $\mathbf{A}_1, \mathbf{A}_2, \ldots, \mathbf{A}_n$*. We further define matrix* $\mathbf{B}$ *and matrix* $\mathbf{C}$ *as* $\mathbf{B}^\intercal = (\mathbf{A}_1^\intercal, \mathbf{A}_2^\intercal, \ldots, \mathbf{A}_n^\intercal)$*,* $\mathbf{C} = \mathbf{A}_1 + \mathbf{A}_2 + \cdots + \mathbf{A}_n$*. Then if* $\sigma_S(\mathbf{C}) \geq \alpha$*, then we have* $\sigma_S(\mathbf{C}) \geq \alpha/\sqrt{n}$*.*

*Proof.* We only need to prove that for any given unit vector $\mathbf{x} \in \mathbb{R}^S$, $\|\mathbf{B}\mathbf{x}\|_2 \geq \frac{\alpha}{\sqrt{O}}$. Since we have

$$
\|\mathbf{B}\mathbf{x}\|_2 = \|(\mathbf{A}_1 \mathbf{x}^\intercal, \ldots, \mathbf{A}_n \mathbf{x}^\intercal)^\intercal\|_2 = \sqrt{\|\mathbf{A}_1 \mathbf{x}\|_2^2 + \cdots + \|\mathbf{A}_n \mathbf{x}\|_2^2} \geq \frac{1}{\sqrt{n}}\|\mathbf{A}_1 \mathbf{x} + \cdots + \mathbf{A}_n \mathbf{x}\|_2 = \frac{1}{\sqrt{n}}\|\mathbf{C}\mathbf{x}\|_2 \geq \frac{\alpha}{\sqrt{n}}.
$$

Thus, we finished the proof of the lemma. $\square$

**Corollary E.1.** *According to the definition of observable condition, we have* $\sigma_S(\mathbb{O}_{h,n}) \geq \alpha$*, for all* $h \in [H]$*, and* $\sigma_S(\mathbb{O}_{h,m}) \geq \alpha/\sqrt{O}$ *for all* $h \in [H]$ *and all* $m \in [n-1]$*.*

According to the definition of matrix notation, we can directly obtain the following lemma:

**Lemma E.5.** *(Bound the regret of Operator Estimates) The following two inequalities holds true for all* $h \in [H]$*:*

$$
\sum_{\tau_{h,n}} \left\|\left[\prod_{j=0}^{h} \mathbb{M}_{j,n}^k\right] - \left[\prod_{j=0}^{h} \mathbb{M}_{j,n}^\star\right]\right\|_1 \pi_n^t(\tau_{h,n}) \leq \frac{\sqrt{S}}{\alpha}\left(\sum_{j=0}^{h} \sum_{\tau_{j,n}} \left\|\left(\mathbb{M}_{j,n}^k - \mathbb{M}_{j,n}^\star\right)\left[\prod_{h'=0}^{j-1} \mathbb{M}_{h',n}^\star\right]\right\|_1 \pi_n^t(\tau_{j,n})\right),
$$

and for all agent $m \in [n-1]$ we further have

$$\sum_{\tau_{h,n},\tau_{h,m}} \left\| \left[ \prod_{j=0}^{h} \mathbb{M}_{j,m}^k \right] \mathbf{b}_{0,m}^k - \left[ \prod_{j=0}^{h} \mathbb{M}_{j,m}^{\star} \right] \right\|_1 \pi_n^t(\tau_{h,n}) \pi_m(\tau_{h,m})$$

$$\leq \frac{\sqrt{S}}{\alpha} \sum_{j=0}^{h} \sum_{\tau_{j,n},\tau_{j,m}} \left\| \left( \mathbb{M}_{j,m}^k - \mathbb{M}_{j,m}^{\star} \right) \left[ \prod_{h'=0}^{j-1} \mathbb{M}_{h',m}^{\star} \right] \right\|_1 \pi_n^t(\tau_{h,n}) \pi_m(\tau_{h,m}).$$

**Lemma E.6.** *(Constraints for the Operator Estimates from OMLE) With probability at least $1 - \delta$, for all $m \in [n-1]$, the following events hold true.*

$$\sum_{t=1}^{k-1} \sum_{\tau_{h,n}} \pi_n^t(\tau_{h,n}) \cdot \left\| \left( \mathbb{M}_{h,n}^k - \mathbb{M}_{h,n}^{\star} \right) \left[ \prod_{h'=0}^{h-1} \mathbb{M}_{h',n}^{\star} \right] \right\|_1 = \mathcal{O}\left( \frac{\sqrt{S}}{\alpha} \sqrt{k\beta_n} \right),$$

$$\sum_{t=1}^{k-1} \sum_{\tau_{h,n},\tau_{h,m}} \pi_n^t(\tau_{h,n}) \pi_m(\tau_{h,m}) \cdot \left\| \left( \mathbb{M}_{h,m}^k - \mathbb{M}_{h,m}^{\star} \right) \left[ \prod_{h'=0}^{h-1} \mathbb{M}_{h',m}^{\star} \right] \right\|_1 = \mathcal{O}\left( \frac{\sqrt{S}}{\alpha} \sqrt{k\beta_m} \right),$$

*where $\beta_m$ and $\beta_n$ is defined as*

$$\beta_n = c(H(S^2 A^2 O^2 + SO^2) \log(TSAOH) + \log(nT/\delta)),$$

$$\beta_n = c(H(S^2 AO + SO) \log(TSAOH) + \log(Tn/\delta)).$$

*Proof.* The proof of this lemma is very similar to the proof for Lemma C.3, so we omit it here for clarity. □

**Proof of Theorem E.1** : We only need to consider the following problem:
We are required to bound the following target term for $m \in [n-1]$.

$$\sum_{t=1}^{k} \left( \sum_{h=0}^{H-1} \sum_{\tau_{h,n}} \left\| \left( \mathbb{M}_{h,n}^t - \mathbb{M}_{h,n}^{\star} \right) \left[ \prod_{h'=0}^{h-1} \mathbb{M}_{h',n}^{\star} \right] \right\|_1 \cdot \pi_n^t(\tau_{h,n}) \right)$$

$$\sum_{t=1}^{k} \left( \sum_{h=0}^{H-1} \sum_{\tau_{h,n},\tau_{h,m}} \left\| \left( \mathbb{M}_{h,m}^t - \mathbb{M}_{h,m}^{\star} \right) \left[ \prod_{h'=0}^{h-1} \mathbb{M}_{h',m}^{\star} \right] \right\|_1 \cdot \pi_n^t(\tau_{h,n}) \pi_m(\tau_{h,m}) \right),$$

For agent $n$ we have the following condition:

$$\sum_{t=1}^{k-1} \sum_{\tau_{h,n}} \left\| \left( \mathbb{M}_{h,n}^k - \mathbf{M}_{h,n}^{\star} \right) \left[ \prod_{h'=0}^{h-1} \mathbb{M}_{h',n}^{\star} \right] \right\|_1 \times \pi_n^t(\tau_{h,n}) = \mathcal{O}\left( \frac{\sqrt{S}}{\alpha} \sqrt{k\beta_n} \right), \forall k \in [T].$$

Corresponding to agent $m \in [n-1]$ we have the following condition:

$$\sum_{t=1}^{k-1} \sum_{\tau_{h,n},\tau_{h,m}} \left\| \left( \mathbb{M}_{h,m}^k - \mathbb{M}_{h,m}^{\star} \right) \left[ \prod_{h'=0}^{h-1} \mathbb{M}_{h',m}^{\star} \right] \right\|_1 \cdot \pi_n^t(\tau_{h,n}) \pi_m(\tau_{h,m}) = \mathcal{O}\left( \frac{\sqrt{S}}{\alpha} \sqrt{k\beta_m} \right).$$

We apply Eluder-Dimension Lemma (Proposition 22 of Liu et al. (2022a)), and we can obtain the following bound on the regret with probability at least $1 - \delta$ for all $m \in [n-1]$.

$$\sum_{t=1}^{k} \left( \sum_{h=0}^{H-1} \sum_{\tau_{h,n}} \left\| \left( \mathbb{M}_{h,n}^t - \mathbb{M}_{h,n}^{\star} \right) \left[ \prod_{h'=0}^{h-1} \mathbb{M}_{h',n}^{\star} \right] \right\|_1 \pi_n^t(\tau_{h,n}) \right) = \tilde{\mathcal{O}}\left( \frac{S^{1.5} OAH^3}{\alpha} \sqrt{k\beta_n} \right)$$

$$\sum_{t=1}^{k} \left( \sum_{h=0}^{H-1} \sum_{\tau_{h,n},\tau_{h,m}} \left\| \left( \mathbb{M}_{h,m}^k - \mathbb{M}_{h,m}^{\star} \right) \left[ \prod_{h'=0}^{h-1} \mathbb{M}_{h',m}^{\star} \right] \right\|_1 \pi_n^t(\tau_{h,n}) \pi_m(\tau_{h,m}) \right) = \tilde{\mathcal{O}}\left( \frac{S^{1.5} O^2 A^2 H^3}{\alpha} \sqrt{k\beta_m} \right).$$

Therefore, we achieve the bound of the regret.

**Proof for Theorem 5.1** We are now ready to prove Theorem 5.1. We begin by restating the theorem here for the reader's convenience.

**Theorem E.2.** *If the central agent for Algorithm 5.1 is $i$, we define bonus parameter as $\beta_i = c(H(S^2AO + SO)\log(TSAOH) + \log(Tn/\delta))$, $\beta_m = c(H(S^2A^2O^2 + SO^2)\log(TSAOH) + \log(Tn/\delta))$ $(\forall m \in [n]/\{i\})$ for some constant $c$. Then, with probability at least $1 - \delta$, Algorithm terminates within $4H/\epsilon$ steps of the while loop, and outputs an $\epsilon-$approximate local optimal policy. The total episodes of play is at most*

$$K = \tilde{\mathcal{O}}\left(S^4O^4A^4(S^2A^2O^2 + SO^2) \times poly(H)/(\alpha^4\epsilon^3)\right).$$

*Proof.* We use superscript $t$ to represent variables at the $t^{th}$ step (before $\pi$ is updated) of the while loop. We set $K_i = \tilde{\mathcal{O}}(S^4O^4A^4(S^2A^2O^2 + SO^2) \times \text{poly}(H)/(\alpha^4\epsilon^2))$. According to Theorem E.1, for fixed $i$ and $t$, we have with probability at least $1 - \frac{\delta\epsilon}{8nH}$

$$\max_{\mu_i} V(\mu_i, \pi_{-i}^t) - V(\hat{\pi}_i^t, \pi_{-i}^t) \le \frac{\epsilon}{4}.$$

We then take a union bound over all $t \le 4H/\epsilon$ and for all $i \in [n]$, and we have with probability at least $1 - \delta$ the following inequality holds for all $i \in [n]$ and $t \le 4H/\epsilon$

$$\max_{\mu_i} V(\mu_i, \pi_{-i}^t) - V(\hat{\pi}_i^t, \pi_{-i}^t) \le \frac{\epsilon}{4}. \tag{22}$$

For the empirical estimator $\hat{V}^t$, it's bounded in $[0, H]$. Thus, by Hoeffding's inequality, for fixed $i \in [n]$ and $t$

$$\mathbb{P}\left(\left|\hat{V}^t - V^t\right| \ge \frac{\epsilon}{8}\right) \le 2\exp\left(-\frac{N\epsilon^2}{32H^2}\right).$$

Choosing $N = \frac{CH^2}{\epsilon^2}\log\left(\frac{nHSK\max_{i\in[n]}A_i}{\epsilon\delta}\right)$ for some large constant $C$, we have

$$\mathbb{P}\left(\left|\hat{V}^t - V^t\right| \ge \frac{\epsilon}{8}\right) \le \frac{\epsilon\delta}{16nH}.$$

Applying this inequality for $\hat{V}^t(\hat{\pi}_i^t, \pi_{-i}^t)$ and $\hat{V}^t(\pi^t)$ and taking a union bound over $i \in [n]$ and $t \le 4H/\epsilon$, we can achieve that

$$|\hat{V}(\hat{\pi}_i^t, \pi_{-i}^t) - V(\hat{\pi}_i^t, \pi_{-i}^t)| \le \epsilon/8, \quad |\hat{V}^t(\pi^t) - V(\pi^t)| \le \epsilon/8.$$

We combine this result with equation 22, and we can obtain that with probability at least $1 - \delta$

$$\max_{\mu_i} V(\mu_i, \pi_{-i}^t) - V(\hat{\pi}_i^t, \pi_{-i}^t) \le \frac{\epsilon}{4}, \qquad |\hat{V}(\hat{\pi}_i^t, \pi_{-i}^t) - V(\hat{\pi}_i^t, \pi_{-i}^t)| \le \epsilon/8, \quad |\hat{V}^t(\pi^t) - V(\pi^t)| \le \epsilon/8$$

holds for all $i \in [n]$ and $t \le \frac{4H}{\epsilon}$. On this event,

$$\delta_i^t = \hat{V}^t(\hat{\pi}_i, \pi_{-i}) - \hat{V}^t(\pi^t) \le V(\hat{\pi}_i^t, \pi_{-i}^t) - V(\pi^t) + \epsilon/4.$$

If the while loop doesn't end after the $t-$th iteration and $t \le 4H/\epsilon$, there exists $j^t$ s.t. $\Delta_{j^t}^t \ge \epsilon/2$, so we have

$$V(\hat{\pi}_{j^t}^t, \pi_{-j^t}^t) - V(\pi^t) \ge \Delta_{j^t}^t - \epsilon/4 \ge \epsilon.$$

Since the value function is bound by $H$, so the while loop ends within $4H/epsilon$ steps. Therefore, the inequality above that holds for all $i \in [n]$ and $t \le 4H/epsilon$ holds for simultaneously before the end of the while loop with probability at least $1 - \delta$. Again, on this event, if the while loop stops at the end of $t^{th}$ step, we have $\max_{i\in[n]} \delta_i^t \le \epsilon/2$, then

$$\max_{\mu_i} V(\mu_i, \pi_{-i}^t) - V(\pi^t) = \max_{\mu_i} V(\mu_i, \pi_{-i}^t) - V(\hat{\pi}_i^t, \pi_{-i}^t) + V(\hat{\pi}_i^t, \pi_{-i}^t) - V(\pi^t)$$

$$\le \epsilon/4 + \hat{V}^t(\hat{\pi}_i^t, \pi_{-i}^t) - \hat{V}^t(\pi^t) + 2\epsilon/8$$

$$\le \epsilon/2 + \Delta_i^t$$

$$\le \epsilon.$$

So the returned policy $\pi^t$ is an $\epsilon-$approximate local optimal. Therefore, we can conclude that probability at least $1 - \delta$, within $4H/\epsilon$ steps of the while loop, Algorithm E.1 outputs an $\epsilon-$approximate local optimal policy.

Eventually, we compute the total number of episodes as the total sample complexity. We first compute the number of episodes within each step of the while loop.

$$N + \sum_{i=1}^{n}(K_i + N) = \mathcal{O}\left(\frac{nH^2}{\epsilon^2} \log\left(\frac{nHSK \max_{i \in [n]} A_i}{\epsilon \delta}\right)\right) + \tilde{\mathcal{O}}\left(\frac{S^4 O^4 A^4 (S^2 A^2 O^2 + SO^2) \times \text{poly}(H)}{\alpha^4 \epsilon^2}\right).$$

Since the algorithm ends within at most $4H/\epsilon$, we can compute the total sample complexity.

$$K = \frac{4H}{\epsilon}\left(N + \sum_{i=1}^{n}(K_i + N)\right) = \tilde{\mathcal{O}}\left(\frac{S^4 O^4 A^4 (S^2 A^2 O^2 + SO^2) \times \text{poly}(H)}{\alpha^4 \epsilon^3}\right).$$

$\square$

### E.3 Proof for Theorem 5.2

**Theorem E.3.** *For both randomized and deterministic algorithms, there exists an instance of DEC-MDP wherein the regret scales at least as $\mathcal{O}(\sqrt{A^n T})$. This result underscores the limitation of achieving sample efficiency in algorithms for DEC-POMDP without imposing assumptions on the transition model, even with a memoryless policy.*

*Proof.* The proof for Theorem 5.2 proceeds straightforwardly. We consider a two-step DEC-MDP, commencing from an initial state $s_1$. For all $s \in \mathcal{S}$, we assume the reward function satisfies $r_{h,1}(s) = r_{h,2}(s) = \cdots = r_{h,n}(s)$ for all $h \in [2]$. Consequently, the entire DEC-MDP reduces to a multi-armed bandit problem. By leveraging a classic result on the lower bound of regret for the multi-armed bandit problem (Mannor and Tsitsiklis, 2004), it follows that for any randomized or deterministic algorithm, there exists an instance of the multi-arm bandit problem such that the regret is at least $\mathcal{O}(\sqrt{\tilde{A}T})$, where $\tilde{A}$ denotes the number of arms. Consequently, for any randomized or deterministic algorithm, there exists an instance of DEC-MDP such that the regret is at least $\mathcal{O}(\sqrt{A^n T})$. $\square$

## F POMDP WITH KNAPSACK CONSTRAINTS

### F.1 MODEL

We commence by formally defining the model. We consider the framework of tabular Partially Observable Markov Decision Processes (POMDPs), denoted as $(\mathcal{S}, \mathcal{A}, \mathcal{O}, H, \mathbb{T}, r, \mathcal{M})$, which extends to an episodic POMDP with a $d$-dimensional budget. Each component $\mathbf{M}_i$ of the budget vector $\mathbf{M}$ represents the total budget of the $i^{th}$ cost. At the onset of each episode, the agent is endowed with a budget $\mathbf{M}_1 = \mathbf{M} = (M, M, \ldots, M)$. During the $h^{th}$ step, the agent incurs a cost vector, thereby decrementing the total budget to $\mathbf{M}_{h+1} = \mathbf{M}_h - \mathbf{C}_h$. Subsequently, the budget for the $(h+1)^{th}$ step follows a transition probability $\mathbf{M}_{h+1,i} \sim \mathbb{T}_h(\cdot|\mathbf{M}_{h,i}, o_h, a_h)$. An episode concludes after $H$ steps or when the budget of any dimension $i$ reaches $0$. The primary objective of the agent is to maximize its cumulative reward $\sum_{k=1}^{K} \sum_{h=1}^{H} r_{k,h}$ over $K$ episodes. Furthermore, we impose the knapsack assumption on the cost:

**Assumption F.1.** *Both the budget $\mathbf{M}_i$ and the possible values of costs $\mathbf{C}_i$ are integral multiples of the unit cost $\frac{1}{m}$.*

We conceptualize the POMDP model as a factored Decentralized POMDP (DEC-POMDP). Initially, the policy class is defined as follows:

$$\Pi = \left\{\{\pi_h\}_{h=1}^{H} \mid \pi_h : (\mathcal{A} \times \mathcal{O} \times \mathcal{M}^d)^{H-1} \times \mathcal{O} \times \mathcal{M}^d \rightarrow \mathcal{A}\right\}.$$

We define the joint state space as $\tilde{\mathcal{S}} = \mathcal{S} \times \mathcal{M}$, where the tuple consists of the true state and the budget. We introduce $d$ dummy agents, where the local state of the $i^{th}$ dummy agent corresponds to the $i^{th}$ entry in the budget vector. The true agent is denoted as the $(d+1)^{th}$ agent. The state transition

of the $(d+1)^{\text{th}}$ agent follows $s_{h+1} \sim \mathbb{T}_{h,i}(\cdot \mid s_h, a_h)$, and its observation follows $o_h \sim \mathbb{O}_h(\cdot \mid s_h)$. The transition of the $i^{\text{th}}$ agent is given by: $\mathbf{M}_{h+1,i} \sim \mathbb{T}_h(\cdot \mid \mathbf{M}_{h,i}, o_h, a_h)$. Therefore, the transition of the joint state is:

$$(s_{h+1}, \mathbb{M}_{h+1}) \sim \left[ \prod_{m=1}^{d} \mathbb{T}_{h,m}(\cdot \mid M_{h,m}, o_h, a_h) \right] \mathbb{T}_h(\cdot \mid s_h, a_h).$$

The probability of trajectory $(o_1, a_1, \mathbf{M}_1, o_2, a_2, \mathbf{M}_2, \ldots, o_H, a_H, \mathbf{M}_H)$ for policy $\pi$ is defined as:

$$\mathbb{P}^\pi(\tau_H) = \sum_{s_1, s_2, \ldots, s_H} \mu_1(s_1) \mathbb{O}_1(o_1 \mid s_1) \pi_1(a_1 \mid o_1, \mathbf{M}_1) \mathbb{T}_{1,a_1}(s_2 \mid s_1) \prod_{i=1}^{d} \mathbb{T}_{1,i}(\mathbf{M}_{2,i} | \mathbf{M}_{1,i}, o_1, a_1) \times \cdots$$

$$\times \mathbb{O}(o_{H-1} \mid s_{H-1}) \pi_{H-1}(a_{H-1} \mid o_1, a_1, \mathbf{M}_1, \ldots, o_{H-2}, a_{H-2}, \mathbf{M}_{H-2}, o_{H-1}) \mathbb{T}_{H-1,a_{H-1}}(s_H \mid s_{H-1})$$

$$\times \prod_{i=1}^{d} \mathbb{T}_{H-1,i}(\mathbf{M}_{H,i} \mid \mathbf{M}_{H-1,i}, o_{H-1}, a_{H-1}) \times \mathbb{O}(o_H \mid s_H) \times \pi_{H-1}(a_H \mid o_1, a_1, \mathbf{M}_1, \ldots, a_{H-1}, \mathbf{M}_{H-1}, o_H).$$

The reward function is defined as (for all $h \in [H]$):

$$\tilde{r}_h(o_h, \mathbf{M}_h) = r_h(o_h) \quad \text{if } M_{h,i} > 0 \text{ for all } i \in [d].$$
$$= 0 \quad \text{else.}$$

Hence, we can model the POMDP setting with knapsack constraints as a DEC-POMDP with $d + 1$ agents. For the $(d+1)^{th}$ agent, $o$, $a$, and $s$ are defined as its observation, action, and state, respectively. We interpret $\mathbf{B}_i$ as both the individual state and the observation of the $i^{th}$ agent. The transition of the $i^{th}$ agent is defined as: $\mathbb{T}_{h,i}(\mathbf{M}_{h+1,i} | \mathbf{M}_{h,i}, o_h, a_h)$,

which is influenced by the observation and action of the $(d+1)^{th}$ agent. The actions of agents $1, 2, \ldots, d$ do not affect the model, hence we do not need to consider their actions. The reward function $\tilde{r}_h(o_h, \mathbf{M}_h)$ is a function of the observation of all individuals. Thus, the POMDPwk model can be formulated as a factored DEC-POMDP. The influence graph of POMDPwk is depicted in Figure 2. The maximum indegree is 1.

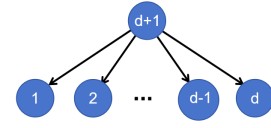

Figure 2: Influential graph for POMDP with constraints

## F.2 ALGORITHM

We can apply Algorithm 4.2 to the the setting of POMDP with constraints. We introduce Algorithm F.2. We define the complete trajectory $\boldsymbol{\tau}_H$ and trajectory $\tau_{H,i}$

---

**Algorithm 8** OMLE for POMDPwk

1: **Initialize**: $\mathcal{B}_{d+1}^1 = \{\hat{\theta}_{d+1} \in \Theta_{d+1} : \min_h \sigma_S(\mathbb{O}_i(\hat{\theta}_{d+1}) \geq \alpha)\}, \mathcal{B}_i = \{\Theta_i\}, \mathcal{D}_i = \{\}, \forall i \in [d]$.
2: **for** $k = 1, \ldots, K$ **do**
3: $\quad$ compute $(\theta_1^k, \theta_2^k, \ldots, \theta_n^k, \pi^k) = \arg\max_{\hat{\theta}_1 \in \mathcal{B}_1^k, \hat{\theta}_2 \in \mathcal{B}_2^k, \ldots, \hat{\theta}_n \in \mathcal{B}_n^k, \pi} V^\pi(\hat{\theta})$
4: $\quad$ follow $\pi^k$ to collect a trajectory $\boldsymbol{\tau}^k = (o_1^k, a_1^k, \mathbf{M}_1^k, \ldots, o_H^k, a_H^k, \mathbf{M}_H^k)$
5: $\quad$ add $(\pi^k, \boldsymbol{\tau}^k)$ into $\mathcal{D}_{d+1}$, and then update

$$\mathcal{B}_{d+1}^{k+1} = \Big\{ \hat{\theta}_{d+1} \in \mathcal{B}_{d+1}^1 : \sum_{(\pi, \boldsymbol{\tau}) \in \mathcal{D}_{d+1}} \log \mathbb{P}^\pi_{\hat{\theta}_{d+1}, d+1} \geq \max_{\theta'_{d+1}} \sum_{(\pi, \boldsymbol{\tau}) \in \mathcal{D}_{d+1}} \log \mathbb{P}^\pi_{\theta'_{d+1}, d+1} - \beta_{d+1} \Big\}$$

6: $\quad$ **for** $i = 1, \ldots, d$ **do**
7: $\quad\quad$ follow $\pi^k$ and model $\left[ \prod_{j=1}^{i} \mathbb{P}_{\theta_j^*, j} \right] \left[ \prod_{j=i+1}^{d} \mathbb{P}_{\theta_j^k} \right] \mathbb{P}^{\pi^k}_{\theta_{d+1}^k}$ to collect a trajectory $\boldsymbol{\tau}^k$.
8: $\quad\quad$ add $(\tau_i^k, \tau_{d+1}^k)$ into $\mathcal{D}_i$ and then update

$$\mathcal{B}_i^{k+1} = \Big\{ \hat{\theta} \in \mathcal{B}_i^1 : \sum_{(\tau_i, \tau_{d+1}) \in \mathcal{D}_i} \log \mathbb{P}_{\hat{\theta}, i}(\tau_i | \tau_{d+1}) \geq \max_{\theta'_i \in \Theta_i} \sum_{(\tau_i, \tau_{d+1}) \in \mathcal{D}_i} \log \mathbb{P}^\pi_{\theta'_i, i}(\tau_i | \tau_{d+1}) - \beta_i \Big\}$$

---

corresponding to agent $i \in [d+1]$ as

$$\boldsymbol{\tau}_H = (o_1, a_1, \mathbf{M}_1, \ldots, o_H, a_H, \mathbf{M}_H), \quad \tau_{H,d+1} = (o_1, a_1, \ldots, o_H, a_H), \quad \tau_{H,i} = (M_{1,i}, M_{2,i}, \ldots, M_{H,i}), \quad i \in [d].$$

The probability of individual $\mathbb{P}^{\pi}_{\theta_{d+1},d+1}\left(\tau_{d+1}|\{\tau_i\}_{i=1}^d\right)$ and $\mathbb{P}_{\theta_i,i}(\tau_i|\tau_{d+1})$, $i \in [d]$ is defined as:

$$\mathbb{P}^{\pi}_{\theta_{d+1},d+1}\left(\tau_{d+1} \mid \{\tau_i\}_{i=1}^d\right) = \sum_{s_1,s_2,\ldots,s_H} \mu_1(s_1)\mathbb{O}_1(o_1 \mid s_1)\pi_1(a_1 \mid o_1, \mathbf{M}_1)\mathbb{T}_{1,a_1}(s_2 \mid s_1) \times \cdots$$

$$\times \mathbb{O}(o_{H-1} \mid s_{H-1})\pi_{H-1}(a_{H-1} \mid o_1, a_1, \mathbf{M}_1, \ldots, o_{H-2}, a_{H-2}, \mathbf{M}_{H-2}, o_{H-1})\mathbb{T}_{H-1,a_{H-1}}(s_H \mid s_{H-1})$$

$$\times \mathbb{O}(o_H \mid s_H) \times \pi_{H-1}(a_H \mid o_1, a_1, \mathbf{M}_1, \ldots, o_{H-1}, a_{H-1}, \mathbf{M}_{H-1}, o_H),$$

$$\mathbb{P}_{\theta_i,i}(\tau_i \mid \tau_{d+1}) = \mathbb{T}_{1,i}(\mathbf{M}_{2,i} \mid \mathbf{M}_{1,i}, o_1, a_1) \times \mathbb{T}_{2,i}(\mathbf{M}_{3,i} \mid \mathbf{M}_{2,i}, o_2, a_2) \times \cdots \times \mathbb{T}_{H-1}(\mathbf{M}_{H,i} \mid \mathbf{M}_{H-1,i}, o_{H-1}, a_{H-1}).$$

### F.3 THEORETICAL GUARANTEE

We establish the following theoretical guarantee concerning the regret of Algorithm F.2.

**Theorem F.1.** *Let* $\beta_{d+1} = c(H(SA + SO)\log(TSAOH) + \log(Td/\delta))$ *and* $\beta_i = c(HSM^2m^2OA\log(TSMmAOH) + \log(dT/\delta))$ *for all* $i \in [d]$, *where* $c$ *is a constant. Then, with probability at least* $1 - \delta$, *Algorithm F.2 ensures the following inequality:*

$$Regret(k) \leq \tilde{\mathcal{O}}\left(\frac{S^2AO}{\alpha^2}\sqrt{k(S^2A + SO)} \times poly(H) + d(Mm)^2OA\sqrt{k(Mm)^2OA} \times poly(H)\right).$$

The proof of this theorem closely follows the derivation outlined in previous sections. Here, we present key steps while omitting detailed proofs for some of the lemmas for clarity.

**Lemma F.1.** *With probability at least* $1 - \delta$, *the following events hold:*

$$\max_{(\theta_{d+1},k)\in\Theta_{d+1}\times[T]} \sum_{t=1}^k \log\left(\frac{\mathbb{P}^{\pi^t}_{\theta_{d+1},d+1}(\tau^t_{d+1} \mid \{\tau^t_i\}_{i=1}^d)}{\mathbb{P}^{\pi^t}_{\theta^*_{d+1},d+1}(\tau^t_{d+1} \mid \{\tau^t_i\}_{i=1}^d)}\right) > c(H(SA + SO)\log(TSAOH) + \log(Td/\delta)),$$

$$\max_{(\theta_i,k)\in\Theta_i\times[T]} \sum_{t=1}^k \log\left(\frac{\mathbb{P}_{\theta_i,i}(\tau^t_i \mid \tau^t_{d+1})}{\mathbb{P}_{\theta^*_i,i}(\tau^t_i \mid \tau^t_{d+1})}\right) > c\left(H(SM^2m^2OA)\log(TSMmAOH) + \log(dT/\delta)\right), i \in [d].$$

**Lemma F.2.** *The following event holds with probability at least* $1 - \delta$ *for all* $\theta_i \in \Theta_i$, $i \in [d]$.

$$\sum_{t=1}^k \left(\sum_{\tau} \left|\mathbb{P}^{\pi^t}_{\theta_{d+1},d+1}(\tau_{d+1} \mid \{\tau_i\}_{i=1}^d) - \mathbb{P}^{\pi^t}_{\theta^*_{d+1},d+1}(\tau_{d+1} \mid \{\tau_i\}_{i=1}^d)\right| \prod_{i=1}^d \mathbb{P}_{\theta^*_i,i}(\tau_i \mid \tau_{d+1})\right)^2$$

$$\leq c\left(\sum_{t=1}^k \log\left(\frac{\mathbb{P}^{\pi^t}_{\theta_{d+1},d+1}(\tau^t_{d+1} \mid \{\tau^t_i\}_{i=1}^d)}{\mathbb{P}^{\pi^t}_{\theta^*_{d+1},d+1}(\tau^t_{d+1} \mid \{\tau^t_i\}_{i=1}^d)}\right) + H(SA + SO)\log(TSAOH) + \log(Td/\delta)\right),$$

$$\sum_{t=1}^k \left(\sum_{\tau} \left|\mathbb{P}_{\theta_i,i}(\tau_i \mid \tau_{d+1}) - \mathbb{P}_{\theta^*_i,i}(\tau_i \mid \tau_{d+1})\right| \left[\prod_{j=1}^{i-1} \mathbb{P}_{\theta^*_j,j}\right]\left[\prod_{j=i+1}^d \mathbb{P}_{\theta^t_j,j}\right] \mathbb{P}^{\pi^t}_{\theta_{d+1},d+1}(\tau_{d+1} \mid \{\tau_i\}_{i=1}^d)\right)^2$$

$$\leq c\left(\sum_{t=1}^k \log\left(\frac{\mathbb{P}_{\theta_i,i}(\tau^t_i \mid \tau^t_{d+1})}{\mathbb{P}_{\theta^*_i,i}(\tau^t_i \mid \tau^t_{d+1})}\right) + H(SM^2m^2OA)\log(TSMmOAH) + \log(dT/\delta)\right),$$

*where for all* $m \in [d]$, $\theta_m \in \Theta_m$, *we denote* $\mathbb{P}_{\theta_m,m}$ *as* $\mathbb{P}_{\theta_m,m}(\tau_m \mid \tau_{d+1})$.

**Definition F.1.** *For* $m \in [n-1]$, *we define the* $B-$*operator as follows:*

$$\mathbf{b}_{0,d+1} = \mathbb{O}_1\mu_1 \in \mathbb{R}^O,$$

$$\mathbf{B}_{h,d+1}(o,a) = \mathbb{O}_{h+1}\mathbb{T}_{h,a} \cdot diag(\mathbb{O}_h(o \mid \cdot))\mathbb{O}_h^{\dagger} \in \mathbb{R}^{O \times O}, \forall h \in [H],$$

$$\mathbf{b}_{h,i}(\tau_{h+1,i}) = \left[\prod_{h'=1}^h \mathbb{T}_{h',i}(M_{h'+1,i} \mid M_{h',i}, o_{h'}, a_{h'})\right],$$

$$\mathbf{b}_{h,d+1}(\tau_{h,d+1}) = \left[\prod_{h'=1}^h \mathbf{B}_{h',d+1}(o_{h'}, a_{h'})\right]\mathbf{b}_{0,d+1}.$$

**Lemma F.3.** *With probability at least $1 - \delta$, the following inequality holds true for all $h \in [H], k \in [T], i \in [d]$:*

$$\sum_{t=1}^{k-1} \sum_{\boldsymbol{\tau}_h} \pi^t(\boldsymbol{\tau}_h) \|\left(\mathbf{B}_{h,d+1}^k(o_h, a_h) - \mathbf{B}_{h,d+1}(o_h, a_h)\right) \mathbf{b}_{h-1,d+1}(\tau_{h-1,d+1})\|_1 \left[\prod_{i=1}^{d} \mathbf{b}_{h-1,i}(\tau_{h,i})\right] = \mathcal{O}\left(\frac{\sqrt{S}}{\alpha}\sqrt{k\beta_{d+1}}\right),$$

$$\sum_{t=1}^{k-1} \sum_{\boldsymbol{\tau}_h} \|\left(\mathbb{T}_{h-1,i}^k(M_{h,i} \mid M_{h-1,i}, o_{h-1}, a_{h-1}) - \mathbb{T}_{h-1,i}(M_{h,i} \mid M_{h-1,i}, o_{h-1}, a_{h-1})\mathbf{b}_{h-2,i}(\tau_{h-1,i}))\|_1 \times$$

$$\cdot \left[\prod_{j=1}^{i-1} \mathbf{b}_{h-1,j}(\tau_{h,j})\right] \left[\prod_{j=i+1}^{d} \mathbf{b}_{h-1,j}^t(\tau_{h,j})\right] \|\mathbf{b}_{h,d+1}^t(\tau_{h,d+1})\|_1 \pi_{d+1}^t(\boldsymbol{\tau}_h) = \mathcal{O}\left(\sqrt{k\beta_i}\right).$$

**Lemma F.4.** *the regret by the error of operator estimates is bounded by the following term:*

$$Regret(k) \le \sum_{t=1}^{k} \sum_{h=1}^{H-1} \sum_{\boldsymbol{\tau}_h} \frac{S^{1.5}}{\alpha} \|(\mathbf{B}_{h,d+1}^t - \mathbf{B}_{h,d+1})\mathbf{b}_{h-1,d+1}(\tau_{h-1,d+1})\|_1 \cdot \pi^t(\boldsymbol{\tau}_h) \cdot \left[\prod_{i=1}^{d} \mathbf{b}_{h-1,i}(\tau_{h,i})\right]$$

$$+ \sum_{t=1}^{k} \sum_{h=1}^{H-1} \sum_{i=1}^{d} \sum_{\boldsymbol{\tau}_h} \|(\mathbb{T}_{h-1,i}^t - \mathbb{T}_{h-1,i})\mathbf{b}_{h-2,i}\|_1 \left[\prod_{j=1}^{i-1} \mathbf{b}_{h-1,j}(\tau_{h,j})\right] \left[\prod_{j=i+1}^{d} \mathbf{b}_{h-1,j}^t(\tau_{h,j})\right] \|\mathbf{b}_{h,d+1}^t(\tau_{h,d+1})\|_1 \pi^t(\boldsymbol{\tau}_h),$$

*where for all $h \in [H]$, we denote $\mathbf{B}_{h,d+1}^t$ as $\mathbf{B}_{h,d+1}^t(o_h, a_h)$ and denote $\mathbf{B}_{h,d+1}$ as $\mathbf{B}_{h,d+1}(o_h, a_h)$. For all $i \in [d]$ and $h \in [H]$, we denote $\mathbb{T}_{h,i}^t$ as $\mathbb{T}_{h,i}^t(M_{h,i} \mid M_{h-1,i}, o_{h-1,a_{h-1}})$, and denote $\mathbb{T}_{h,i}$ as $\mathbb{T}_{h,i}(M_{h,i} \mid M_{h-1,i}, o_{h-1,a_{h-1}})$. Moreover, for all $h \in [H], i \in [d]$, we denote $\mathbf{b}_{h,i}$ as $\mathbf{b}_{h,i}(\tau_{h+1,i})$.*

**Theorem F.2.** *Let $\beta_{d+1} = c(H(SA + SO)\log(TSAOH) + \log(Td/\delta))$ and $\beta_i = c(HSM^2m^2OA\log(TSMmAOH) + \log(dT/\delta))$ for all $i \in [d]$, where $c$ is a constant. Then, with probability at least $1 - \delta$, Algorithm F.2 ensures the following inequality:*

$$Regret(k) \le \tilde{\mathcal{O}}\left(\frac{S^2AO}{\alpha^2}\sqrt{k(S^2A + SO)} \times poly(H) + d(Mm)^2OA\sqrt{k(Mm)^2OA} \times poly(H)\right).$$

*Proof.* The target is to bound the following $d + 1$ terms:

$$\sum_{t=1}^{k} \sum_{h=1}^{H-1} \sum_{\boldsymbol{\tau}_h} \|(\mathbf{B}_{h,d+1}^t - \mathbf{B}_{h,d+1})\mathbf{b}_{h-1,d+1}(\tau_{h-1,d+1})\|_1 \cdot \pi^t(\boldsymbol{\tau}_h) \cdot \left[\prod_{i=1}^{d} \mathbf{b}_{h-1,i}(\tau_{h,i})\right],$$

$$\sum_{t=1}^{k} \sum_{h=1}^{H-1} \sum_{\boldsymbol{\tau}_h} \|(\mathbb{T}_{h-1,i}^t - \mathbb{T}_{h-1,i})\mathbf{b}_{h-2,i}\|_1 \left[\prod_{j=1}^{i-1} \mathbf{b}_{h-1,j}(\tau_{h,j})\right] \left[\prod_{j=i+1}^{d} \mathbf{b}_{h-1,j}^t(\tau_{h,j})\right] \|\mathbf{b}_{h,d+1}^t(\tau_{h,d+1})\|_1 \pi^t(\boldsymbol{\tau}_h), i \in [d].$$

The condition we have is that with probability at least $1 - \delta$,

$$\sum_{t=1}^{k-1} \sum_{h=1}^{H-1} \sum_{\boldsymbol{\tau}_h} \|(\mathbf{B}_{h,d+1}^k - \mathbf{B}_{h,d+1})\mathbf{b}_{h-1,d+1}(\tau_{h-1,d+1})\|_1 \cdot \pi^t(\boldsymbol{\tau}_h) \cdot \left[\prod_{i=1}^{d} \mathbf{b}_{h-1,i}(\tau_{h,i})\right] = \mathcal{O}\left(\frac{\sqrt{S}}{\alpha}\sqrt{k\beta_{d+1}}\right),$$

$$\sum_{t=1}^{k-1} \sum_{h=1}^{H-1} \sum_{\boldsymbol{\tau}_h} \|(\mathbb{T}_{h-1,i}^k - \mathbb{T}_{h-1,i})\mathbf{b}_{h-2,i}\|_1 \left[\prod_{j=1}^{i-1} \mathbf{b}_{h-1,j}\right] \left[\prod_{j=i+1}^{d} \mathbf{b}_{h-1,j}^t\right] \|\mathbf{b}_{h,d+1}^t(\tau_{h,d+1})\|_1 \pi^t(\boldsymbol{\tau}_h) = \mathcal{O}\left(\sqrt{k\beta_i}\right), i \in [d].$$

Then, we can apply Eluder dimension lemma (Lemma **??**) to obtain that the target is bounded by

$$\sum_{t=1}^{k} \sum_{h=1}^{H-1} \sum_{\boldsymbol{\tau}_h} \|(\mathbf{B}_{h,d+1}^t - \mathbf{B}_{h,d+1})\mathbf{b}_{h-1,d+1}(\tau_{h-1,d+1})\|_1 \cdot \pi^t(\boldsymbol{\tau}_h) \cdot \left[\prod_{i=1}^{d} \mathbf{b}_{h-1i}(\tau_{h,i})\right] = \mathcal{O}\left(\frac{S^{1.5}H^2OA}{\alpha}\sqrt{k\beta_{d+1}}\right),$$

$$\sum_{t=1}^{k} \sum_{h=1}^{H-1} \sum_{\boldsymbol{\tau}_h} \|(\mathbb{T}_{h-1,i}^t - \mathbb{T}_{h-1,i})\mathbf{b}_{h-2,i}(\tau_{h-1,i})\|_1 \left[\prod_{j=1}^{i-1} \mathbf{b}_{h-1,j}(\tau_{h,j})\right] \left[\prod_{j=i+1}^{d} \mathbf{b}_{h-1,j}^t(\tau_{h,j})\right] \|\mathbf{b}_{d+1}^t(\tau_{h,d+1})\|_1 \pi^t(\boldsymbol{\tau}_h)$$

$$= \mathcal{O}\left((Mm)^2OA\sqrt{k\beta_i}\right), i \in [d].$$

Therefore, we can achieve that the regret is bounded by

$$R^k \le \tilde{\mathcal{O}}\left(\frac{S^2 AO}{\alpha^2}\sqrt{k(S^2 A + SO)} \times \text{poly}(H) + d(Mm)^2 OA\sqrt{k(Mm)^2 OA} \times \text{poly}(H)\right).$$

$\square$

### F.4 IMPROVEMENT TO ACHIEVE SHARPER BOUND

#### F.4.1 MOTIVATION

Algorithm F.2 does not appear to achieve the optimal sample complexity. The intuition is that dummy agents $1, 2, \ldots, d$ can observe their exact state. Therefore, they experience an MDP process. If we directly apply OMLE to the single-agent MDP setting, the algorithm yields the regret forms like $R^k \le \tilde{\mathcal{O}}(S^2 A\sqrt{k(S^2 A)} \times \text{poly}(H))$. The regret is scaled in $A^{1.5}$. However, if we apply the UCB-VI algorithm to the single-agent MDP, we can obtain: $R^k \le \tilde{\mathcal{O}}(H^2\sqrt{S^2 Ak})$. The regret is scaled in $A^{0.5}$. Therefore, by combining UCB-VI with OMLE, we might achieve a sharper bound on the regret.

We still use OMLE to estimate the model parameter $\theta_{d+1}$ for the $d + 1^{th}$ agent. For dummy agent $1, 2, \ldots, d$, we first use number of times each state tuple $(M_i, o, a, M_i')$ and $(M_i, o, a)$ is visited. Namely we have

$$N_h^k(M_i, o, a, M_i') = \sum_{t=1}^{k} 1_{(M_{h,i}^t, o_h^t, a_h^t, M_{h+1,i}^t)=(M_i,o,a,M_i')}, N_h^k(M_i, o, a) = \sum_{t=1}^{k} 1_{(M_{h,i}^t, o_h^t, a_h^t)=(M_i,o,a)},$$

$$\hat{\mathbb{T}}_h^k(M_i' \mid M_i, o, a) = \frac{N_h^k(M_i, o, a, M_i')}{N_h^k(M_i, o, a)}.$$

We define the bonus as follows:

$$b_h^k(M_i, o, a) = \sqrt{\frac{2Mm\ln(MmOAKHd/\delta)}{N_h^k(M_i, o, a)}} + \frac{2\ln(MmOAKHd/\delta)}{N_h^k((M_i, o, a))},$$

$$\beta_{d+1} = c(H(SA + SO)\log(TSAOH) + \log(Td/\delta)).$$

---

**Algorithm 9** OMLE for POMDPwk

1: **Initialize**: $\mathcal{B}_{d+1}^1 = \{\hat{\theta}_{d+1} \in \Theta_{d+1} : \min_h \sigma_S(\mathbb{O}_i(\hat{\theta}_{d+1}) \ge \alpha)\}, \mathcal{B}_i^1 = \{\Theta_i\}, \mathcal{D}_i = \{\}$, for all players $i \in [d]$.
2: **for** $k = 1, \ldots, K$ **do**
3:     compute $(\theta_1^k, \theta_2^k, \ldots, \theta_n^k, \pi^k) = \arg\max_{\hat{\theta}_1 \in \mathcal{B}_1^k, \hat{\theta}_2 \in \mathcal{B}_2^k, \ldots, \hat{\theta}_n \in \mathcal{B}_n^k, \pi} V^\pi(\hat{\theta})$
4:     follow $\pi^k$ to collect a trajectory $\boldsymbol{\tau}^k = (o_1^k, a_1^k, \mathbf{M}_1^k, \ldots, o_H^k, a_H^k, \mathbf{M}_H^k)$
5:     add $(\pi^k, \boldsymbol{\tau}^k)$ into $\mathcal{D}_{d+1}$, and then update

$$\mathcal{B}_{d+1}^{k+1} = \left\{\hat{\theta}_{d+1} \in \mathcal{B}_{d+1}^1 : \sum_{(\pi,\boldsymbol{\tau}) \in \mathcal{D}_{d+1}} \log \mathbb{P}_{\hat{\theta}_{d+1},d+1}^\pi \ge \max_{\theta'_{d+1}} \sum_{(\pi,\boldsymbol{\tau}) \in \mathcal{D}_{d+1}} \log \mathbb{P}_{\theta'_{d+1},d+1}^\pi - \beta_{d+1}\right\}$$

6:     **for** $i = 1, 2, \ldots, d$ **do**

$$\mathcal{B}_i^{k+1} = \left\{\hat{\theta}_i \in \mathcal{B}_i^1 : \sum_{M_{h+1,i}} \left|\mathbb{T}_{\hat{\theta}_{d+1},h}(M_{h+1,i} \mid M_{h,i}, o, a) - \hat{\mathbb{T}}_h^k(M_{h+1,i} \mid M_{h,i}, o, a)\right| \le b_h^k(M_{h,i}, o, a), \forall(h, o, a, M_{h,i})\right\}$$

---

### F.5 THEORETICAL GUARANTEE

**Theorem F.3.** *With probability at least $1 - \delta$, Algorithm F.4.1 guarantees the following bound on the regret:*

$$Regret(k) \le \tilde{\mathcal{O}}\left(\frac{S^2 AO}{\alpha^2}\sqrt{k(S^A + SO)} \times poly(H) + dHBm\sqrt{kOA}\right).$$

The proof of this theorem closely follows the derivation outlined in previous sections. Here, we present key steps while omitting detailed proofs for some of the lemmas for clarity.

**Lemma F.5.** *With probability at least $1 - \delta$, for all $o, a, M_{h,i}, k, h$, we have*

$$\sum_{M_{h+1,i}} \left| \mathbb{T}_{\theta_{d+1}^*, h}(M_{h+1,i} \mid M_{h,i}, o, a) - \hat{\mathbb{T}}_h^k(M_{h+1,i} \mid M_{h,i}, o, a) \right| \leq b_h^k(M_{h,i}, o, a).$$

*Proof.* Consider a fixed tuple $(o, a, M_{h,i}, h, k)$. Define $\mathcal{H}_{h,t}$ as the history starting from the beginning of episode 1 to step $h$ at episode $t$ (including step $h$, i.e. up to $s_h^t, a_h^t$). Define random variables $\{X_t\}_{t \geq 0}$ as

$$X_t = 1_{(o_h^t, a_h^t, M_{h,i}^t, M_{h+1,i}^t)=(o,a,M_{h,i},M_{h+1,i})} - 1_{(o_h^t, a_h^t, M_{h,i}^t)=(o,a,M_{h,i})} \mathbb{T}_{\theta_{d+1}^*, h}(M_{h+1,i} \mid M_{h,i}, o, a).$$

We now show that $\{X_t\}_{t \geq 0}$ is a martingale sequence adapted to filtration $\{\mathcal{H}_{h,t}\}_{t \geq 0}$. Note that $\mathbb{E}[X_t \mid \mathcal{H}_{h,t}] = 0$. We have $|X_t| \leq 1$. To use Azuma-Bernstein's inequality, we note that $\mathbb{E}\left[X_t^2 \mid \mathcal{H}_{h,t}\right]$ is bounded as:

$$\mathbb{E}\left[X_t^2 \mid \mathcal{H}_{h,t}\right] = 1_{(o_h^t, a_h^t, M_{h,i}^t)=(o,a,M_{h,i})} \mathbb{T}_{\theta_{d+1}^*, h}(M_{h+1,i} \mid M_{h,i}, o, a) \left(1 - \mathbb{T}_{\theta_{d+1}^*, h}(M_{h+1,i} \mid M_{h,i}, o, a)\right),$$

where we use the fact that the variance of a Bernoulli with parameter $p$ is $p(1-p)$. This means that

$$\sum_{t=1}^k \mathbb{E}\left[X_t^2 \mid \mathcal{H}_{h,t}\right] = N_h^k(M_i, o, a) \mathbb{T}_{\theta_{d+1}^*, h}(M_{h+1,i} \mid M_{h,i}, o, a) \left(1 - \mathbb{T}_{\theta_{d+1}^*, h}(M_{h+1,i} \mid M_{h,i}, o, a)\right).$$

Now we apply Bernstein's inequality on the martingale difference sequence $\{X_t\}_{t \geq 0}$, we have

$$\left| \sum_{t=1}^k X_t \right| \leq \sqrt{\frac{2 \mathbb{T}_{\theta_{d+1}^*, h}(M_{h+1,i} \mid M_{h,i}, o, a) \left(1 - \mathbb{T}_{\theta_{d+1}^*, h}(M_{h+1,i} \mid M_{h,i}, o, a)\right) \ln(1/\delta)}{N_h^k(M_{h,i}, o, a)}} + \frac{2 \ln(1/\delta)}{N_h^k(M_{h,i}, o, a)}$$

$$\leq \sqrt{\frac{2 \mathbb{T}_{\theta_{d+1}^*, h}(M_{h+1,i} \mid M_{h,i}, o, a) \ln(1/\delta)}{N_h^k(M_{h,i}, o, a)}} + \frac{2 \ln(1/\delta)}{N_h^k(M_{h,i}, o, a)}.$$

We apply a union bound over all $B_{h,i} \in \mathcal{B}_i, o \in \mathcal{O}, a \in \mathcal{A}, h \in [H], k \in [K], i \in [d]$, and we can achieve that

$$\left| \mathbb{T}_{\theta_{d+1}^*, h}(M_{h+1,i} \mid M_{h,i}, o, a) - \hat{\mathbb{T}}_h^k(M_{h+1,i} \mid M_{h,i}, o, a) \right|$$

$$= \left| \sum_{t=1}^k X_t \right| \leq \sqrt{\frac{2 \mathbb{T}_{\theta_{d+1}^*, h}(M_{h+1,i} \mid M_{h,i}, o, a) L}{N_h^k(M_{h,i}, o, a)}} + \frac{2L}{N_h^k(M_{h,i}, o, a)},$$

where we define $\ln(MmOAKHd/\delta)$. $\qquad\qquad\square$

**Corollary F.1.** *With probability at least $1 - \delta$,*

$$(\theta_1^*, \theta_2^*, \ldots, \theta_d^*) \in \bigcap_{k \in [K]} (\mathcal{B}_1^k \times \mathcal{B}_1^k \times \cdots \times \mathcal{B}_{d+1}^k).$$

**Lemma F.6.** *With probability at least $1 - \delta$, the following event holds.*

$$\max_{(\theta_{d+1}, k) \in \Theta_{d+1} \times [T]} \sum_{t=1}^k \log\left(\frac{\mathbb{P}_{\theta_{d+1}, d+1}^{\pi^t}(\tau_{d+1}^t \mid \{\tau_i^t\}_{i=1}^d)}{\mathbb{P}_{\theta_{d+1}^*, d+1}^{\pi^t}(\tau_{d+1}^t \mid \{\tau_i^t\}_{i=1}^d)}\right) > c(H(SA + SO) \log(TSAOH) + \log(Td/\delta)).$$

**Lemma F.7.** *We can obtain that the following event holds with probability at least $1 - \delta$ for all $\theta_{d+1} \in \Theta_{d+1}$ and $k \in [T]$.*

$$\sum_{t=1}^k \left( \sum_{\tau} \left| \mathbb{P}_{\theta_{d+1}, d+1}^{\pi^t}(\tau_{d+1} \mid \{\tau_i\}_{i=1}^d) - \mathbb{P}_{\theta_{d+1}^*, d+1}^{\pi^t}(\tau_{d+1} \mid \{\tau_i\}_{i=1}^d) \right| \left[ \prod_{i=1}^d \mathbb{P}_{\theta_i^*, i}(\tau_i \mid \tau_{d+1}) \right] \right)^2$$

$$\leq c \left( \sum_{t=1}^k \log\left(\frac{\mathbb{P}_{\theta_{d+1}, d+1}^{\pi^t}(\tau_{d+1}^t \mid \{\tau_i^t\}_{i=1}^d)}{\mathbb{P}_{\theta_{d+1}^*, d+1}^{\pi^t}(\tau_{d+1}^t \mid \{\tau_i^t\}_{i=1}^d)}\right) + H(SA + SO) \log(TSAOH) + \log(Td/\delta) \right).$$

**Lemma F.8.** *With probability at least $1 - \delta$, the following event holds:*

$$\sum_{t=1}^{k-1} \sum_{\boldsymbol{\tau}_h} \pi^t(\boldsymbol{\tau}_h) \|\left(\mathbf{B}_{h,d+1}^k(o_h, a_h) - \mathbf{B}_{h,d+1}(o_h, a_h)\right) \mathbf{b}_{h-1,d+1}(\tau_{h-1,d+1})\|_1 \left[\prod_{i=1}^d \mathbf{b}_{h-1,i}(\tau_{h,i})\right] = \mathcal{O}\left(\frac{\sqrt{S}}{\alpha}\sqrt{k\beta_{d+1}}\right).$$

**Theorem F.4.** *With probability at least $1 - \delta$, Algorithm F.4.1 guarantees the following bound on the regret:*

$$Regret(k) \leq \tilde{\mathcal{O}}\left(\frac{S^2 AO}{\alpha^2}\sqrt{k(S^A + SO)} \times poly(H) + dHBm\sqrt{kOA}\right).$$

*Proof.* Initially, using similar techniques as in the section of finding global optimal for factored DEC-POMDP, we have with probability at least $1 - \delta$, $\theta_{d+1} \in \bigcap_{k \in [K]} B_{d+1}^k$. We combine this with result in F.1, and we can obtain that with probability at least $1 - \delta$,

$$(\theta_1^*, \theta_2^*, \ldots, \theta_{d+1}^*) \in \bigcap_{k \in [K]} \left(\mathcal{B}_1^k \times \mathcal{B}_2^k \times \cdots \times \mathcal{B}_n^k\right).$$

Therefore, we can bind the regret as:

$$R^k = \sum_{t=1}^k V_{\boldsymbol{\theta}^*}^* - V_{\boldsymbol{\theta}^*}^{\pi^t} \leq \sum_{t=1}^k V_{\boldsymbol{\theta}^t}^{\pi^t} - V_{\boldsymbol{\theta}^*}^{\pi^t} \leq \sum_{t=1}^k \sum_{\boldsymbol{\tau}_H} H\left|\mathbb{P}_{\boldsymbol{\theta}^t}^{\pi^t}(\boldsymbol{\tau}_H) - \mathbb{P}_{\boldsymbol{\theta}^*}^{\pi^t}(\boldsymbol{\tau}_H)\right|.$$

According to the factorization condition, we have

$$\sum_{t=1}^k \sum_{\boldsymbol{\tau}_H} \left|\mathbb{P}_{\boldsymbol{\theta}^t}^{\pi^t}(\boldsymbol{\tau}_H) - \mathbb{P}_{\boldsymbol{\theta}^*}^{\pi^t}(\boldsymbol{\tau}_H)\right|$$

$$= \sum_{t=1}^k \sum_{\boldsymbol{\tau}_H} \left|\left[\prod_{i=1}^d \mathbb{P}_{\theta_i^t,i}(\tau_i \mid \tau_{d+1})\right] \mathbb{P}_{\theta_{d+1}^t,d+1}^{\pi^t}(\tau_{d+1} \mid \{\tau_i\}_{i=1}^d) - \left[\prod_{i=1}^d \mathbb{P}_{\theta_i^*,i}(\tau_i \mid \tau_{d+1})\right] \mathbb{P}_{\theta_{d+1}^*,d+1}^{\pi^t}(\tau_{d+1} \mid \{\tau_i\}_{i=1}^d)\right|.$$

Then we bind the term $\sum_{t=1}^k \sum_{\boldsymbol{\tau}_H} \left|\mathbb{P}_{\boldsymbol{\theta}^t}^{\pi^t}(\boldsymbol{\tau}_H) - \mathbb{P}_{\boldsymbol{\theta}^*}^{\pi^t}(\boldsymbol{\tau}_H)\right|$ via the following decomposition.

$$\sum_{t=1}^k \sum_{\boldsymbol{\tau}_H} \left|\left[\prod_{i=1}^d \mathbb{P}_{\theta_i^t,i}(\tau_i \mid \tau_{d+1})\right] \mathbb{P}_{\theta_{d+1}^t,d+1}^{\pi^t}(\tau_{d+1} \mid \{\tau_i\}_{i=1}^d) - \left[\prod_{i=1}^d \mathbb{P}_{\theta_i^*,i}(\tau_i \mid \tau_{d+1})\right] \mathbb{P}_{\theta_{d+1}^*,d+1}^{\pi^t}(\tau_{d+1} \mid \{\tau_i\}_{i=1}^d)\right|$$

$$\leq \sum_{t=1}^k \sum_{\boldsymbol{\tau}_H} \left|\prod_{i=1}^d \mathbb{P}_{\theta_i^t,i}(\tau_i \mid \tau_{d+1}) - \prod_{i=1}^d \mathbb{P}_{\theta_i^*,i}(\tau_i \mid \tau_{d+1})\right| \mathbb{P}_{\theta_{d+1}^t,d+1}^{\pi^t}(\tau_{d+1} \mid \{\tau_i\}_{i=1}^d)$$

$$+ \sum_{t=1}^k \sum_{\boldsymbol{\tau}_H} \left|\mathbb{P}_{\theta_{d+1}^t,d+1}^{\pi^t}(\tau_{d+1} \mid \{\tau_i\}_{i=1}^d) - \mathbb{P}_{\theta_{d+1}^*,d+1}^{\pi^t}(\tau_{d+1} \mid \{\tau_i\}_{i=1}^d)\right| \prod_{i=1}^d \mathbb{P}_{\theta_i^*,i}(\tau_i \mid \tau_{d+1})$$

$$\leq \sum_{t=1}^k \sum_{\boldsymbol{\tau}_H} \sum_{i=1}^d \left[\prod_{j=1}^{i-1} \mathbb{P}_{\theta_j^*,j}\right]\left|\mathbb{P}_{\theta_i^*,i}(\tau_i \mid \tau_{d+1}) - \mathbb{P}_{\theta_i^t,i}(\tau_i \mid \tau_{d+1})\right|\left[\prod_{j=i+1}^d \mathbb{P}_{\theta_j^*,j}\right] \mathbb{P}_{\theta_{d+1}^t,d+1}^{\pi^t}(\tau_{d+1} \mid \{\tau_i\}_{i=1}^d)$$

$$+ \sum_{t=1}^k \sum_{\boldsymbol{\tau}_H} \left|\mathbb{P}_{\theta_{d+1}^t,d+1}^{\pi^t}(\tau_{d+1} \mid \{\tau_i\}_{i=1}^d) - \mathbb{P}_{\theta_{d+1}^*,d+1}^{\pi^t}(\tau_{d+1} \mid \{\tau_i\}_{i=1}^d)\right|\left[\prod_{i=1}^d \mathbb{P}_{\theta_i^*,i}(\tau_i \mid \tau_{d+1})\right],$$

where for all $j \in [d], \theta_j \in \Theta_j$, let $\mathbb{P}_{\theta_j,j}$ denote $\mathbb{P}_{\theta_j,j}(\tau_j \mid \tau_{d+1})$. Moreover, for the term $\left|\mathbb{P}_{\theta_{d+1}^t,d+1}^{\pi^t}(\tau_{d+1} \mid \{\tau_i\}_{i=1}^d) - \mathbb{P}_{\theta_{d+1}^*,d+1}^{\pi^t}(\tau_{d+1} \mid \{\tau_i\}_{i=1}^d)\right|$, we have the folllowing inequality

$$\sum_{t=1}^k \sum_{\boldsymbol{\tau}_H} \left|\mathbb{P}_{\theta_{d+1}^t,d+1}^{\pi^t}(\tau_{d+1}|\{\tau_i\}_{i=1}^d) - \mathbb{P}_{\theta_{d+1}^*,d+1}^{\pi^t}(\tau_{d+1}|\{\tau_i\}_{i=1}^d)\right|\left[\prod_{i=1}^d \mathbb{P}_{\theta_i^*,i}(\tau_i|\tau_{d+1})\right]$$

$$= \sum_{t=1}^k \sum_{\boldsymbol{\tau}_H} \left\|\prod_{h=1}^{H-1} \mathbf{B}_{h,d+1}^t(o_{h,m}, a_{h,m})\mathbf{b}_{0,d+1}^t - \prod_{h=1}^{H-1} \mathbf{B}_{h,d+1}(o_{h,m}, a_{h,m})\mathbf{b}_{0,d+1}\right\|_1 \pi^t(\boldsymbol{\tau}_H) \cdot \left[\prod_{i=1}^d \mathbf{b}_{H-1,i}(\tau_{H,i})\right]$$

$$\leq \sum_{t=1}^k \sum_{h=1}^{H-1} \sum_{\boldsymbol{\tau}_h} \frac{S^{1.5}}{\alpha}\|(\mathbf{B}_{h,d+1}^t - \mathbf{B}_{h,d+1})\mathbf{b}_{h-1,d+1}(\tau_{h-1,d+1})\|_1 \cdot \pi^t(\boldsymbol{\tau}_h) \cdot \left[\prod_{i=1}^d \mathbf{b}_{h-1,i}(\tau_{h,i})\right].$$

Now we consider the term $\left| \mathbb{P}^{\pi^t}_{\theta^t_{d+1},d+1}(\tau_{d+1} \mid \{\tau_i\}_{i=1}^d) - \mathbb{P}^{\pi^t}_{\theta^*_{d+1},d+1}(\tau_{d+1} \mid \{\tau_i\}_{i=1}^d) \right|$. According to the selection of parameter in the implementation of the algorithm, we can derive that for all $o \in \mathcal{O}, a \in \mathcal{A}, M_{h,i} \in \mathcal{M}_i, k \in [K], h \in [H], i \in [d]$,

$$\sum_{M_{h+1,i}} \left| \mathbb{T}_{\theta^*_{d+1},h}(M_{h+1,i} \mid M_{h,i}, o, a) - \mathbb{T}_{\theta^k_{d+1},h}(M_{h+1,i} \mid M_{h,i}, o, a) \right|$$

$$\leq \sum_{M_{h+1,i}} \left| \mathbb{T}_{\theta^*_{d+1},h}(M_{h+1,i} \mid M_{h,i}, o, a) - \hat{\mathbb{T}}^k_h(M_{h+1,i} \mid M_{h,i}, o, a) \right|$$

$$+ \sum_{M_{h+1,i}} \left| \mathbb{T}_{\theta^k_{d+1},h}(M_{h+1,i} \mid M_{h,i}, o, a) - \hat{\mathbb{T}}^k_h(M_{h+1,i} \mid M_{h,i}, o, a) \right|$$

$$\leq 2b^k_h(M_{h,i}, o, a).$$

Moreover, we have

$$\sum_{t=1}^k \sum_{\boldsymbol{\tau}_H} \sum_{i=1}^d \left[ \prod_{j=1}^{i-1} \mathbb{P}_{\theta^*_j,j} \right] \left| \mathbb{P}_{\theta^*_i,i}(\tau_i \mid \tau_{d+1}) - \mathbb{P}_{\theta^t_i,i}(\tau_i \mid \tau_{d+1}) \right| \left[ \prod_{j=i+1}^d \mathbb{P}_{\theta^*_j,j} \right] \mathbb{P}^{\pi^t}_{\theta^t_{d+1},d+1}\left(\tau_{d+1} \mid \{\tau_i\}_{i=1}^d\right)$$

$$\leq \sum_{t=1}^k \sum_{h=1}^{H-1} \sum_{i=1}^d \sum_{\boldsymbol{\tau}_h} \|(\mathbb{T}^t_{h-1,i} - \mathbb{T}_{h-1,i})\mathbf{b}_{h-1,i}(\tau_{h-1,i})\|_1 \left[ \prod_{j=1}^{i-1} \mathbf{b}_{h-1,j} \right] \left[ \prod_{j=i+1}^d \mathbf{b}^t_{h-1,j} \right] \|\mathbf{b}^t_{h,d+1}(\tau_{h,d+1})\|_1 \pi^t(\boldsymbol{\tau}_h)$$

$$= \sum_{t=1}^k \sum_{h=1}^{H-1} \sum_{i=1}^d \sum_{M_{h,i}} \mathbb{E}_{M_{h-1,i},o_{h-1},a_{h-1}} \left[ \left| \mathbb{T}_{\theta^*_{d+1},h-1}(M_{h,i} \mid M_{h-1,i}, o_{h-1}, a_{h-1}) - \mathbb{T}_{\theta^t_{d+1},h-1}(M_{h,i} \mid M_{h-1,i}, o_{h-1}, a_{h-1}) \right| \right]$$

$$\leq \sum_{t=1}^k \sum_{h=1}^{H-1} \sum_{i=1}^d \mathbb{E}_{M_{h-1,i},o_{h-1},a_{h-1}} \left[ 2b^t_{h-1}(M_{h-1,i}, o_{h-1}, a_{h-1}) \right],$$

where for all $i \in [d]$, $\theta_i \in \Theta_i$, we denote $\mathbb{P}_{\theta_i,i}$ as $\mathbb{P}_{\theta_i,i}(\tau_i \mid \tau_{d+1})$. For all $h \in [H], i \in [d]$, we denote $\mathbf{b}_{h,i}$ as $\mathbf{b}_{h,i}(\tau_{h+1,i})$, and we denote $\mathbf{b}^t_{h,i}$ as $\mathbf{b}^t_{h,i}(\tau_{h+1,i})$.

We can then bound the summation of the bonus term with the following inequality:

$$\sum_{t=1}^k \sum_{M_i,o,a} \frac{1}{\sqrt{N^t_h(M_i,o,a)}} = \sum_{M_i,o,a} \sum_{r=1}^{N^k_h(M_i,o,a)} \frac{1}{\sqrt{r}} \leq \sum_{M_i,o,a} 2\sqrt{\sum_{M_i,o,a}} \leq \sqrt{MmOA \sum_{M_i,o,a} N^k_h(M_i,o,a)} = \sqrt{kMmOA}.$$

Therefore, we can deduce that

$$\sum_{t=1}^k \sum_{\boldsymbol{\tau}_H} \sum_{i=1}^d \left[ \prod_{j=1}^{i-1} \mathbb{P}_{\theta^*_j,j} \right] \left| \mathbb{P}_{\theta^*_i,i}(\tau_i \mid \tau_{d+1}) - \mathbb{P}_{\theta^t_i,i}(\tau_i \mid \tau_{d+1}) \right| \left[ \prod_{j=i+1}^d \mathbb{P}_{\theta^*_j,j} \right] \mathbb{P}^{\pi^t}_{\theta^t_{d+1},d+1}(\tau_{d+1} \mid \{\tau_i\}_{i=1}^d)$$

$$\leq \sum_{t=1}^k \sum_{h=1}^{H-1} \sum_{i=1}^d \mathbb{E}_{M_{h-1,i},o_{h-1},a_{h-1}} \left[ 2b^t_{h-1}(M_{h-1,i}, o_{h-1}, a_{h-1}) \right]$$

$$\leq \mathcal{O}\left( dH\sqrt{kOA}Mm \ln(MmOAKHd/\delta) \right).$$

We are only left to bound the term

$$\sum_{t=1}^k \sum_{h=1}^{H-1} \sum_{\boldsymbol{\tau}_h} \frac{S^{1.5}}{\alpha} \|(\mathbf{B}^t_{h,d+1} - \mathbf{B}_{h,d+1})\mathbf{b}_{h-1,d+1}(\tau_{h-1,d+1})\|_1 \cdot \pi^t(\boldsymbol{\tau}_h) \cdot \left[ \prod_{i=1}^d \mathbf{b}_{h-1,i}(\tau_{h,i}) \right].$$

The condition we have is that with probability at least $1 - \delta$,

$$\sum_{t=1}^{k-1} \sum_{h=1}^{H-1} \sum_{\boldsymbol{\tau}_h} \|(\mathbf{B}^k_{h,d+1} - \mathbf{B}_{h,d+1})\mathbf{b}_{h-1,d+1}(\tau_{h-1,d+1})\|_1 \cdot \pi^t(\boldsymbol{\tau}_h) \cdot \left[ \prod_{i=1}^d \mathbf{b}_{h-1,i}(\tau_{h,i}) \right] = \mathcal{O}\left( \frac{\sqrt{S}}{\alpha}\sqrt{k\beta_{d+1}} \right).$$

We similarly apply Eluder-Dimension lemma (Lemma **??**), and we can achieve that

$$\sum_{t=1}^{k}\sum_{h=1}^{H-1}\sum_{\boldsymbol{\tau}_h}\|(\mathbf{B}_{h,d+1}^{t}-\mathbf{B}_{h,d+1})\mathbf{b}_{h-1,d+1}(\tau_{h-1,d+1})\|_1\cdot\pi^t(\boldsymbol{\tau}_h)\cdot\left[\prod_{i=1}^{d}\mathbf{b}_{h-1,i}(\tau_{h,i})\right]=\mathcal{O}\left(\frac{S^{1.5}H^2OA}{\alpha}\sqrt{k\beta_{d+1}}\right).$$

Therefore, we can achieve that the regret is bounded by

$$R^k\leq\tilde{\mathcal{O}}\left(\frac{S^2AO}{\alpha^2}\sqrt{k(S^2A+SO)}\times\mathrm{poly}(H)+dHMm\sqrt{kOA}\right).$$

$\square$

