# OpenReview forum: "Provable Learning for DEC-POMDPs: Factored Models and Memoryless Agents"
_ICLR.cc/2025/Conference — Submitted to ICLR 2025_

### Official Review · Reviewer_Ssnx · 2024-10-30

**Soundness:** 3
**Presentation:** 3
**Contribution:** 3
**Rating:** 6
**Confidence:** 2

**Summary:**

The paper addresses the challenge of achieving sample efficiency in decentralized partially observable Markov decision processes (DEC-POMDPs). By proposing algorithms under a factored structure model and memoryless policies, the authors break the curse of dimensionality for certain scenarios.

**Strengths:**

The paper addresses the challenge of achieving sample efficiency in decentralized partially observable Markov decision processes (DEC-POMDPs). By proposing algorithms under a factored structure model and memoryless policies, the authors break the curse of dimensionality for certain scenarios.

**Weaknesses:**

1. The analysis for memoryless policies lacks applicability to DEC-POMDPs where agents require memory, making the solution scope somewhat narrow for environments needing historical data for decision-making. Suggesting possible extensions to the analysis for environments where agents rely on historical data for effective decision-making would broaden the solution's applicability.

2. Certain technical assumptions, like weakly revealing conditions, are critical to the sample efficiency claims but are not sufficiently discussed in terms of real-world feasibility. It would be helpful if the authors could provide additional discussion on the real-world feasibility of the weakly revealing condition.

**Questions:**

1. How does the proposed algorithm address potential scalability issues as the number of agents and their interaction complexity increase within the factored structure model?
2. How does the algorithm handle partial observability when the weakly revealing condition does not hold? Would additional assumptions be required to maintain sample efficiency?
3. In scenarios where agents use memory-based policies, what adaptations would be necessary to extend the sample-efficiency guarantees presented here?
4. Can the authors provide a comparative analysis of their approach against recent advances that incorporate centralized training with decentralized execution in DEC-POMDPs?

---

> ### Author Response · Authors · 2024-11-20
> **Rebuttal Part 1**
>
> We sincerely thank Reviewer Ssnx for their valuable feedback on our work. We have now addressed your questions and concerns as outlined below.
>
> - **Explanation for Increase Interaction of Factored Model:** As demonstrated in Theorem 4.2, when the maximum indegree of the vertices in the influential graph is $d$, the regret scales at least as $\Omega(\sqrt{A^{d+1}T})$. Our algorithm achieves regret of
> $$
> \tilde{\mathcal{O}}\Big(\alpha^{-2}S^{2}OA^{d+1}\sqrt{k(S^2A^{d+1}+SO)}\Big),
> $$
> which is already near-optimal in terms of the exponential dependence on $d$. Thus, when $d$ is large, deriving a sample-efficient algorithm under the factored model is impossible. As the number of interactions between agents increases, one possible solution is to apply our algorithmic framework under different, reasonable assumptions on the model. For example, we could explore the case where agents adopt memory-limited policies in the factored model with complex interactions, which would allow us to derive non-exponential regret.
>
> - **Explanation for Partially Observable Condition:**
>
>   Thank you for raising the valuable question regarding the weakly revealing condition of the partially observable model. We now provide a more detailed discussion on this condition.
>
>   - **Motivation and Feasibility:** Initially, we would like to point out that the weakly revealing condition is a commonly adopted assumption in works involving partially observable models (see [1,2,3,4,5]). It is also a standard assumption for Hidden Markov Models (HMMs), which can be seen as an uncontrolled version of POMDPs (see [6]). A direct intuition for understanding the weakly revealing condition, as provided in [2], is as follows: let $\nu_1, \nu_2 \in \Delta(\mathcal{S})$ denote two different probability vectors over the state space. We impose the assumption that $\mathbb{O}_h \nu_1 \neq \mathbb{O}_h \nu_2$ for all $h \in [H]$. In other words, we assume that for two different mixtures of states, we obtain two distinct mixtures of observations.
>
>     This assumption and intuition are quite feasible in real-world problems. For example, a prominent application of partially observable models is medical diagnosis, where we cannot directly observe the underlying condition but can only observe symptoms of the disease. In the mathematical modeling of medical diagnosis, it is reasonable to assume that two different diseases exhibit two distinct sets of symptoms; otherwise, it would be impossible to develop effective treatment methods.
>
>     To formalize this intuition in a more rigorous mathematical form for further derivation, we assume that the rank of the observation matrix is lower-bounded by $\alpha$. We can further show (Theorem 6 in [2]) that there exists a POMDP for which any algorithm requires at least $\Omega(\min\left\\{1/\alpha H, A^{H-1}\right\\})$ samples to learn an optimal policy. This result further demonstrates that the requirement on the rank of $\mathbb{O}_h$ is necessary.
>
>   - **Our Framework Works Beyond Weakly Revealing:** You have also raised a valuable question about the adaptation required when the weakly revealing condition does not hold. Initially, we can immediately see that it is impossible to derive an efficient algorithm if the observation provides no information about the state (see also Proposition 1 in [2]). Therefore, alternative assumptions on the observation model are required in the absence of the weakly revealing condition. Furthermore, our algorithmic framework is applicable to other conditions on partially observable models, including the low-rank future sufficiency condition ([7,8]), B-stability condition ([5]), and decodability condition ([9]). Thus, when the weakly revealing condition does not hold, we can still impose these alternative assumptions on the model to maintain sample efficiency.

---

> ### Author Response · Authors · 2024-11-20
> **Rebuttal Part 2**
>
> - **Adaptation for Memory-Base Policy:** *Our framework naturally extends to a memory-finite setting with length $L$, as outlined in lines 428-431 of our paper.*
>
>   Specifically, the framework can be readily adapted to scenarios where agents consider observations and actions from the preceding $L$ steps. In such cases, the policy class expands to:
>
>   $$
>   \left\\{ \left\\{ \otimes\_{m=1}^n \pi\_{h,m} \right\\}\_{h \in [H]} \mid \pi\_{h,m}: (\mathcal{O}\_m \times \mathcal{A}\_m)^{\min(h,L)-1} \times \mathcal{O}\_m \to \Delta(\mathcal{A}\_m) \right\\}.
>   $$
>
>   We modify the equation from lines 1789 to 1795 as follows:
>
>   $$
>   \mathbb{O}\_{h,i}(o\_{h,i} \mid s_h) = \sum\_{\\{ o\_{h,m} \\}\_{m \in [n] \setminus \\{i\\}}} \mathbb{O}\_h(\mathbf{o}\_h \mid s\_h),
>   $$
>
>   $$
>   \mathbb{T}\_{h,i,a\_{h,i},\dots,a\_{h+L-1,i}}(s\_{h+L} \mid s_h, o\_{h,i}, \dots, o\_{h+L-1,i}) = \sum_{\\{(o\_{h,m},a\_{h,m})\\}\_{m \in [n] \setminus \\{i\\}}, \dots, \\{(o\_{h+L-1,m},a\_{h+L-1,m})\\}\_{m \in [n] \setminus \\{i\\}}}
>   $$
>
>   $$
>   \sum\_{s\_{h+1,\dots,s\_{h+L-1}}} \prod\_{r=0}^{L-1} \left[ \prod\_{j \in [n] \setminus \\{i\\}} \pi\_{h+r,j}(a\_{h+r,j} \mid o\_{h+r,j}) \right] \frac{\mathbb{O}\_{h+r}(\mathbb{o}\_{h+r} \mid s\_{h+r})}{\mathbb{O}\_{h+r,i}(o\_{h+r,i} \mid s\_{h+r})} \mathbb{T}\_{h+r,\mathbf{a}\_{h+r}}(s\_{h+r+1} \mid s\_{h+r}).
>   $$
>
>   We can similarly modify Equation (20) from lines 1799 to 1804, and define the transition operators $ \mathbb{O}\_{h,m}(o\_{h,i}, o\_{h,m} \mid s_h) $ and
>
>   $$
>   \mathbb{T}\_{h,m,a\_{h,i},\dots,a\_{h+L-1,i},a\_{h,m},\dots,a\_{h+L-1,m}}(s\_{h+L} \mid s_h, o\_{h,i}, \dots, o\_{h+L-1,i}, o\_{h,m}, \dots, o\_{h+L-1,m}).
>   $$
>
>   In this case, the number of episodes needed to achieve local optimality is at most:
>
>   $$
>   K = \tilde{\mathcal{O}}( S^4 O^{4L} A^{4L} (S^2 A^{2L} O^{2L} + SO^2) \cdot \text{poly}(H) / (\alpha^4 \epsilon^3) ).
>   $$
>
>   Thus, in this case, we further require the assumption that the agent relies on past information for $L = \mathcal{O}(1)$ steps.
>
>
> - **Discussion on CTDE:** Thank you for raising this valuable question. We will now present our discussion on learning DEC-POMDPs with the CTDE method as follows:
>
>   The DEC-POMDP model and the learning of DEC-POMDPs using CTDE originally stem from [10]. Recent advances in CTDE include methods based on value function factorization, centralized critic methods (see [13,14]); parameter sharing (where each agent uses a different network to estimate the value function or policy, and all agents share the same networks, [11,12]); alternating learning (fixing the policy of all agents except one, and allowing the learning agent to converge, generating a best-response to the policies of other agents; this process continues until no agent can improve its policy further, resulting in a Nash equilibrium [15,16]); and addressing nonstationarity (see [17]).
>
>   Our work also aims to provide a CTDE algorithm, but from a theoretical perspective. Specifically, during the training process of our algorithm (Algorithm 1, 2, 3), we have access to the trajectory history of all agents and construct a confidence interval to estimate model parameters. In the planning step, we output independent policies for each agent, allowing them to select actions based on their own trajectories from previous steps.
>
>   Moreover, we aim to provide a theoretical explanation for empirical CTDE methods, including VDN, Q-MIX, and Q-PLEX. Specifically, all these approaches assume that the total value decomposition can be factorized into different components, with each component depending only on the trajectory of a finite subset of agents rather than all agents. In such cases, these algorithms achieve satisfactory experimental performance. Our work primarily focuses on the theoretical aspects, aiming to offer insights into these value factorization-based algorithms. Specifically, we aim to identify sufficient conditions for value decomposition and prove that sample-efficient guarantees exist under these conditions. To achieve this, we introduce a factored structure model and demonstrate that similar value factorization, as assumed by the VDN algorithm, exists within this model. We then develop a sample-efficient guarantee based on this model. We believe our work provides a theoretical foundation that complements recent empirical studies on DEC-POMDPs and CTDE.

---

> ### Author Response · Authors · 2024-11-20
> **Reference**
>
> **Reference**
>
> [1] Jin, Chi, et al. "Sample-efficient reinforcement learning of undercomplete pomdps." Advances in Neural Information Processing Systems 33 (2020): 18530-18539.
>
> [2] Liu, Qinghua, et al. "When is partially observable reinforcement learning not scary?." Conference on Learning Theory. PMLR, 2022.
>
> [3] Liu, Qinghua, Csaba Szepesvári, and Chi Jin. "Sample-efficient reinforcement learning of partially observable markov games." Advances in Neural Information Processing Systems 35 (2022): 18296-18308.
>
> [4] Liu, Qinghua, et al. "Optimistic mle: A generic model-based algorithm for partially observable sequential decision making." Proceedings of the 55th Annual ACM Symposium on Theory of Computing. 2023.
>
> [5] Chen, Fan, Yu Bai, and Song Mei. "Partially observable rl with b-stability: Unified structural condition and sharp sample-efficient algorithms." arXiv preprint arXiv:2209.14990 (2022).
>
> [6] Anandkumar, Animashree, Daniel Hsu, and Sham M. Kakade. "A method of moments for mixture models and hidden Markov models." Conference on learning theory. JMLR Workshop and Conference Proceedings, 2012.
>
> [7] Lingxiao Wang, Qi Cai, Zhuoran Yang, and Zhaoran Wang. Embed to control partially observed systems: Representation learning with provable sample efficiency. arXiv preprint arXiv:2205.13476, 2022.
>
> [8] Qi Cai, Zhuoran Yang, and Zhaoran Wang. Reinforcement learning from partial observation: Linear function approximation with provable sample efficiency. In International Conference on Machine Learning
>
> [9] Yonathan Efroni, Chi Jin, Akshay Krishnamurthy, and Sobhan Miryoosefi. Provable reinforcement learning with a short-term memory. arXiv preprint arXiv:2202.03983, 2022
>
> [10]D. S. Bernstein, R. Givan, N. Immerman, and S. Zilberstein. The complexity of decentralized control of Markov decision processes. Mathematics of Operations Research,
>
> [11]J. K. Gupta, M. Egorov, and M. Kochenderfer. Cooperative multi-agent control using deep reinforcement learning.
>
> [12]J. Foerster, I. A. Assael, N. De Freitas, and S. Whiteson. Learning to communicate with deep multi-agent reinforcement learning. Advances in Neural Information Processing Systems,
>
> [13]C. Amato. An introduction to decentralized training and execution in cooperative multi-agent reinforcement learning. arXiv preprint arXiv:2405.06161, 2024.
>
> [14]C. Yu, A. Velu, E. Vinitsky, J. Gao, Y. Wang, A. Bayen, and Y. Wu. The surprising effectiveness of PPO in cooperative multi-agent games. Advances in Neural Information Processing Systems
>
> [15]B. Banerjee, J. Lyle, L. Kraemer, and R. Yellamraju. Sample bounded distributed reinforcement learning for decentralized POMDPs. In Proceedings of the National Conference on Artificial Intelligence
>
> [16]K. Su, S. Zhou, J. Jiang, C. Gan, X. Wang, and Z. Lu. MA2QL: A minimalist approach to fully decentralized multi-agent reinforcement learning. In Proceedings of the International Conference on Autonomous Agents and Multiagent Systems
>
> [17]T. Willi, A. H. Letcher, J. Treutlein, and J. Foerster. COLA: Consistent learning with opponentlearning awareness. In Proceedings of the International Conference on Machine Learning, 2022.

---

> > ### Comment · Reviewer_Ssnx · 2024-11-22
> >
> > Thanks for the response, and I will keep my score.

---

### Official Review · Reviewer_kBu6 · 2024-11-01

**Soundness:** 4
**Presentation:** 2
**Contribution:** 3
**Rating:** 5
**Confidence:** 3

**Summary:**

This paper deals with synthesis of decentralized multi-agent policies in a POMDP framework through reinforcement learning. It observes that while multi-agent RL policies have empirically met with success, their theoretical performance bounds -- and the limits of their performance -- are largely unknown. To tackle this challenge, the paper proposes a series of results on sample complexity of policy synthesis, showing that, while it is generally provably burdensome to determine an optimal policy, doing so might be possible in particular subclasses of policies and/or environments.

**Strengths:**

The paper deals with an important topic -- planning for DEC-POMDPs -- and treats it refreshingly formally. It includes theoretical hardness bounds on sample complexity, a treatment of algorithms for global and local optimality, and a mention of a very important and interesting application. The paper truly appears to be a tour de force in the multi-agent planning community.

**Weaknesses:**

In short, I believe this is a wonderful paper, but inappropriate in size for ICLR. In a traditional understanding of the paper format, a paper should be largely self-contained without recourse to any supplementary material like an appendix. Indeed, along these lines, the ICLR guidelines state "reviewers are not required to read the appendix". However, in this case the appendix is 3 times longer than the paper: the paper itself is mostly just a list of results where there is no indication of how they are proven and the only example which could possibly illustrate the derived technical work appears in a total of two paragraphs.

I find it impossible to truly evaluate this paper without treating the appendix exactly as the "main" paper, but if that needs to be done, then the paper (which is actually 48 pages long) is simply attempting to "game" the strict page limit. Again, I strongly, strongly support the publication of this paper, just not in this venue.

**Questions:**

I have no questions. I find the paper theoretically valuable and very relevant to the scope of ICLR, but not appropriate for publication due to its format.

---

> ### Author Response · Authors · 2024-11-20
> **Rebuttal**
>
> We sincerely thank reviewer kBu6 on valuable feedback and support on our work. Here, we provide our reason for presenting the work with most of the technical proofs in the Appendix.
>
> **Our brief explanation is that placing the proof in the Appendix, while presenting the core result in the main paper, is a commonly adopted approach in theoretical works for this conference.** We will further elaborate on this as follows.
>
> We first apologize for placing the proof in the supplementary material and for any inconvenience this may have caused. We chose to include the proof in the Appendix because it is common practice in learning theory works to present complete proofs there. For example, several recent ICLR papers on reinforcement learning theory, such as [1, 2, 3, 4,5,6,7,8,9,10], follow this structure. In line with these precedents, we decided to present the main results in the main body of the paper and include the technical proofs in the Appendix.
>
> Additionally, based on feedback from other reviewers, it is possible to follow the main contribution and ideas of the paper by reading only the main body without needing to refer to the proofs in the supplementary material. Due to space limitations, we felt that highlighting the main result in the main text while leaving the detailed proofs in the Appendix would help readers better grasp the core contribution of the work, which is a standard approach for theoretical papers.
>
> We appreciate your valuable suggestion, and in future versions of the paper, we will include a sketch of our proof and key ideas in the main body, so that readers can more easily understand the proof by reading the supplementary material in detail.
>
>
> [1] Faster Last-iterate Convergence of Policy Optimization in Zero-Sum Markov Games (ICLR 2023), Shicong Cen, Yuejie Chi, Simon S. Du, Lin Xiao
>
> [2] Variance-Aware Sparse Linear Bandits (ICLR 2023),Yan Dai, Ruosong Wang, Simon S. Du
>
> [3]Learning Rationalizable Equilibria in Multiplayer Games (ICLR 2023), Yuanhao Wang, Dingwen Kong, Yu Bai, Chi Jin
>
> [4]Representation Learning for General-sum Low-rank Markov Games (ICLR 2023), Chengzhuo Ni, Yuda Song, Xuezhou Zhang, Chi Jin, Mengdi Wang
>
> [5]Amortila, Philip, et al. "Harnessing density ratios for online reinforcement learning." (ICLR 2024 Spotlight)
>
> [6] The Role of Coverage in Online Reinforcement Learning (ICLR-23) Tengyang Xie, Dylan J Foster, Yu Bai, Nan Jiang, Sham Kakade.
>
> [7] PARL: A Unified Framework for Policy Alignment in Reinforcement Learning., ICLR, 2024
>
> [8] Guo et al. Sample-Efficient Learning of POMDPs with Multiple Observations In Hindsight. ICLR, 2024.
>
> [9]Ch Ni, Y Song, X, Z Ding, C Jin, M Wang. Representation Learning for Low-rank General-sum Markov Games. ICLR 2023.
>
> [10]M Yin, M Wang, Y-X Wang. Offline Reinforcement Learning with Differentiable Function Approximation is Provably Efficient. ICLR 2023. [link]

---

> > ### Comment · Reviewer_kBu6 · 2024-11-26
> >
> > I appreciate the authors' reply. While I appreciate the effort, I do not think that any show of prior "counterexamples" can fix my concern. The conference guidelines, which should certainly supersede empirical examples to the contrary, essentially state that the paper should be self-contained ("reviewers are not required to read the appendix"), and in such a case I do not believe that the paper is readable as is. I appreciate that there might be other papers that suffer from the same issue and that got accepted (and I would have the same concern about those), but that fact does not help to evaluate this paper on the basis of its nominal 10 pages. I would like to keep my original score, and truly wish the authors all the best in continuing with this exciting and deep work.

---

### Official Review · Reviewer_p38d · 2024-11-03

**Soundness:** 3
**Presentation:** 4
**Contribution:** 3
**Rating:** 8
**Confidence:** 3

**Summary:**

In this paper, the authors consider decentralized multi-agent reinforcement learning in a partially observable setting (i.e., RL for DEC-POMDPs). They consider two different settings based on different simplifying assumptions: 1) the dynamics can be factored such that any agent's dynamics only (directly) depend on a subset of all the other agents, and 2) the agents have no memory. For both settings, the authors define a variation of 'Optimistic Maximum Likelihood Estimation' (OMLE), an RL algorithm for standard POMDPs. The authors prove these algorithms yield better regret bounds than is possible for general DEC-POMDPs. More precisely, for the factored setting, the regret of their algorithms scale exponentially with the maximum number of agents that influence a single agent (instead of the total number of agents), while for the memoryless setting, they show an $\epsilon$-optimal policy can be found in a polynomial number of samples.

**Strengths:**

* The paper discussed an interesting and relevant topic. Dealing with partial observability is a key problem for applying (MA)RL in the real world.
* Much existing MARL research is application-focused. This paper instead takes a more theoretical approach, which makes it more novel.
* The paper is well written: the problem settings are well explained and the authors provide both pseudocode and an intuitive explanation of their algorithms. As someone who is not an expert on MARL, I found the paper relatively easy to follow.

**Weaknesses:**

* The properties of a factored structure model are, in my opinion, not sufficiently explained. In particular, the factorization (Sec. 4.1) requires that the transition function for a single agent depends only on the *actions* of a set of other agents. In other words, there can be no dependency between the *states* of different agents. This should be explicitly mentioned in the text.
* Both the assumptions of a factored structure model and memoryless agents are strong and limit real-life applicability. However, I think it is justified to make such assumptions in a theoretical paper.
* The proofs in the paper are long (~25 pages) and dense. The authors give some intuition on how exponential dependencies are avoided, but it is hard to verify the results in more detail. However, I am not sure if the authors can realistically improve this.

**Questions:**

* You do not give a proof or citation for Thm. 3.1: where does this come from? Can you give any intuition as to why it holds?
* I am a bit confused by your application model (Sec. 4.4): could you explain it in more detail? In particular, I have the following question:
	* You note in App. E1 that episodes terminate after any agent exceeds its budget. It seems like this implies all agents can affect each other (i.e., by terminating the episode), and thus, your model cannot be factored. Can you explain why this is not the case?

---

> ### Author Response · Authors · 2024-11-20
> **Rebuttal Part 1**
>
> We sincerely thank Reviewer p38d for the very positive evaluation of our work, as well as for the valuable feedback and suggestions. Below are our explanations for your concerns:
>
> ## Explanation for the Weakness:
>
> - **Properties of Factored Model:** Thank you very much for your valuable suggestion. In the next version of our paper, we will provide additional explanation of the factored structure model by including the following paragraph:
>
>   We assume that the local state transition of each individual agent depends solely on the actions of other agents, with no dependency between the states of different agents. Notably, we would like to point out that our framework can also be extended to the factored setting, where the local state transition function for each agent depends on the local states, observations, and actions of other agents. For the sake of clarity, we have selected the factored model above as a representative case to illustrate our algorithmic framework.
>
> - **Motivation for Memoryless Agent and Factored Structure Model**
>
>   Thank you for your understanding and approval. We will further demonstrate our motivation for considering factored structure model and memoryless agent here:
>
>   - **Motivation for Memoryless Agent:** Since deriving a sample-efficient algorithm without any assumptions on the model is not feasible, we consider a widely studied case that is often examined when there are challenges in the general model. Works on memoryless agents include [1] and [2]. Notably, our algorithmic framework can also be extended to a memory-limited setting with length $L$, as outlined in lines 428-431 of our paper. Specifically, the framework can be easily adapted to scenarios where agents consider observations and actions from the preceding $L$ steps. In such cases, the policy class expands to:
>
>   $$
>   \left\{ \left\{ \otimes_{m=1}^n \pi_{h,m} \right\}_{h \in [H]} \mid \pi_{h,m}: (\mathcal{O}_m \times \mathcal{A}_m)^{\min(h,L)-1} \times \mathcal{O}_m \to \Delta(\mathcal{A}_m) \right\}.
>   $$
>
>   The number of episodes required to achieve an $\epsilon$-approximate local optimal policy is at most:
>
>   $$
>   K = \tilde{\mathcal{O}}(S^4 O^{4L} A^{4L} (S^2 A^{2L} O^{2L} + SO^2) \cdot \text{poly}(H) / (\alpha^4 \epsilon^3)).
>   $$
>
>   - **Motivation for Factored Model:** We consider the factored model for two main reasons. First, factored models have wide applicability in real-world scenarios. A prime example of a factored structure model is presented in [3], which describes a large production line with $d$ machines arranged sequentially. In this model, machine $i$ has $S_i$ possible states and $O_i$ possible observations. At each step, each machine is influenced only by its neighboring machines. Specifically, for all $2 \leq m \leq d-1$, we have $\mathsf{pa}(m) = \{m-1, m+1\}$, with $\mathsf{pa}(2) = \{1\}$ and $\mathsf{pa}(d) = \{d-1\}$. [4] also discusses a factored structure model in the context of robotics, where the transitions of different robotic body parts are relatively independent. As a result, the joint transition dynamics of these body parts can be expressed in a factored form.
>
>   Second, we consider the factored model because we aim to provide theoretical insights for empirical works on DEC-POMDPs, such as VDN and Q-MIX, which adopt the value decomposition assumption. Specifically, these approaches assume that the total value function can be decomposed into separate terms, with each term depending only on the trajectory of individual agents. Empirical studies have shown that, under this assumption, these algorithms perform well in practice. To offer a theoretical foundation for these empirical results, we aim to establish sufficient conditions for value decomposition and provide sample-efficient guarantees under these conditions. Consequently, we propose a factored model that inherently supports value decomposition and prove that it leads to non-exponential sample complexity.

---

> ### Author Response · Authors · 2024-11-20
> **Rebuttal Part 2**
>
> ## Explanation for the Questions:
>
> - **Proof of Theorem 3.1:** We sincerely apologize for the lack of detailed presentation in our submitted paper. The proof follows from a classical example from [5]. We initially omitted the proof because it is very similar to the proof of Theorem 5.2 and because we used the classical comb lock example, which is commonly employed as a counterexample in theoretical works on RL. We have now included the proof in detail in the modified version of our paper. Below is the detailed presentation of the proof:
>
>   We consider a two-step DEC-MDP, commencing from an initial state $s_1$. For all $s \in \mathcal{S}$, we assume the reward function satisfies $r_{h,1}(s) = r_{h,2}(s) = \cdots = r_{h,n}(s)$ for all $h \in [2]$. Consequently, the entire DEC-MDP reduces to a multi-armed bandit problem. By leveraging a classic result on the lower bound of regret for the multi-armed bandit problem [11], it follows that for any randomized or deterministic algorithm, there exists an instance of the multi-arm bandit problem such that the regret is at least $\mathcal{O}(\sqrt{\tilde{A}T})$, where $\tilde{A}$ denotes the number of arms. Consequently, for any randomized or deterministic algorithm, there exists an instance of DEC-MDP such that the regret is at least $\mathcal{O}(\sqrt{A^nT})$.
>
>   The hard instances can be intuitively understood as follows: The game starts from a fixed state $s_1$ in the first step. In the second step, all agents receive a high reward in one particular state (referred to as the "good state"), and they receive no reward in any other state. The state transitions to the good state from $s_1$ only when all agents take a specific joint action. However, since the agents do not know the correct joint action, they must explore the entire action space, which grows exponentially with $n$. As a result, the regret also grows exponentially with $n$.
>
> - **Factorization in POMDP with Knapsack Constraints:** We sincerely apologize again for the lack of detail in our paper. We will now provide a more detailed explanation of the factored property of POMDPs with Knapsack constraints.
>
>   We define the joint state space as $\tilde{\mathcal{S}} = \mathcal{S} \times \mathcal{M}$, where the tuple consists of the true state and the budget. We introduce $d$ dummy agents, where the local state of the $i^{\text{th}}$ dummy agent corresponds to the $i^{\text{th}}$ entry in the budget vector. The true agent is denoted as the $(d+1)^{\text{th}}$ agent. The state transition of the $(d+1)^{\text{th}}$ agent follows $s\_{h+1} \sim \mathbb{T}\_{h,i}(\cdot \mid s_h, a_h)$, and its observation follows $o_h \sim \mathbb{O}_h(\cdot \mid s_h)$. The transition of the $i^{\text{th}}$ agent is given by: $\mathbf{M}\_{h+1,i} \sim \mathbb{T}\_h(\cdot \mid \mathbf{M}\_{h,i}, o_h, a_h)$. Therefore, the transition of the joint state is:
>
>   $$
>   (s\_{h+1}, \mathbb{M}\_{h+1}) \sim \left[\prod\_{m=1}^d \mathbb{T}\_{h,m}(\cdot \mid M\_{h,m}, o\_h, a\_h)\right] \mathbb{T}\_h(\cdot \mid s\_h, a\_h).
>   $$
>
>   To represent the termination of the episode, we define the reward function as follows: $\tilde{r}\_h(o\_h, \mathbf{M}\_h)=r_h(o_h)$ if
>   $M\_{h,i} > 0 \text{ for all } i \in [d]$, and otherwise $\tilde{r}\_h(o\_h, \mathbf{M}\_h)=0$.
>
>
>   That is, the agent receives no reward if the budget reaches zero. Thus, the model maintains a factored structure. Notably, there is a slight difference from the formal definition in Section 4.1, as the transition now also depends on the observation. However, as mentioned earlier, we can still apply our algorithmic framework with only minor modifications.
>
>
>
> [1] Kara, A. and Yuksel, S. (2022). Near optimality of memory-finite feedback policies in partially observed markov decision processes. Journal of Machine Learning Research,
>
> [2]Kara, A. D. and Yüksel, S. (2023). Convergence of memory-finite q learning for pomdps and near optimality of learned policies under filter stability. Mathematics of Operations Research
>
> [3] Osband, I. and Van Roy, B. (2014). Near-optimal reinforcement learning in factored mdps. Advances in Neural Information Processing Systems
>
> [4]Chen, X. , Hu, J. , Li, L. and Wang, L. (2020). Efficient reinforcement learning in factored mdps with application to constrained rl. arXiv preprint arXiv:2008.13319
>
> [5]Mannor, S. and Tsitsiklis, J. N. (2004). The sample complexity of exploration in the multi-armed bandit problem. Journal of Machine Learning Research, 5 623-648

---

> > ### Comment · Reviewer_p38d · 2024-11-25
> > **Response to rebuttal**
> >
> > I’d like to thank the authors for their detailed responses. I am personally a bit confused by the authors comment that their approach generalises to settings where the dynamics also depend on the local states and observations of other agents: I would have expected this to be a much more prominent part of the original paper. Moreover, I think providing this information only in the discussion phase does not give reviewers sufficient time to check whether or not the claims are correct.
> >
> > Regardless, I feel that the edits make to the paper make it sufficiently clear what assumptions are made, and am satisfied with the answers given to my questions. Thus, I’ll keep my score.

---

### Official Review · Reviewer_mApi · 2024-11-03

**Soundness:** 4
**Presentation:** 3
**Contribution:** 3
**Rating:** 8
**Confidence:** 3

**Summary:**

This paper presents learning algorithms for decentralized partially observable Markov decision processes (DEC-POMDPs) with theoretical guarantees. Starting with a hardness result for general DEC-POMDPs, the authors argue that achieving sample efficiency in this broad setting is challenging. Therefore, they focus on two specific cases: (1) factored models, where the state space decomposes into individual components, and (2) memoryless policies based solely on current observations. Within these frameworks, the authors propose various sample-efficient algorithms that leverage the structural assumptions to achieve global or local optimality, with a detailed analysis of sample efficiency.

**Strengths:**

The paper is well-written, and the proposed algorithms appear novel, particularly within the DEC-POMDP setting. The theoretical analysis supporting the algorithms is thorough, with rigorous arguments on sample efficiency and optimality.

**Weaknesses:**

The work appears to only partially achieve the stated goals. Although I understand that assumptions are necessary to ensure sample efficiency, the choice of these two structures (factored models and memoryless policies) as primary focuses lacks clear justification. In addition, other DEC-POMDP frameworks exist (e.g., models where the reward depends on system states or state-action pairs). It would strengthen the paper by discussing how the proposed algorithms and sample-efficiency results might generalize or adapt to such alternative models.

**Questions:**

1) See above, how general are the factored and memoryless policy structures discussed in the paper? Are they widely applicable to DEC-POMDPs, or is their utility restricted to specific subclasses?
2) The authors connect local optimality with Nash equilibrium, but this connection seems unconventional. Typically, Nash equilibrium implies that no individual agent can improve their reward by unilaterally altering their strategy. However, the paper’s definition of local optimality appears to rely on a value function representing the overall reward of all agents.
3) Sample complexity is generally interpreted as the number of steps required to reduce regret or cost. Could the authors clarify the connection between Theorem 3.1’s result and sample complexity? Additionally, how does Theorem 4.1 demonstrate improved sample efficiency?
4) Some multi-agent reinforcement learning studies address the exponential action space growth using mean-field control techniques. It might benefit the paper to discuss these works.
5) Minor point: I couldn’t locate the proof for Theorem 3.1.
6) Minor point: Is the “an” notation in Definition 4.2 intended to refer to “pa” in Definition 4.1? Please clarify for consistency.

---

> ### Author Response · Authors · 2024-11-20
> **Rebuttal Part 1**
>
> We sincerely thank Reviewer ntUa for valuable feedback on our work. We now address your questions as follows:
>
> ## Explanation for the Weakness
>
> - **Justification for Memoryless Policy and Factored Model:**
>   *Memoryless policies and factored structure settings are widely studied models in the RL literature, with broad practical applications.*
>
>   We apologize for the lack of clarity in presenting the justification for these two assumptions. Our motivations are explained in the introduction and other sections of the paper. Thank you for your valuable feedback. In the next version, we will emphasize these motivations more clearly.
>   - **Memoryless Policy Model:** The motivation for considering a memoryless policy stems from its widespread use in models where studying the most general partially observable model is challenging. For instance, [1] explores learning optimal memoryless policies for POMDPs by approximating the belief model through discretizing the belief space. [2] provides convergence analysis for a Q-learning algorithm designed for POMDPs with memoryless policies. Therefore, we adopt the memoryless model as a natural choice, following prior studies in the field.
>   - **Factored Structure Model:** Our choice of the factored structure model is inspired by empirical works on DEC-POMDPs, such as VDN and Q-MIX, which all adopt the value decomposition assumption. Specifically, these approaches assume that the total value function can be decomposed into separate terms, with each term depending only on the trajectory of individual agents. Empirical studies have shown that under this assumption, algorithms perform well in practice. To provide a theoretical foundation for these empirical results, we aim to establish sufficient conditions for value decomposition and demonstrate sample-efficient guarantees under these conditions. As a result, we propose a factored model that inherently supports value decomposition and prove that it leads to non-exponential sample complexity.
>
> - **Explanation for Other DEC-POMDP Framework:**
>   *Our framework is applicable to a wide range of DEC-POMDP models, and we present our algorithms within the current model as representative examples, as this approach is more accessible for readers.*
>
>   Thank you for your valuable suggestion regarding the discussion of the DEC-POMDP model. Our algorithmic framework indeed applies to other variations of the DEC-POMDP model, as you have pointed out. For instance, if the reward function depends on the agent's observation and action (i.e., $r_{i,h} = r_{i,h}(o_{i,h}, a_{i,h})$), or if it depends on the state (i.e., $r_{i,h} = r_{i,h}(s_h)$), or even if it depends on the action, observation, and state (i.e., $r_{i,h} = r_{i,h}(s_h, a_{i,h}, o_{i,h})$), our algorithmic framework still applies. Furthermore, alternative definitions of the transition function are possible. For example, if the transition function depends on the state, observation, and action (i.e., $T_h(s_{h+1} | s_h, \mathbf{a}_h, \mathbf{o}_h)$), our algorithm remains valid in these settings as well.
>
>   Given the numerous possible definitions for the model, we have chosen the current model definition as a representative example to illustrate our algorithm. This choice is made for clarity and ease of understanding for readers, and it aligns with standard approaches in studies of partially observable reinforcement learning. For example, in works on POMDPs such as [3, 4], the reward function is typically defined as depending only on the observation. Following these common definitions in the field of partially observable RL, we similarly define the reward function in our framework.

---

> ### Author Response · Authors · 2024-11-20
> **Rebuttal Part 2**
>
> ## Explanation for the Questions
>
> - **Generality of Our Proposed Model:** *As we have explained previously, our algorithmic framework is not limited to the definition presented in the main paper.*
>   We will provide a more detailed explanation as follows:
>   - **Memoryless policy:** Our framework naturally extends to a memory-finite setting with length $L$, as outlined in lines 428-431 of our paper. Specifically, the framework can be readily adapted to scenarios where agents consider observations and actions from the preceding $L$ steps. In such cases, the policy class expands to:
>     $$
>     \left\\{ \left\\{ \otimes\_{m=1}^n \pi\_{h,m} \right\\}\_{h \in [H]} \mid \pi\_{h,m}: (\mathcal{O}\_m \times \mathcal{A}\_m)^{\min(h,L)-1} \times \mathcal{O}\_m \to \Delta(\mathcal{A}\_m) \right\\}.
>     $$
>
>     We modify the equation from lines 1789 to 1795 as follows:
>
>     $$
>     \mathbb{O}\_{h,i}(o\_{h,i} \mid s\_h) = \sum\_{\\{ o\_{h,m} \\}_{m \in [n] \setminus \\{i\\}}} \mathbb{O}\_h(\mathbf{o}\_h \mid s\_h),
>     $$
>
>   $$
>         \mathbb{T}\_{h,i,a\_{h,i},\dots,a\_{h+L-1,i}}(s\_{h+L} \mid s\_h, o\_{h,i}, \dots, o\_{h+L-1,i}) = \sum\_{\\{(o\_{h,m},a\_{h,m})\\}_{m \in [n] \setminus \\{i\\}}, \dots, \\{(o\_{h+L-1,m},a\_{h+L-1,m})\\}\_{m \in [n] \setminus \\{i\\}}}
>         $$
>
>
>     $$
>     \sum\_{s\_{h+1,\dots,s\_{h+L-1}}} \prod\_{r=0}^{L-1} \left[ \prod\_{j \in [n] \setminus \\{i\\}} \pi\_{h+r,j}(a\_{h+r,j} \mid o\_{h+r,j}) \right] \frac{\mathbb{O}\_{h+r}(\mathbb{o}\_{h+r} \mid s\_{h+r})}{\mathbb{O}\_{h+r,i}(o\_{h+r,i} \mid s\_{h+r})} \mathbb{T}\_{h+r,\mathbf{a}\_{h+r}}(s\_{h+r+1} \mid s\_{h+r}).
>     $$
>
>     We can similarly modify Equation (20) from lines 1799 to 1804, and define the transition operators $\mathbb{O}\_{h,m}(o\_{h,i}, o\_{h,m} \mid s\_h)$ and
>
>     $$
>     \mathbb{T}\_{h,m,a\_{h,i},\dots,a\_{h+L-1,i},a\_{h,m},\dots,a\_{h+L-1,m}}(s\_{h+L} \mid s_h, o\_{h,i}, \dots, o\_{h+L-1,i}, o\_{h,m}, \dots, o\_{h+L-1,m}).
>     $$
>
>     In this case, the number of episodes needed to achieve local optimality is at most:
>
>     $$
>     K = \tilde{\mathcal{O}}\left( S^4 O^{4L} A^{4L} (S^2 A^{2L} O^{2L} + SO^2) \cdot \text{poly}(H) / (\alpha^4 \epsilon^3) \right).
>     $$

---

> ### Author Response · Authors · 2024-11-20
> **Rebuttal Part 3**
>
> - **Factored Structure Model:** Our algorithmic framework accommodates multiple definitions of the reward function. For instance, if the reward function depends on the agent's observation and action (i.e., $r_{i,h} = r_{i,h}(o_{i,h}, a_{i,h})$), or if it depends on the state (i.e., $r_{i,h} = r_{i,h}(s_h)$), or even if it depends on the action, observation, and state (i.e., $r_{i,h} = r_{i,h}(s_h, a_{i,h}, o_{i,h})$), our framework still applies.
>
>     Moreover, our framework is also flexible with respect to the definition of the transition function. We consider the following transition factorization:
>
>     $$
>     \mathbb{T}\_h(\mathbf{s}^\prime \mid \mathbf{s}, \mathbf{a}) = \prod\_{m=1}^n \mathbb{T}\_{h,m}(s^\prime \mid s\_m, s\_{\mathsf{pas}(m)}, a_m, a\_{\mathsf{paa}(m)}),
>     $$
>
>     where $\mathsf{pas}(m) \subset [n]$ represents the set of agents whose local states influence the transition of agent $ m $, and $\mathsf{paa}(m) \subset [n]$ represents the set of agents whose actions influence the transition of agent $m$.
>
>     Next, we define $\max_{m \in [n]} |\mathsf{pas}(m)| := d_s = \mathcal{O}(1)$ and $d_a = \max_{m \in [n]} |\mathsf{paa}(m)| = \mathcal{O}(1)$. With these definitions, the regret is bounded as follows in a DAG setting:
>
>     $$
>     \text{Regret}(k) \leq \tilde{\mathcal{O}}\left( \alpha^{-2} S^{d_s+2} O A^{d_a+1} \sqrt{k(S^{d_s+2} A^{d_a+1} + SO)} \right).
>     $$
>
>     Furthermore, we can consider the case where the transition is influenced by the local states, observations, and actions of other agents. In this case, we have:
>
>     $$
>     \mathbb{T}\_h(\mathbf{s}^\prime \mid \mathbf{s}, \mathbf{a}) = \prod\_{m=1}^n \mathbb{T}_{h,m}(s^\prime \mid s_m, s\_{\mathsf{pas}(m)}, a_m, a\_{\mathsf{paa}(m)}, o\_{\mathsf{pao}(m)}),
>     $$
>
>     where $\mathsf{pao}(m) \subset [n]$ represents the set of agents whose observations influence the transition of agent $ m $. In this case, the regret is bounded by:
>
>     $$
>     \text{Regret}(k) \leq \tilde{\mathcal{O}}\left( \alpha^{-2} S^{d_s+2} O^{d_o+1} A^{d_a+1} \sqrt{k(S^{d_s+2} A^{d_a+1} O^{d_o} + SO)} \right),
>     $$
>
>     where we define $\max_{m \in [n]} |\mathsf{pao}(m)| := d_o = \mathcal{O}(1)$.
>
>     Moreover, we can also consider the case where the observation is influenced by the actions of other agents. Specifically, we have:
>
>     $$
>     \mathbb{O}\_h(\mathbf{o} \mid \mathbf{s}, \mathbf{a}) = \prod\_{m=1}^n \mathbb{O}\_{h,m}(o_m \mid s_m, a\_{\mathsf{paao}(m)}),
>     $$
>
>     where $\mathsf{paao}(m) \subset [n]$ represents the set of agents whose actions influence the observation of agent $ m $. In this case, the regret is bounded by:
>
> $$
> \text{Regret}(k) \leq \tilde{\mathcal{O}}\left( \alpha^{-2} S^{d_s+2} O^{d_o+1} A^{d_a+1} \sqrt{k(S^{d_s+2} A^{d_a+1} O^{d_o} + SO)} \right),
> $$
>
> where we define $\max_{m \in [n]} |\mathsf{pao}(m)| := d_o = \mathcal{O}(1)$.
>
> Moreover, we can also consider the case where the observation is influenced by the actions of other agents. Specifically, we have:
>
> $$
> \mathbb{O}\_h(\mathbf{o} \mid \mathbf{s}, \mathbf{a}) = \prod\_{m=1}^n \mathbb{O}\_{h,m}(o_m \mid s_m, a\_{\mathsf{paao}(m)}),
> $$
>
> where $\mathsf{paao}(m) \subset [n]$ represents the set of agents whose actions influence the observation of agent $ m $. In this case, the regret is bounded by:
>
> $$
>         \text{Regret}(k) \leq \tilde{\mathcal{O}}\left( \alpha^{-2} S^{d_s+2} O^{d_o+1} A^{d_a+1} \sqrt{k(S^{d_s+2} A^{d_a+1} O^{d_o} + SOA^{d_{ao}})} \right),
> $$
>
> where we define $d_{ao} = \max_{m \in [n]} |\mathsf{paao}(m)| = \mathcal{O}(1)$. fine $\max_{m \in [n]} |\mathsf{paao}(m)| := d_{ao} = \mathcal{O}(1)$.
>
> There are numerous definitions of the model, and we have chosen our current setup for the sake of clarity. Specifically, our assumption is easier for readers to follow, aligns with the standard definitions used in studies on partially observable reinforcement learning, and better illustrates the motivation we discussed earlier.

---

> ### Author Response · Authors · 2024-11-20
> **Rebuttal Part 4**
>
> - **Explanation for the Nash Equilibria:** We first apologize for any confusion caused by our discussion in the main text and will explain the matter in more detail. We establish a connection between local optimality and Nash equilibria, as achieving local optimality is equivalent to achieving Nash equilibrium in a DEC-POMDP with an alternative formulation. Since DEC-POMDP is a special case of a potential game, our algorithm can be further extended to achieve Nash equilibria in general potential games with partially observable contexts.
>  In game theory, particularly in multi-agent games (MG) and partially observable multi-agent games (POMG), Nash equilibria are defined as the points where no agent can improve its own value function by changing its policy.
>
> In our setting, each agent receives a reward $ r_{i,h}(o_{i,h}) $ at each time step, and the goal of DEC-POMDP is to maximize the expected total accumulated rewards for all agents. An alternative definition of DEC-POMDP, as seen in previous studies, is that in each step, all agents receive the same value function:
> $$
> r\_{h}(\mathbf{o}\_h) = \sum\_{m=1}^n r\_{h,m}(o\_{h,m}).
> $$
> In this case, all agents share the same value function
> $$
> V\_{1,1}^{\boldsymbol{\pi}} = V\_{2,1}^{\boldsymbol{\pi}} = \cdots = V\_{n,1}^{\boldsymbol{\pi}} = \mathbb{E}\_{\boldsymbol{\pi}} \left[ \sum\_{h=1}^H \sum\_{m=1}^n r\_{h,m}(o\_{h,m}) \right] := V^{\boldsymbol{\pi}},
> $$
> and the goal is for each agent to maximize its own value function. This is equivalent to the goal of our model.
> However, there is no significant difference between them, as they lead to equivalent target algorithms.
> When applying our algorithm to this alternative model, we guarantee that
> $$
> V^{\boldsymbol{\pi}} = \max_{i \in [n]} \left[ \max_{\pi_i^\prime} V^{\pi_i^\prime, \pi_{-i}} \right] = \max_{i \in [n]} \left[ \max_{\pi_i^\prime} V^{\pi_i^\prime, \pi_{-i}}_{i,1} \right],
> $$
> which satisfies the definition of Nash equilibria.
>
> We connect the concept of local optimality to Nash equilibria, showing that local optimality is equivalent to Nash equilibria in an equivalent version of the DEC-POMDP model, which is also widely studied. Moreover, since DEC-POMDP is a special case of a potential game with a partially observable context, our algorithm for achieving local optimality can also be applied to achieve Nash equilibria in potential games with partially observable contexts.
>
> - **Explanation for Theorem 3.1:** Our objective in achieving global optimality is to minimize the regret, which is defined as:
> $$
> \text{Regret}(T) = \sum_{k=1}^T \left( V^{\boldsymbol{\pi}^*} - V^{\boldsymbol{\pi}^k} \right).
> $$
> Furthermore, our goal is to develop an algorithm with regret that is not exponential in the number of agents, addressing what is commonly referred to as the curse of multi-agency. Theorem 3.1 demonstrates that, without any assumptions on the model, the regret scales at least as $\Omega(\sqrt{A^n T})$. This implies that it is impossible to derive an algorithm with non-exponential regret in the general case. Consequently, we focus on deriving algorithms under a factored structure model.

---

> ### Author Response · Authors · 2024-11-20
> **Rebuttal Part 5**
>
> - **Explanation for Theorem 4.1:** While our objective for achieving global optimality is to minimize regret, our objective for achieving local optimality is to minimize the sample complexity required to achieve an $ \epsilon $-approximate local optimal policy. While the targets differ slightly, the algorithm for local optimality does not directly follow from the global optimality algorithm. However, we can still derive the local optimality algorithm from the global optimality algorithm with only minor modifications. Specifically, we apply Algorithm 1 by adjusting the parameter and policy selection to
> $$
> (\theta\_1^k, \theta\_2^k, \ldots, \theta\_n^k, \pi\_m^k) = \arg\max\_{\hat{\theta}\_1 \in \mathcal{B}^k\_1, \hat{\theta}\_2 \in \mathcal{B}^k\_2, \ldots, \hat{\theta}\_n \in \mathcal{B}^k\_n, \pi\_m} V^{\pi\_m, \pi\_{-m}}(\hat{\boldsymbol{\theta}}),
> $$
> while keeping the other procedures unchanged. By combining this with Algorithm 4, we can directly obtain an algorithm for achieving local optimality.
> In this case, the number of samples required is
> $$
> K = \tilde{\mathcal{O}}\left( S^4 A^{2d + 2} \left( S^2 A^{d + 1} + SO \right) \cdot \text{poly}(H) / (\alpha^4 \epsilon^3) \right).
> $$
> In contrast, the sample complexity in Theorem 4.1 is
> $$
> K = \tilde{\mathcal{O}}\left( \sum_{m=1}^n S^4 O^2 A^{2d_m + 2} (S^2 A^{d + 1} + SO) \cdot \text{poly}(H) / (\alpha^4 \epsilon^3) \right).
> $$
> Since $d_m \leq d$ for all $m \in [n]$, Theorem 4.1 reduces the sample complexity. We illustrate this with an example in Figure 1, where directly applying the global optimality framework results in a sample complexity that scales as $\mathcal{O}(A^{12})$, while Theorem 4.1 only requires $\mathcal{O}(A^8)$.
>
> - **Discussion on Mean-Field Method:** Thank you for this valuable suggestion. We have added an additional discussion about the mean-field method in the updated version of our paper. We discuss the mean-field method and its relationship to our work as follows:
>
> In traditional multi-agent systems, the decision-making of each agent depends on the actions of other agents, which leads to exponential growth in the action space. The mean-field method simplifies this by averaging the behavior of other agents, assuming that each agent's decision is influenced by the mean field (i.e., the average behavior of other agents) rather than by the individual actions of each agent.
> Previous works that use the mean-field method to address exponential growth in multi-agent RL include [6], [7], and [8]. In [9], it is shown that if all agents are homogeneous and exchangeable, mean-field control can provide a good approximation to an $N$-agent problem. Similarly, [10] provides a comparable approximation for a $K$-class

---

> ### Author Response · Authors · 2024-11-20
> **Reference**
>
> **Reference**
>
>
> [1] Kara, A. and Yuksel, S. (2022). Near optimality of memory-finite feedback policies in partially observed markov decision processes. Journal of Machine Learning Research.
>
> [2] Kara, A. D. and Yüksel, S. (2023). Convergence of memory-finite q learning for pomdps and near optimality of learned policies under filter stability. Mathematics of Operations Research.
>
> [3]Liu Q, Chung A, Szepesvári C, et al. When is partially observable reinforcement learning not scary?, Conference on Learning Theory. PMLR, 2022: 5175-5220.
>
> [4] Liu, Qinghua, Csaba Szepesvári, and Chi Jin. "Sample-efficient reinforcement learning of partially observable markov games." Advances in Neural Information Processing Systems 35 (2022): 18296-18308.
>
> [5]Washim Uddin Mondal, Vaneet Aggarwal, Satish V. Ukkusuri, Mean-Field Control based Approximation of Multi-Agent Reinforcement Learning in Presence of a Non-decomposable Shared Global State
>
> [6]Yang, Yaodong, et al. "Mean field multi-agent reinforcement learning." International conference on machine learning. PMLR, 2018.
>
> [7]Pasztor, Barna, Ilija Bogunovic, and Andreas Krause. "Efficient model-based multi-agent mean-field reinforcement learning." arXiv preprint arXiv:2107.04050 (2021).
>
> [8]Qiu, Dawei, et al. "Mean-field multi-agent reinforcement learning for peer-to-peer multi-energy trading." IEEE Transactions on Power Systems 38.5 (2022): 4853-4866.
>
> [9]Haotian Gu, Xin Guo, Xiaoli Wei, and Renyuan Xu. Mean-field controls with Q-learning for cooperative
> MARL: convergence and complexity analysis. SIAM Journal on Mathematics of Data Science, 3(4):1168-1196, 2021.
>
> [10]Washim Uddin Mondal, Mridul Agarwal, Vaneet Aggarwal, and Satish V Ukkusuri. On the approximation of cooperative heterogeneous multi-agent reinforcement learning (marl) using mean field control (mfc).
> Journal of Machine Learning Research, 23(129):1-46, 2022a.
>
> [11]Mannor, S. and Tsitsiklis, J. N. (2004). The sample complexity of exploration in the multi-armed
> bandit problem. Journal of Machine Learning Research, 5 623-648

---

> > ### Comment · Reviewer_mApi · 2024-11-26
> >
> > I appreciate the authors for their detailed responses. I will increase my score.

---

### Official Review · Reviewer_ntUa · 2024-11-03

**Soundness:** 3
**Presentation:** 3
**Contribution:** 2
**Rating:** 5
**Confidence:** 3

**Summary:**

The paper studies how to develop provable sample-efficient algorithm for RL in Dec-POMDPs without suffering from the curse of multi-agency.

**Strengths:**

The strength is that the paper is studying an interesting question that is untouched by existing literature by identifying some interesting findings on certain assumptions or structures that can lead to sample efficient RL algorithm without exponential dependency on $n$.

**Weaknesses:**

The main weakness of the paper lies in that 1) some assumptions are fully justified. For example, the memory-less policy is supported by two papers of Kara and Yuksel. However, the two papers actually studies finite-memory-based policy of length, sat $L$. I believe memory-less settings are much more restrictive settings. Another example is the form of the reward functions, which seems to be designed to facilitate the theory analysis. 2) The results on the factored model are a bit incremental, which is an extension of the studies on factored MDP/MG with recently identified weakly-revealing assumption.

**Questions:**

see the weakness

---

> ### Author Response · Authors · 2024-11-20
> **Rebuttal Part 1**
>
> We sincerely thank Reviewer ntUa for their valuable feedback on our work. We now address your questions as follows:
>
> - **Discussion on Memoryless Policy:**
>   *Our framework naturally extends to a memory-limited setting with length $L$, as outlined in lines 428-431 of our paper.*
>
>   Specifically, the framework can be readily adapted to scenarios where agents consider observations and actions from the preceding $L$ steps. In such cases, the policy class expands to:
>   $$
>   \\{\\{\otimes\_{m=1}^n \pi\_{h,m}\\}\_{h\in[H]} \mid \pi\_{h,m}:(\mathcal{O}\_m \times \mathcal{A}\_m)^{\min(h,L)-1} \times \mathcal{O}\_m \to \Delta(\mathcal{A}\_m) \\}.
>   $$
>   We modify the equation from lines 1789 to 1795 as follows:
>   $$
>   \mathbb{O}\_{h,i}(o\_{h,i} \mid s\_h) = \sum\_{\\{o\_{h,m}\\}\_{m \in [n] \setminus \\{i\\}}} \mathbb{O}\_h(\mathbf{o}\_h \mid s\_h),
>   $$
>   $$
>   \mathbb{T}\_{h,i,a_{h,i},\dots,a\_{h+L-1,i}}(s\_{h+L} \mid s\_h, o\_{h,i}, \dots, o\_{h+L-1,i}) = \sum\_{\\{(o_{h,m},a\_{h,m})\\}\_{m \in [n] \setminus \\{i\\}}, \ldots, \\{(o\_{h+L-1,m},a\_{h+L-1,m})\\}\_{m \in [n] \setminus \\{i\\}}}
>   $$
>   $$
>   \sum\_{s\_{h+1,\ldots,s\_{h+L-1}}} \prod\_{r=0}^{L-1} \left[\prod\_{j \in [n] \setminus \{i\}} \pi\_{h+r,j}(a\_{h+r,j} \mid o\_{h+r,j})\right] \frac{\mathbb{O}\_{h+r}(\mathbf{o}\_{h+r} \mid s\_{h+r})}{\mathbb{O}\_{h+r,i}(o\_{h+r,i} \mid s\_{h+r})} \mathbb{T}\_{h+r,\mathbf{a}\_{h+r}}(s\_{h+r+1} \mid s\_{h+r}).
>   $$
>   We can similarly modify Equation (20) from lines 1799 to 1804, and define the transition operators $\mathbb{O}\_{h,m}(o\_{h,i},o\_{h,m} \mid s\_h)$ and $\mathbb{T}\_{h,m,a\_{h,i},\dots,a\_{h+L-1,i},a\_{h,m},\dots,a\_{h+L-1,m}}(s\_{h+L} \mid s\_h, o\_{h,i}, \dots, o\_{h+L-1,i}, o\_{h,m}, \dots, o\_{h+L-1,m})$. In this case, the number of episodes needed to achieve local optimality is at most:
>   $$
>   K = \tilde{\mathcal{O}}(S^4 O^{4L} A^{4L} (S^2 A^{2L} O^{2L} + SO^2) \cdot \text{poly}(H) / (\alpha^4 \epsilon^3)).
>   $$
>   Importantly, our framework can be directly generalized to more complex settings than the memoryless policy assumption. However, introducing such complexity in the notation would likely confuse readers, which is why we present the memoryless policy model as a representative case. The analysis for more complicated models follows directly.
>   In addition to clarity, another reason for choosing the memoryless policy model is our aim to provide theoretical explanations for empirical methods like Value Decomposition Networks (VDN). These empirical approaches assume value decomposition and show promising experimental results under this assumption. We also aim to explain why value decomposition ensures the efficiency of the algorithm. Under the memoryless policy assumption, we have the following value decomposition:
>   $$
>   V\_n^{\pi\_n, \boldsymbol{\pi}\_{-n}}(\theta\_n) = \sum\_{\tau\_{H,n}} \mathbb{P}\_{\theta_n}^{\pi_n}(\tau\_{H,n}) \cdot \left(\sum\_{h=1}^H r\_h(o\_{h,n})\right),
>   $$
>   $$
>   V_m^{\pi_n, \boldsymbol{\pi}\_{-n}}(\theta_m) = \sum\_{\tau\_{H,m}, \tau\_{H,n}} \mathbb{P}\_{\theta_m}^{\pi_n, \pi_m}(\tau\_{H,m}, \tau\_{H,n}) \cdot \left(\sum\_{h=1}^H r_h(o\_{h,m})\right).
>   $$
>   This decomposition illustrates why imposing the value decomposition assumption is reasonable. Thus, we have find sufficient condition for value decomposition and further show that we can derive sample-efficient guarantee under this assumption.

---

> ### Author Response · Authors · 2024-11-20
> **Rebuttal Part 2**
>
> - **Justification for Factored Model:**
>   *Our algorithm can be generalized from algorithms for factored MGs/MDPs, as there are significant differences between partially observable and fully observable settings.*
>
>   Below are the reasons we believe our approach cannot be directly extended from studies of factored MDP/MG:
>   - **Significant Differences Between Partially Observable and Fully Observable Models:** Initially, there is a significant gap between partially observable and fully observable models, even in the single-agent setting. None of the existing algorithms for partially observable reinforcement learning (POMDP, POMG) can be directly derived from their fully observable counterparts. For example, in the single-agent case, the difference between MDP and POMDP is substantial. The works on POMDPs (e.g., [3, 4, 5]) employ completely different techniques compared to MDP algorithms like UCB-VI and UCB-LSVI. Specifically, POMDP approaches use maximum likelihood estimation to construct confidence intervals and incorporate model parameters such that the total log-likelihood assigned to the data is close to the maximum possible log-likelihood. In contrast, MDP-based methods typically rely on frequency-based techniques to approximate transition probabilities, which are not applicable to partially observable settings.
>   - **Distinct from Studies on Factored MDP/MG:** Secondly, we employ fundamentally different methods compared to works on factored MDP/MG. Recent and promising approaches for factored MDPs, such as those in [1] and [2], use UCB-VI style algorithmic frameworks. However, none of these methods are suitable for partially observable settings, as in DEC-POMDPs the state is not directly observable, and the agent’s policy is based on the entire past trajectory. This introduces significant challenges in both planning and estimating the state transition function. Consequently, we adopt an entirely different technique: optimistic maximum likelihood estimation for estimating both the transition and observation models. We construct a confidence interval using maximum likelihood estimation to incorporate model parameters such that the total log-likelihood is close to the highest achievable value. During the planning phase, we select model parameters that maximize the value function and policies that optimize the corresponding value function, which is completely distinct from the approaches in [1, 2].
>   - **Technical Challenges in Our Model:** The main technical challenge in learning our proposed model lies in estimating its parameters. Specifically, the dimensionality of the model grows as $ \Omega(A^n O^n) $, due to the exponential expansion of the joint action and observation spaces with the number of agents. As a result, the sample complexity for parameter estimation is also subject to $ \Omega(A^n O^n) $ . To address this challenge, we construct separate confidence intervals for each model parameter, which helps mitigate the exponential sample complexity with respect to $ n $, as the dimensionality of each parameter $ \theta_m $ does not grow exponentially with $ n $ . Additionally, we implement a carefully designed sampling strategy, instead of directly sampling from the true transition model. This allows us to precisely control the statistical error in joint trajectory probability estimation by independently managing the error of each trajectory operator.

---

> ### Author Response · Authors · 2024-11-20
> **Reference**
>
> [1] Chen, Xiaoyu, et al. "Efficient reinforcement learning in factored mdps with application to constrained rl." arXiv preprint arXiv:2008.13319 (2020).
>
> [2] Tian, Yi, Jian Qian, and Suvrit Sra. "Towards minimax optimal reinforcement learning in factored markov decision processes." Advances in Neural Information Processing Systems 33 (2020): 19896-19907.
>
> [3]Liu Q, Chung A, Szepesvári C, et al. When is partially observable reinforcement learning not scary?, Conference on Learning Theory. PMLR, 2022: 5175-5220.
>
> [4] Liu, Qinghua, Csaba Szepesvári, and Chi Jin. "Sample-efficient reinforcement learning of partially observable markov games." Advances in Neural Information Processing Systems 35 (2022): 18296-18308.
>
> [5] Chen, Fan, Yu Bai, and Song Mei. "Partially observable rl with b-stability: Unified structural condition and sharp sample-efficient algorithms." arXiv preprint arXiv:2209.14990 (2022).

---

> ### Comment · Reviewer_ntUa · 2024-11-21
> **Thanks for your response.**
>
> To my understanding, the intuitive benefit of memoryless policy is that if all the other agents $-i$ are using a fixed memoryless policy $\pi_{-i}$, agent $i$ is equivalently solving an POMDP with $s_h, o_{h, i}$ as the Markovian state. However, it seems not the case for L-step finite memory. In other words, $(s\_h, o\_{h,i}, \dots, o\_{h+L-1,i})$ seems not to be Markovian.
>
> Specifically, in your extension, the following extension on the transition seems still assuming all the other agents are using a memoryless policy.
>
> $$\mathbb{T}\_{h,i,a_{h,i},\dots,a\_{h+L-1,i}}(s\_{h+L} \mid s\_h, o\_{h,i}, \dots, o\_{h+L-1,i}) = \sum\_{\{(o\_{h,m},a\_{h,m})\}_{m \in [n] \setminus \{i\}}, \ldots, \{(o\_{h+L-1,m},a\_{h+L-1,m})\}\_{m \in [n] \setminus \{i\}}} $$ $$ \sum\_{s\_{h+1,\ldots,s\_{h+L-1}}} \prod\_{r=0}^{L-1} \left[\prod\_{j \in [n] \setminus {i}} \pi\_{h+r,j}(a\_{h+r,j} \mid o\_{h+r,j})\right] . \frac{\mathbb{O}\_{h+r}(\mathbf{o}\_{h+r} \mid s\_{h+r})}{\mathbb{O}\_{h+r,i}(o\_{h+r,i} \mid s\_{h+r})} \mathbb{T}\_{h+r,\mathbf{a}\_{h+r}}(s\_{h+r+1} \mid s\_{h+r})$$
>
> i.e., it has the term expressed as $\pi\_{h+r,j}(a\_{h+r,j} \mid o\_{h+r,j})$, which is memoryless.
>
> I would like to see a detailed explanation on how to fix the issue.

---

> > ### Author Response · Authors · 2024-11-25
> > **Follow up Response Part 1**
> >
> > Thank you for your response. We first apologize and acknowledge that the explanation in our initial response regarding the memory-finite policy applies only to the case where the central agent adopts a memory-finite policy. Without further assumptions on the model, we are unable to address the case where all agents adopt a memory-finite policy of length $L$. Below, we provide an explanation to address this issue:
> >
> > - **Add other additional assumption on the transition model:**
> >   One way to develop a sample-efficient algorithm for policies with length $L$ is to introduce additional assumptions on the transition model. Specifically, if we assume that the model dimension (or more precisely, the eluder dimension of the model) does not scale exponentially with $n$, we can develop a sample-efficient algorithm that only scales exponentially with $L$. For instance, in the general case, the model dimension is $SO^n + S^2 A^n$. However, if we assume that the transition function for each state pair depends on the actions of at most $d$ agents, and that the observations are independent, the model dimension scales as $SO + S^2 A^d$. Other assumptions that reduce the model dimension include the factored structure assumption, as discussed in the first part of our paper.
> >
> >   Thus, when the model dimension does not scale exponentially with $n$, the sample complexity is given by:
> >   $$
> >   K = \tilde{\mathcal{O}}\left(S^4 O^{4L} A^{4L} \left(S^2 A^{2L} O^{2L} + SO^2 \right) \cdot \text{poly}(H) / (\alpha^4 \epsilon^3) \right).
> >   $$
> >   In summary, one way to address the issue is by introducing additional assumptions on the model dimension.

---

> > ### Author Response · Authors · 2024-11-25
> > **Follow Up Response Part 2**
> >
> > - **Justification for Memoryless Model:** We acknowledge the limitations of the memoryless agent model. However, it is a common practice in theoretical works to introduce additional assumptions when addressing fundamental hardness in the general model. Therefore, we believe that studying the memoryless policy model remains well-justified. Below are our justifications for studying this model:
> >
> >   - **Widely-Studied Model in Previous Works:** The memoryless policy model is widely studied in the partially observable RL literature. Several works, including [1, 2, 3], focus exclusively on memoryless policies, and the memoryless partially observable model has broad applications in empirical studies. Therefore, given the fundamental challenges of studying the most general model, we believe that investigating the memoryless agent model is both reasonable and insightful.
> >
> >   - **Theoretical Explanation for Empirical Algorithms:** Our work also aims to provide theoretical insights into empirical methods, such as Value Decomposition Networks (VDN). These empirical approaches assume value decomposition and demonstrate promising experimental results under this assumption. We seek to explain why value decomposition contributes to the efficiency of the algorithm.
> >
> >     Specifically, we aim to identify sufficient conditions for value decomposition and show that under these conditions, a sample-efficient guarantee can be achieved. Under the memoryless policy assumption, we derive the following value decomposition:
> >     $$
> >     V\_n^{\pi\_n, \boldsymbol{\pi}\_{-n}}(\theta\_n) = \sum\_{\tau\_{H,n}} \mathbb{P}\_{\theta\_n}^{\pi\_n}(\tau\_{H,n}) \cdot \left(\sum\_{h=1}^H r\_h(o\_{h,n})\right),
> >     $$
> >     $$
> >     V\_m^{\pi\_n, \boldsymbol{\pi}\_{-n}}(\theta\_m) = \sum\_{\tau\_{H,m}, \tau\_{H,n}} \mathbb{P}\_{\theta\_m}^{\pi\_n, \pi\_m}(\tau\_{H,m}, \tau\_{H,n}) \cdot \left(\sum\_{h=1}^H r\_h(o\_{h,m})\right).
> >     $$
> >     This decomposition illustrates why the value decomposition assumption is reasonable. In conclusion, we have identified sufficient conditions for value decomposition and demonstrated that sample-efficient guarantees can be derived under this assumption.

---

> > ### Author Response · Authors · 2024-11-25
> > **Follow Up Response Part 3**
> >
> > - **Technical Novelty in Memoryless Agent Setting.** We would like to highlight the technical challenges involved in deriving algorithms for the memoryless agent setting, which demonstrates that the problem is non-trivial. Specifically, in developing a sample-efficient algorithm to find $\pi_m^* = \arg\max_{\mu_m} V^{\mu_m, \pi_{-m}}$ with a given $\pi_{-m}$, we face the following challenges：
> >
> >     1. In DEC-MDPs, given $\pi_{-m}$, the model reduces to a single-agent problem with action space $\mathcal{A}_m$. However, this reduction does not apply to DEC-POMDPs, precluding the use of single-agent algorithms for sample-efficient guarantees, as is possible in MDPs.
> >     2. Another challenge arises from the exponential growth of the joint action and observation spaces with the number of agents, leading to a model dimension that scales as $\mathcal{O}(A^n O^n)$. As a result, constructing a single confidence interval to estimate the model parameters results in sample complexity scaling as $\mathcal{O}(A^n O^n)$.
> >
> >     We overcome these challenges by defining the trajectory probability as:
> >     $$
> >     \mathbb{O}\_{h,i}(o\_{h,i} \mid s\_h) = \sum\_{\{o\_{h,m}\}\_{m \in [n] \setminus \\{i\\}}} \mathbb{O}\_h(\mathbf{o}\_h \mid s\_h),
> >     $$
> >
> >     $$
> >     \mathbb{T}\_{h,i,a\_{h,i}}(s\_{h+1} \mid s\_h, o\_{h,i}) = \sum\_{\\{(o\_{h,m}, a\_{h,m})\\}\_{m \in [n] \setminus \\{i\\}}} \left[\prod\_{j \in [n] \setminus \\{i\\}} \pi\_{h,j}(a\_{h,j} \mid o\_{h,j})\right] \frac{\mathbb{O}\_h(\mathbf{o}\_h \mid s\_h)}{\mathbb{O}\_{h,i}(o\_{h,i} \mid s\_h)} \mathbb{T}\_{h,\mathbf{a}\_h}(s\_{h+1} \mid s\_h).
> >     $$
> >     We then assign parameters for the trajectory probability of each agent, estimating and updating these parameters with separate confidence intervals. Since the dimension of the parameter $\theta_m$ ($m \in [n]$) is at most $H(S^2 A^2 O^2 + SO^2) + S$, we achieve an $\epsilon$-approximate local optimum with a sample complexity that avoids exponential scaling in $n$.
> >
> > Based on the discussion above, we acknowledge the limitations of the memoryless agent context. However, we still believe that studying the memoryless agent setting is well justified, and that the algorithm proposed for this setting presents significant technical novelty and valuable insights into empirical algorithms for DEC-POMDPs.
> >
> > [1] Azizzadenesheli, Kamyar, Alessandro Lazaric, and Animashree Anandkumar. "Open problem: Approximate planning of pomdps in the class of memoryless policies." Conference on Learning Theory. PMLR, 2016.
> >
> > [2] Steckelmacher, Denis, et al. "Reinforcement learning in POMDPs with memoryless options and option-observation initiation sets." Proceedings of the AAAI conference on artificial intelligence. Vol. 32. No. 1. 2018.
> >
> > [3] Li, Yanjie, Baoqun Yin, and Hongsheng Xi. "Finding optimal memoryless policies of POMDPs under the expected average reward criterion." European Journal of Operational Research 211.3 (2011): 556-567.

---

### Official Review · Reviewer_Mjwf · 2024-11-05

**Soundness:** 2
**Presentation:** 3
**Contribution:** 3
**Rating:** 5
**Confidence:** 2

**Summary:**

This paper studies how to learn to solve decentralized partially observable Markov decision processes (DEC-POMDP). It shows that the sample complexity for arbitrary problems is exponential in the number of agents.

To make the problem more tractable, the paper considers two different assumptions: first, a factorization of the transition function and the observation function, and second, a class of memoryless policies.

Finally, the paper provides a MARL algorithm for each assumption and analyzes the learning regret of each.

**Strengths:**

- Clear theoretical claims.
- The paper does an excellent work interleaving the main core results and intuition about them.

**Weaknesses:**

- **There are no proofs in the main document.** The paper delegates all the proofs to the supplemental material. However, I was not able to read the supplemental material. Therefore, as this paper's main contribution is theoretical, this review must be taken with care.
- **Narrow class of problems.** Although the paper provides an example of the methodology's application to a POMDP with Knapsack constraints, it does not discuss which types of multi-agent problems the assumptions made would be applicable.
- **Lack of empirical evidence.** The paper provides no empirical support to validate the proposed algorithm.

**Questions:**

1. Could you clarify how the notion of regret fits the centralized training with decentralized execution described in the introduction?
2. Some prior work considers a similar assumption on the max in-degree $d$ of the factored MDPs [1,2]. Does it have a similarity with this approach? Would it be possible to make a similar analysis by making such assumptions [3]?
3. Is there a reason to assume that the components of the factored transition function depend only on the actions of the other agents and not their local states?


[1] Diuk, C., Li, L., and Leffler, B. R. (2009). The adaptive *k*-meteorologists problem and its application to structure learning and feature selection in reinforcement learning. *ICML*, 249–256. <https://doi.org/10.1145/1553374.1553406>
[2 ] Strehl, A. L., Diuk, C., and Littman, M. L. (2007). Efficient structure learning in factored-state MDPs. *AAAI*, 645–650. <http://www.aaai.org/Library/AAAI/2007/aaai07-102.php>
[3] Chakraborty, D., and Stone, P. (2011). Structure learning in ergodic factored MDPs without knowledge of the transition function’s in-degree. *ICML*, 737–744. <https://icml.cc/Conferences/2011/papers/418_icmlpaper.pdf>

---

> ### Author Response · Authors · 2024-11-20
> **Rebuttal Part 1**
>
> We sincerely thank Reviewer Mjwf for their valuable feedback on our work. We now address your concerns as follows:
>
> ## Clarification for the Weakness
>
> - **Explanation for the Proofs:**
>   *We apologize for leaving the proof in the supplementary details and the inconvenience it might cause. We choose to leave the proof in the Appendix since it is common for works on learning theory to leave the complete proofs in the appendix.*
>
>   Examples of this type of theoretical works in recent ICLR conference include [1,2,3,4] (we select these papers as examples since they also work on the theory of reinforcement learning). Due to the space limit, highlighting the main result in the main text and leaving the complete proof in the appendix better helps the reader grasp the important contribution of the work, which is quite a natural practice for theoretical works. For the convenience of readers, we submitted a complete PDF version of our paper in the initial submission, with the appendix following the main text. We will consider providing a sketch of the proof in the main text to help readers verify the theoretical results.
>
> - **Explanation for the Narrow Class:**
>   *The Factored Structure model is commonly observed in multi-agent problems.*
>
>   We present an algorithm for factored structure models, as such models are common in reinforcement learning (RL) studies. In addition to the POMDP with knapsack constraints, an excellent example of a factored structure model is provided by [5], which describes a large production line with $d$ machines arranged sequentially, where machine $i$ has $S_i$ possible states and $O_i$ possible observations. At each step, each machine is influenced only by its neighbors. Specifically, in this case, we have $\mathsf{pa}(m) = \{m-1, m+1\}$ for all $2 \leq m \leq d-1$, with $\mathsf{pa}(2) = \{1\}$ and $\mathsf{pa}(d) = \{d-1\}$. [6] also presents a factored structure model in the context of robotics, where the transitions of different robotic body parts are relatively independent. As a result, the joint transition dynamics of these body parts can be expressed in a factored form.
>
>   In our paper, we first illustrate that it is impossible to derive sample-efficient algorithms for DEC-POMDPs without any assumptions on the model. Therefore, following a common approach in theoretical studies, we derive algorithms under reasonable assumptions on the model. Given the wide applicability of factored models, we derive a non-exponential algorithm in this setting.
>
>   Thank you for your valuable feedback; we will include these motivations in the revised version of our paper.
>
> - **Explanation for Experiment:**
>   We include mostly theoretical results in our paper for two main reasons. First, due to the computational efficiency challenges (as explained in [7]), almost all previous theoretical studies on partially observable reinforcement learning have focused exclusively on theoretical analysis. Notable examples of such theoretical works include [7,8,9,10,11]. In line with this approach, we also emphasize theoretical analysis. The second reason is that the goal of our work is to provide a theoretical understanding of well-known empirical algorithms such as VDN. Specifically, while empirical studies often impose value decomposition on the model, our work demonstrates that we can derive non-exponential sample-efficient guarantees.
>
>   Therefore, we believe that focusing on theoretical results better highlights our aim of offering a theoretical foundation for previously proposed empirical algorithms.

---

> ### Author Response · Authors · 2024-11-20
> **Rebuttal Part 2**
>
> ## Clarification for the Questions
>
> - **Explanation for the Regret:** The aim of our work is to provide a theoretical understanding of empirical studies. To achieve this, we consider the setting in [12], one of the most influential empirical works on DEC-POMDPs. In [12], the policies are trained in a centralized manner, but the execution of the policy is decentralized. Specifically, during the training process, we have access to the trajectory histories of all agents. However, the output policy $\boldsymbol{\pi} = \pi_1 \times \pi_2 \times \cdots \times \pi_n$ satisfies the condition that $\pi_m: \\{ (\mathcal{O}_m \times \mathcal{A}_m)^{h-1} \times \mathcal{O}_m \rightarrow \Delta_m \\}\_{h \in [H]} $  for all agents $m \in [n]$. This means that the action selection of agent $m$ only depends on its own trajectory.
>
>   In [12], the goal is to maximize the expected cumulative total reward of all agents:
>   $$V^{\boldsymbol{\pi}} = \mathbb{E}\_{\boldsymbol{\pi}} \left[ \sum\_{h=1}^H \sum\_{m=1}^n r\_{h,m}(o\_{h,m}) \right].$$
>
>   Following the setup in [12], we define regret as:
>   $$\text{Regret}(T) = \sum_{k=1}^T \left( V^{\boldsymbol{\pi}^*} - V^{\boldsymbol{\pi}^k} \right),$$
>   which compares our output policy with the best policy that maximizes the expected cumulative total reward. We also train our policy in a centralized manner, where we estimate the transition model and select the policy using the historical trajectory information of all agents. The output is a decentralized policy, where each agent selects its actions based solely on its own trajectory information.
>
> - **Discussion on Related Work:** We sincerely thank you for bringing these valuable related works to our attention. In the modified version of our paper, we add the discussion of these works in our paper. We will compare these works with our study here in detail:
>
>   - [13] proposes an algorithmic framework based on KWIK for learning probabilistic concepts. The paper applies their algorithm, $k$-Meteorologists, to the reinforcement learning problem in factored models. While [13] focuses mainly on the experimental performance of the algorithm, our work primarily focuses on the theoretical aspects and addresses the problem involving $n$ agents. [13] provides empirical insights suggesting that more efficient algorithms can be derived when restricted to factored structure models.
>
>   - [14] addresses the problem of reinforcement learning in factored MDPs and proposes an efficient algorithm using dynamic Bayesian networks (DBNs). [14] also compares a factored structure for state transitions similar to ours. Additionally, like Algorithm 1 in our paper, their algorithmic framework consists of two main parts: (1) estimating the transition model and value function (lines 1-5 in SLF-Rmax Algorithm), and (2) selecting the optimal policy (lines 7-14 in SLF-Rmax Algorithm). Our approach differs from [14] in both methodology and model. Specifically, we consider a more complex model involving multiple agents and partial observability. In terms of methodology, we estimate the transition model and value function using a different approach. Specifically, we construct a confidence interval for the model parameters such that the total log-likelihood assigned to the data is close to the maximum possible value. We then select the model parameters that maximize the corresponding value function as the estimate of the true model parameters.
>
>   - While we focus on the decentralized partially observable Markov decision process with rewards, aiming to derive an algorithm that maximizes the expected total cumulative reward for all agents, [15] addresses a completely different but also very important problem in factored model. They consider a factored state MDP, their goal is to provide an algorithm that guarantees a return in factored MDPs close to the optimal return. Since factored-state MDPs can be considered a special case of Markov chains, they apply the properties of ergodic stochastic processes to factored MDPs. In [15], they impose an additional assumption (Condition 1) stating that in factored-state MDPs, to achieve an expected return of $T(\epsilon)$ steps that exceeds $U^\star - 3\epsilon$ in the true FMDP from any start state, planning must be based on a $\widehat{P}$ derived from the correct set of parent factors for each $P_{i,a}$.
>
>   In our work, since we consider a fundamentally different problem from [15], there is no need to impose such assumptions on the stopping time. We impose an alternative assumption on the revealing condition and propose an algorithm that breaks the curse of multi-agency.

---

> ### Author Response · Authors · 2024-11-20
> **Rebuttal Part 3**
>
> - **Justification for Factored Model:**
>   *Our brief response is that our algorithmic framework extends to scenarios where the transition function depends on the local states of other agents. We present our current approach as a representative example to introduce the algorithm in a manner that is more accessible to the readers.*
>
>   Our algorithm can indeed be generalized to the setting where the transition function depends on both the actions and local states of other agents. We consider the following transition factorization:
>   $$\mathbb{T}_h(\mathbf{s}^\prime \mid \mathbf{s}, \mathbf{a}) = \prod_{m=1}^n \mathbb{T}_{h,m}(s^\prime \mid s_m, s_{\mathsf{pas}(m)}, a_m, a_{\mathsf{paa}(m)}),$$
>   where $\mathsf{pas}(m) \subset [n]$ represents the set of agents whose local states influence the transition of agent $m$, and $\mathsf{paa}(m) \subset [n]$ represents the set of agents whose actions influence the transition of agent $m$.
>
>   Next, we define $\max_{m \in [n]} |\mathsf{pas}(m)| := d_s = \mathcal{O}(1)$ and $d_a = \max_{m \in [n]} |\mathsf{paa}(m)| = \mathcal{O}(1)$. With these definitions, the regret is bounded as follows in dag setting:
>   $$\text{Regret}(k) \leq \tilde{\mathcal{O}}(\alpha^{-2} S^{d_s+2} O A^{d_a+1} \sqrt{k(S^{d_s+2} A^{d_a+1} + SO)}).$$
>
>   Furthermore, we can consider the case where the transition is influenced by the local states, observations, and actions of other agents. In this case, we have:
>   $$\mathbb{T}\_h(\mathbf{s}^\prime \mid \mathbf{s}, \mathbf{a}) = \prod\_{m=1}^n \mathbb{T}\_{h,m}(s^\prime \mid s\_m, s\_{\mathsf{pas}(m)}, a_m, a\_{\mathsf{paa}(m)}, o\_{\mathsf{pao}(m)}),$$
>   where $\mathsf{pao}(m) \subset [n]$ represents the set of agents whose observations influence the transition of agent $m$. In this case, the regret is bounded by:
>   $$\text{Regret}(k) \leq \tilde{\mathcal{O}}(\alpha^{-2} S^{d_s+2} O^{d_o+1} A^{d_a+1} \sqrt{k(S^{d_s+2} A^{d_a+1} O^{d_o} + SO)}),$$
>   where we define $\max\_{m \in [n]} |\mathsf{pao}(m)| := d_o = \mathcal{O}(1)$.
>
>   Moreover, we can also consider the case where the observation is influenced by the actions of other agents. Specifically, we have:
>   $$\mathbb{O}\_h(\mathbf{o} \mid \mathbf{s}, \mathbf{a}) = \prod\_{m=1}^n \mathbb{O}\_{h,m}(o_m \mid s_m, a_{\mathsf{paao}(m)}),$$
>   where $\mathsf{paao}(m) \subset [n]$ represents the set of agents whose actions influence the observation of agent $m$. In this case, the regret is bounded by:
>   $$\text{Regret}(k) \leq \tilde{\mathcal{O}}(\alpha^{-2} S^{d_s+2} O^{d_o+1} A^{d_a+1} \sqrt{k(S^{d_s+2} A^{d_a+1} O^{d_o} + SOA^{d{ao}})}),$$
> where we define $\max_{m \in [n]} |\mathsf{paao}(m)| := d_{ao} = \mathcal{O}(1)$.
>
> Our algorithmic framework is not necessarily limited to the case presented in Section 4.1. We introduced a simplified model for clarity, as it is easier for the reader to follow. Including all dependencies with respect to the observation and local states would result in a cumbersome notation. Given the variety of definitions for factored models, we have selected the most representative one that clearly demonstrates our algorithm, framework, and analysis. The analysis for other models follows similarly.

---

> ### Author Response · Authors · 2024-11-20
> **Rebuttal Part 4**
>
> **Reference**
>
> [1] Faster Last-iterate Convergence of Policy Optimization in Zero-Sum Markov Games (ICLR 2023), Shicong Cen, Yuejie Chi, Simon S. Du, Lin Xiao
>
> [2] Variance-Aware Sparse Linear Bandits (ICLR 2023),Yan Dai, Ruosong Wang, Simon S. Du
>
> [3]Learning Rationalizable Equilibria in Multiplayer Games (ICLR 2023), Yuanhao Wang, Dingwen Kong, Yu Bai, Chi Jin
>
> [4]Representation Learning for General-sum Low-rank Markov Games (ICLR 2023), Chengzhuo Ni, Yuda Song, Xuezhou Zhang, Chi Jin, Mengdi Wang
>
> [5] Near-optimal reinforcement learning in factored mdps. Advances in Neural Information Processing Systems, 27, Osband, I. and Van Roy, B. (2014)
>
> [6] Efficient reinforcement learning in factored mdps with application to constrained rl. arXiv preprint arXiv:2008.13319, Chen, X. , Hu, J. , Li, L. and Wang, L. (2020).
>
> [7]Partially Observable RL with B-Stability: Unified Structural Condition and Sharp Sample-Efficient Algorithms Fan Chen, Song Mei, Yu Bai (2022). ICLR 2023
>
> [8]Sample-Efficient Reinforcement Learning of Partially Observable Markov Games, Qinghua Liu, Csaba Szepesvári, Chi Jin, Neural Information Processing Systems (NIPS) 2022.
>
> [9]Optimistic MLE-A Generic Model-based Algorithm for Partially Observable Sequential Decision Making. Qinghua Liu, Praneeth Netrapalli, Csaba Szepesvari, Chi Jin, Annual ACM Symposium on Theory of Computing (STOC) 2023.
>
> [10] When Is Partially Observable Reinforcement Learning Not Scary? Qinghua Liu, Alan Chung, Csaba Szepesv ́ari, Chi Jin, Conference of Learning Theory (COLT) 2022.
>
> [11] Sample-Efficient Reinforcement Learning of Undercomplete POMDPs, Chi Jin, Sham M. Kakade, Akshay Krishnamurthy, Qinghua Liu, Neural Information Processing Systems (NeurIPS), 2020
>
> [12] Monotonic value function factorisation for deep multi-agent reinforcement learning[J]. Rashid T, Samvelyan M, De Witt C S, et al. Journal of Machine Learning Research, 2020, 21(178): 1-51.
>
> [13] Diuk, C., Li, L., and Leffler, B. R. (2009). The adaptive k-meteorologists problem and its application to structure learning and feature selection in reinforcement learning. ICML, 249-256
>
> [14] Strehl, A. L., Diuk, C., and Littman, M. L. (2007). Efficient structure learning in factored-state MDPs. AAAI, 645-650.
>
> [15]Chakraborty, D., and Stone, P. (2011). Structure learning in ergodic factored MDPs without knowledge of the transition function's in-degree. ICML, 737-744.

---

> > ### Comment · Reviewer_Mjwf · 2024-11-26
> >
> > I appreciate the authors' careful responses to my questions. However, as noted in my review, I could not evaluate the paper's main claims since the proofs are provided only in the supplemental material, which I did not have time to read. Consequently, I decided to maintain my recommendation, as I cannot give a positive recommendation without a properly assessing the paper's claims.

---

### Meta-Review · Area_Chair_s2EW · 2024-12-20

**Metareview:**

The reviewers for this paper have diverse assessments, with overall ratings for the paper being borderline. On the positive side, the reviewers acknowledged that the paper tackles an important challenge related to the curse of multiagency in decentralized partially observable MDPs, and the proposed algorithms and theoretical analysis make valuable contributions to the area of cooperative multi-agent RL. On the negative side, several reviewers raised common concerns related to strong assumptions/settings for the proposed algorithms, lack of proofs in the main paper, and lack of empirical evaluation. We want to thank the authors for their responses and active engagement during the discussion phase. These responses did help in improving the reviewers' assessment of the paper; however, the paper's ratings still stand borderline. Nevertheless, this is potentially impactful work, and we encourage the authors to incorporate the reviewers' feedback when preparing a future paper revision.

**Additional Comments On Reviewer Discussion:**

This is a borderline paper leaning toward rejection. The reviewers pointed out several weaknesses in the paper, and raised common concerns related to strong assumptions/settings for the proposed algorithms, lack of proofs in the main paper, and lack of empirical evaluation. There was an extensive discussion among the authors and reviewers, though the raised concerns remain.

---

### Decision · Program_Chairs · 2025-01-22

Reject